# QUANTIFYING CROSS-DOMAIN KNOWLEDGE DISTILLATION IN THE PRESENCE OF DOMAIN SHIFT

## ABSTRACT

Cross-domain knowledge distillation often suffers from domain shift. Although domain adaptation methods have shown strong empirical success in addressing this issue, their theoretical foundations remain underdeveloped. In this paper, we study knowledge distillation in a teacher–student framework for regularized linear regression and derive high-dimensional asymptotic excess risk for the student estimator, accounting for both covariate shift and model shift. This asymptotic analysis enables a precise characterization of the performance gain in cross-domain knowledge distillation. Our results demonstrate that, even under substantial shifts between the source and target domains, it remains feasible to identify an imitation parameter for which the student model outperforms the student-only baseline. Moreover, we show that the student's generalization performance exhibits the double descent phenomenon.

## 1 INTRODUCTION

The success of modern machine learning tasks typically requires the availability of large-scale labeled datasets. However, collecting labeled data for a new target task is often challenging and expensive. When data in the target domain is scarce, it is possible to leverage labeled data from related source domains. Knowledge distillation (KD) (Hinton et al., 2015), originally proposed for model compression, is a popular technique that transfers knowledge from a capable teacher model trained on a source domain to a smaller student model. This is achieved by guiding the student model to mimic the teacher model's outputs. The extra information in the teacher's predictions often improves the student model's performance when target domain data is limited. KD has recently achieved remarkable success across several fields including image classification (Radford et al., 2021; Li et al., 2024), speech recognition (Mingote et al., 2020), and language models (Gu et al., 2023; Agarwal et al., 2024).

We denote the source domain data and target domain data as $(\mathbf{X}_1, \mathbf{y}_1)$ and $(\mathbf{X}_2, \mathbf{y}_2)$, respectively. This work focuses on the following cross-domain KD process: a teacher model is first trained on the source domain data, and its predicted labels for the target domain inputs are then used to supervise the training of the student model by minimizing the per-sample objective function,

$$\mathcal{L}(\xi) = \xi\ell(y_2^{\mathsf{t}}, y_2^{\mathsf{s}}) + (1 - \xi)\ell(y_2, y_2^{\mathsf{s}}), \tag{1}$$

where $\ell$ denotes the loss function, $y_2$ is the ground-truth label, $y_2^{\mathsf{t}}$ is the teacher's predicted label, and $y_2^{\mathsf{s}}$ denotes the student's prediction. The weight parameter $\xi$, known as the imitation parameter (Lopez-Paz et al., 2015), balances the contributions of the teacher's predictions and the observed labels during training. We summarize the key findings of this paper in the following informal lemma.

**Lemma 1.** *(informal) Under mild conditions, the excess risk in linear regression with quadratic loss admits a unique minimizer $\xi^*$, which can be negative.*

Cross-domain KD often suffers from a *shift* between the source and target domains. For instance, the source domain may consist of standard American English speech, while a region-specific voice assistant must handle local dialects. Another example is a face detection model trained on images of light-skinned individuals (source domain) being applied to images of dark-skinned individuals (target domain). Learning a discriminative predictor under such domain shifts between source and target domains is known as domain adaptation (Ganin et al., 2016). While much of the literature on

domain adaptation has focused on improving the performance of KD, relatively little is understood about when – and how effectively – the student model can learn from the teacher in the presence of domain shift.

Recently, Emrullah Ildiz et al. (2025) analyzed the weak-to-strong (W2S) generalization of linear models in a cross-domain setting, and identified the form of the optimal surrogate model. However, their analysis relies on the condition that the covariance matrices of the source and target domains are jointly diagonalizable, which limits its ability to capture the influence of eigenvectors. Moreover, their results are restricted to the setting $\xi = 1$ (i.e., pure teacher supervision), leaving the trade-off between distillation and learning from observed student data unexplored. Motivated by these limitations, we take a step toward a more complete understanding of the performance gains of cross-domain KD for linear regression.

In this paper, we present a theoretical analysis of cross-domain KD in the context of linear models, leveraging tools from random matrix theory. For ridge regression, we study two cases: (i) a deterministic-parameter setting, in which the teacher and student parameter vectors are non-random; and (ii) a random-parameter setting, in which a shared parameter vector is drawn from a prior distribution. We also extend our analysis to the ridgeless regression setting. All proofs of the theoretical results are provided in the appendix. We summarize our contributions as follows:

- **High-dimensional risk characterization.** We derive precise high-dimensional asymptotics for the risk of cross-domain KD via a bias–variance decomposition. Our results reveal how the excess risk depends on the parameter vectors and the input distributions in both domains, generalizing the student-only setting of Hastie et al. (2022).

- **Utility of cross-domain KD.** (*"Stones from other mountains can polish jade"*). Intuitively, large domain shifts between the teacher's and student's training data might limit – or even harm – the value of teacher supervision for the student's generalization. Surprisingly, our analysis shows that even under substantial domain discrepancies, it is still possible to find an $\xi \in \mathbb{R}$ such that the student model can outperform the student-only baseline. The existence of such $\xi$ depends on the geometry of the models and the covariance matrices of both domains.

- **Double descent phenomenon.** We observe that the excess risk, as a function of the dimension-to-sample-size ratio, exhibits the double-descent phenomenon in KD for teacher-student model – previously documented by Hastie et al. (2022); Nakkiran et al. (2021) in student-only models, and by Moniri & Hassani (2025) for $\xi = 1$ under no domain shift with isotropic covariance.

## 1.1 RELATED WORKS

**Theory of KD.** In recent years, a growing body of work has sought to understand the effects of KD. The theoretical understanding of distillation began with Phuong & Lampert (2019), who initially investigated linear student networks. Wei et al. (2021); Borup & Andersen (2021); Das & Sanghavi (2023); Pareek et al. (2024); Jeong & Chung (2025) theoretically studied self-distillation, a variant of KD in which the student model has the same architecture as the teacher and is trained on the same data. Menon et al. (2021) showed that a "Bayes teacher" providing true class probabilities can reduce the variance of the student's objective, leading to improved performance. Harutyunyan et al. (2023) proposed a framework that highlighted a delicate interplay among the teacher's accuracy, the student's margin with respect to the teacher predictions, and the complexity of the teacher predictions. From an information-theoretic perspective, Dissanayake et al. (2025) quantified and explained the transferred knowledge and knowledge left to distill for a downstream task.

**Cross-domain KD and domain adaptation.** Many studies have explored various methods to address the domain shift problem in the field of KD. Empirical works include Su & Maji (2016); Kundu et al. (2019); Asami et al. (2017); Li et al. (2023); Xu et al. (2024); Tang et al. (2025). Ye et al. (2024) proposed the Maximum Conditional Mutual Information method, which enables the teacher model to capture more contextual information to generate more accurate estimates of the Bayes conditional probability distribution. The emergence of large language models (LLMs) has brought new advancements, such as distillation across vastly different architectures and scalable cross-domain transfer. For more details, readers may refer to Fedus et al. (2022); Ouyang et al. (2022); Yang et al. (2024). From a theoretical perspective, Emrullah Ildiz et al. (2025) focused

on the setting where the student is trained using only the teacher's predictions, and analyzed the conditions under which the student can outperform the teacher in cross-domain KD.

**Weak-to-strong generalization**. Weak-to-strong (W2S) generalization (Burns et al., 2024), which concerns using predictions generated by a weaker teacher model to train a more powerful student model, is closely connected to KD. Emrullah Ildiz et al. (2025) provided an analysis of ridgeless regression and proved that when using a weak model as the surrogate (teacher), W2S training can provably outperform training with true labels. Charikar et al. (2024) assumed that the models are selected over a convex set, and quantified the gain of the weak-label trained strong model over the weak model. Wu & Sahai (2025) explored W2S generalization for classification in a spiked covariance model. Medvedev et al. (2025) explained how W2S generalization can arise in random feature models described by two-layer networks. Theoretical research in this area has continued to grow, see Dong et al. (2025); Shin et al. (2025); Moniri & Hassani (2025); Oh et al. (2025), for example.

## 1.2 NOTATIONS

We use $\|\cdot\|$ to denote the spectral norm for matrices and the Euclidean norm for vectors, and $\|\cdot\|_{\mathsf{F}}$ for the Frobenius norm of a matrix. Standard big-O and small-o notations are employed. Moreover, we denote $x_n = o_{a.s.}(a_n)$, if $x_n/a_n \to 0$ almost surely. For any sequences $a_n \geq 0$ and $b_n \geq 0$, we write $a_n \lesssim b_n$ if $a_n = O(b_n)$, and $a_n \sim b_n$ if both $a_n \lesssim b_n$ and $b_n \lesssim a_n$. We use $\delta(\cdot)$ to denote the indicator function, which takes the value 1 if the condition $\cdot$ holds, and 0 otherwise. Throughout the paper, $c$ and $C$ denote constants that may vary from line to line. For a random variable $x$, we use $x \sim D$ to indicate that $x$ follows the distribution $D$.

## 2 PRELIMINARIES

### 2.1 PROBLEM SETUP

Suppose there are $N_1$ covariates $\{\mathbf{x}_j^{(1)}\}_{j=1}^{N_1}$ drawn i.i.d. from an $M$-dimensional source distribution $D_1$ and $N_2$ covariates $\{\mathbf{x}_j^{(2)}\}_{j=1}^{N_2}$ drawn i.i.d. from an $M$-dimensional target distribution $D_2$. We consider a linear regression task specified by an unknown parameter vector $\boldsymbol{\beta}_i \in \mathbb{R}^M$:

$$y_j^{(i)} = \boldsymbol{\beta}_i^{\mathsf{T}} \mathbf{x}_j^{(i)} + \varepsilon_j^{(i)}, \ i = 1, 2, \ 1 \leq j \leq N_i,$$

where $\varepsilon_j^{(i)} \in \mathbb{R}$ is a zero-mean random noise term with variance $\sigma^2$. For $i = 1, 2$ and $z \in \mathbb{C}\backslash\mathbb{R}^+$, define

$$\mathbf{X}_i = (\mathbf{x}_1^{(i)}, ..., \mathbf{x}_{N_i}^{(i)}) \in \mathbb{R}^{M \times N_i}, \quad \mathbf{y}_i = (y_1^{(i)}, ..., y_{N_i}^{(i)})^{\mathsf{T}} \in \mathbb{R}^{N_i},$$

$$\mathbf{Q}_i(z) = \left( \frac{1}{N_i} \mathbf{X}_i \mathbf{X}_i^{\mathsf{T}} - z\mathbf{I}_M \right)^{-1}, \quad \boldsymbol{\varepsilon}_i = (\varepsilon_1^{(i)}, ..., \varepsilon_{N_i}^{(i)})^{\mathsf{T}} \in \mathbb{R}^{N_i}.$$

We refer to the case where $D_1 \neq D_2$ as a ***covariate shift***, and the case where $\boldsymbol{\beta}_1 \neq \boldsymbol{\beta}_2$ as a ***model shift***.

**Teacher Model:** The teacher model is finetuned on $\left\{(\mathbf{x}_j^{(1)}, y_j^{(1)})\right\}_{j=1}^{N_1}$:

$$\boldsymbol{\beta}_{\mathsf{t}} = \arg\min_{\boldsymbol{\beta}} \left( \frac{1}{N_1} \|\mathbf{y}_1 - \mathbf{X}_1^{\mathsf{T}}\boldsymbol{\beta}\|^2 + \lambda_{\mathsf{t}}\|\boldsymbol{\beta}\|^2 \right) = \frac{1}{N_1} \mathbf{Q}_1(-\lambda_{\mathsf{t}})\mathbf{X}_1\mathbf{y}_1, \tag{2}$$

where $\lambda_{\mathsf{t}} > 0$ is the teacher regularization parameter. The risk of $\boldsymbol{\beta}_{\mathsf{t}}$ when $M \sim N_1$ in the high-dimensional setting has been studied extensively in the literature such as Dobriban & Wager (2018); Hastie et al. (2022).

**Student Model Trained with Cross-Domain KD:** We use the pre-trained teacher model together with covariates $\{\mathbf{x}_j^{(2)}\}_{j=1}^{N_2}$ to generate predictions:

$$\mathbf{y}_2^{\mathsf{t}} = (y_1^{\mathsf{t}}, ..., y_{N_2}^{\mathsf{t}})^{\mathsf{T}} = \left(\mathbf{x}_1^{(2)}, ..., \mathbf{x}_{N_2}^{(2)}\right)^{\mathsf{T}} \boldsymbol{\beta}_{\mathsf{t}}.$$

The student model is finetuned on the target domain data $\{(\mathbf{x}_j^{(2)}, y_j^{(2)})\}_{j=1}^{N_2}$ and the teacher's predictions $\{(\mathbf{x}_j^{(2)}, y_j^{\mathrm{t}})\}_{j=1}^{N_2}$, using the per-sample objective function defined in equation 1 with an imitation parameter $\xi$, as follows:

$$\boldsymbol{\beta}_{\mathsf{s}} = \arg\min_{\boldsymbol{\beta}} \mathcal{L}(\xi) = \arg\min_{\boldsymbol{\beta}} \xi\left(\frac{1}{N_2}\|\mathbf{y}_2^{\mathsf{t}} - \mathbf{X}_2^{\mathsf{T}}\boldsymbol{\beta}\|^2\right) + (1-\xi)\left(\frac{1}{N_2}\|\mathbf{y}_2 - \mathbf{X}_2^{\mathsf{T}}\boldsymbol{\beta}\|^2\right) + \lambda_{\mathsf{s}}\|\boldsymbol{\beta}\|^2$$
$$= (\mathbf{X}_2\mathbf{X}_2^{\mathsf{T}} + N_2\lambda_{\mathsf{s}}\mathbf{I}_M)^{-1}(\xi\mathbf{X}_2\mathbf{y}_2^{\mathsf{t}} + (1-\xi)\mathbf{X}_2\mathbf{y}_2),$$

$$(3)$$

where $\lambda_{\mathsf{s}}$ is the student regularization parameter. While it is common to restrict $\xi \in [0, 1]$ (Lopez-Paz et al., 2015), we do not impose this constraint, in line with Das & Sanghavi (2023); Pareek et al. (2024). From equation 3, the parameter $\xi$ is independent of $\mathbf{Q}_2$, making it possible to choose a negative $\xi$ that achieves better generalization performance. For the covariates $\mathbf{x}_j^{(i)}$ and the noise terms $\varepsilon_j^{(i)}, i = 1, 2, \ 1 \le j \le N_i$, we make the following assumptions, which are standard in the random matrix theory literature (see, e.g., Bai & Silverstein (2010)).

**Assumption 1.** *Suppose $\mathbf{X}_1, \mathbf{X}_2, \boldsymbol{\varepsilon}_1$, and $\boldsymbol{\varepsilon}_2$ are mutually independent. Moreover, we assume*

*(a) the covariates are generated according to*

$$\mathbf{X}_i = (\boldsymbol{\Sigma}_i)^{1/2}\mathbf{Z}_i, \ \ for \ i = 1, 2,$$

*where $\mathbf{Z}_i = (z_{jk}^{(i)})$ is an $M \times N_i$ random matrix with i.i.d. entries of zero mean and unit variance, and $\boldsymbol{\Sigma}_i$ is a positive semi-definite matrix. Furthermore, we assume for all $p \in \mathbb{N}$, there is a constant $C_p$ such that*

$$\max_{i=1,2} \mathbb{E}|z_{11}^{(i)}|^p \le C_p. \tag{4}$$

*(b) $M \sim N_1 \sim N_2$.*

*(c) $\boldsymbol{\varepsilon}_i \in \mathbb{R}^{N_i}$ is a random vector consisting of i.i.d. entries of zero mean, variance $\sigma^2$, and for all $p \in \mathbb{N}$, there is a constant $c_p$ such that*

$$\max_{i=1,2} \mathbb{E}|\varepsilon_1^{(i)}|^p \le c_p.$$

While we allow $z_{11}^{(1)}$ and $z_{11}^{(2)}$ to follow different distributions – a form of covariate shift – our theoretical results do not depend on their specific distributions, provided that the moment conditions in Assumption 1(a) are satisfied. The requirement that all moments of $z_{11}^{(i)}$ exist can be relaxed to the existence of $(8 + c)$-th moment for any positive constant $c$, with minor modifications to our proof and hence we do not pursue this generalization here. The following assumption on the structure of the covariance matrices is imposed to facilitate theoretical analysis and rule out degenerate cases.

**Assumption 2.** *Let $\tau$ be a small constant. Denote the eigenvalues of $\boldsymbol{\Sigma}_i$ by $\sigma_1^i \ge \sigma_2^i \cdots \ge \sigma_M^i \ge 0$.*

*(a) (Boundedness of $\boldsymbol{\Sigma}_i$). We assume that $\max_{i=1,2} \|\boldsymbol{\Sigma}_i\| = \sigma_1^i < \tau^{-1}$.*

*(b) (Anti-concentration at 0). For $i = 1, 2$, the empirical spectral distribution of $\boldsymbol{\Sigma}_i$ satisfies*

$$\frac{1}{M}\sum_{j=1}^{M} \delta(\sigma_j^i \le \tau) \le 1 - \tau.$$

Let $(\mathbf{x}, y)$ be an unseen sample of the target task, that is $y = \boldsymbol{\beta}_2^{\mathsf{T}}\mathbf{x} + \varepsilon$, where $\mathbf{x} \sim D_2$ and $\varepsilon$ follows the same distribution with $\varepsilon_1^{(2)}$. Under the mean squared loss, the generalization ability is quantified by the risk of the estimator $\boldsymbol{\beta}_{\mathsf{s}}$ :

$$\mathbf{R}(\boldsymbol{\beta}_{\mathsf{s}}) = \mathbb{E}_{\mathbf{x},y}|y - \boldsymbol{\beta}_{\mathsf{s}}^{\mathsf{T}}\mathbf{x}|^2 = \mathbb{E}_{\mathbf{x},y}|(\boldsymbol{\beta}_2 - \boldsymbol{\beta}_{\mathsf{s}})^{\mathsf{T}}\mathbf{x} + \varepsilon|^2 = \|\boldsymbol{\Sigma}_2^{1/2}(\boldsymbol{\beta}_2 - \boldsymbol{\beta}_{\mathsf{s}})\|^2 + \sigma^2,$$

where $\mathbb{E}_{\mathbf{x},y}$ denotes the expectation taken with respect to (w.r.t.) the pair $(\mathbf{x}, y)$. The excess test risk is defined as follows:

$$\mathbf{ER}(\boldsymbol{\beta}_{\mathsf{s}}) = \mathbf{R}(\boldsymbol{\beta}_{\mathsf{s}}) - \sigma^2 = \|\boldsymbol{\Sigma}_2^{1/2}(\boldsymbol{\beta}_2 - \boldsymbol{\beta}_{\mathsf{s}})\|^2. \tag{5}$$

When $\xi = 0$, $\boldsymbol{\beta}_{\mathsf{s}}$ reduces to the ridge regression estimator for the student only model, and we denote the corresponding excess risk by $\mathbf{ER}_0$. Note that $\mathbf{ER}(\boldsymbol{\beta}_{\mathsf{s}})$ can be decomposed into bias and variance as $\mathbf{ER}(\boldsymbol{\beta}_{\mathsf{s}}) = \mathbf{Bias} + \mathbf{Var}$, where

$$\mathbf{Bias} = \|\boldsymbol{\Sigma}_2^{1/2}(\boldsymbol{\beta}_2 - \mathbb{E}_{\mathbf{x},y}\boldsymbol{\beta}_{\mathsf{s}})\|^2,$$

$$\mathbf{Var} = \|\boldsymbol{\Sigma}_2^{1/2}(\boldsymbol{\beta}_{\mathsf{s}} - \mathbb{E}_{\mathbf{x},y}\boldsymbol{\beta}_{\mathsf{s}})\|^2.$$

One may easily check that $\mathbf{ER} = O(1)$ almost surely. In the remainder of this paper, we derive asymptotic expressions for the bias and variance terms to analyze the generalization performance of the student model using tools from random matrix theory.

## 2.2 RANDOM MATRIX THEORY

Before proceeding to the theoretical analysis, we introduce several key quantities from random matrix theory that will appear in our main results. For any distribution $G$ supported on $\mathbb{R}^+ = [0, \infty)$, its Stieltjes transform is defined as

$$m_G(z) = \int \frac{1}{x - z} \mathrm{d}G(x), \ z \notin \mathrm{supp}(G).$$

Next, we define the asymptotic eigenvalue density of random matrices via its Stieltjes transform. This lemma is well-known in the random matrix theory literature (e.g., Bai & Silverstein (2010)).

**Lemma 2.** *Let* $\mathbf{X} = \boldsymbol{\Sigma}^{1/2}\mathbf{Z}$ *be a random matrix, where* $\mathbf{Z} = (z_{jk}) \in \mathbb{R}^{M \times N}$, $M \sim N$ *satisfies Assumption 1(a), and* $\boldsymbol{\Sigma}$ *satisfies Assumption 2. For each* $z \in \mathbb{C}\backslash\mathbb{R}^+$*, there exists a unique* $m \equiv m_M(z) \in \mathbb{C}$ *satisfying the equation*

$$z = -\frac{1}{m} + \frac{1}{N}\mathrm{Tr}\frac{\boldsymbol{\Sigma}}{1 + m\boldsymbol{\Sigma}} = -\frac{1}{m} - \frac{z}{N}\mathrm{Tr}\boldsymbol{\Sigma}\boldsymbol{\Pi}, \ \ with \ \Im z\Im m(z) \geq 0, \tag{6}$$

*where* $\boldsymbol{\Pi}(z) = -(z + zm\boldsymbol{\Sigma})^{-1}$.

# 3 THEORETICAL ANALYSIS

In this section, we analyze the excess risk $\mathbf{ER}(\boldsymbol{\beta}_{\mathsf{s}})$ defined in 5 under three distinct settings. In Section 3.1, we consider the case where $\boldsymbol{\beta}_1$ and $\boldsymbol{\beta}_2$ are deterministic, with their difference being arbitrary. In Section 3.2, we study the scenario in which $\boldsymbol{\beta}_1 = \boldsymbol{\beta}_2$ and the common parameter vector is drawn from a prior distribution. Finally, in Section 3.3, we analyze ridgeless regression under the regime where $M < N_1, N_2$ and the covariance matrices $\boldsymbol{\Sigma}_1, \boldsymbol{\Sigma}_2$ are invertible.

Before presenting the main results, we first introduce some necessary notation. For $M, N_i, \boldsymbol{\Sigma}_i$ and $z < 0$, the Stieltjes transform determined by Lemma 2 is denoted by $m_i(z)$.

Let $\boldsymbol{\Pi}_i(z) = -(z + zm_i(z)\boldsymbol{\Sigma}_i)^{-1}$, $i = 1, 2$. We write $\mathbf{Q}_1 = \mathbf{Q}_1(-\lambda_{\mathsf{t}})$, $\mathbf{Q}_2 = \mathbf{Q}_2(-\lambda_{\mathsf{s}})$, $\boldsymbol{\Pi}_1 = \boldsymbol{\Pi}_1(-\lambda_{\mathsf{t}})$ and $\boldsymbol{\Pi}_2 = \boldsymbol{\Pi}_2(-\lambda_{\mathsf{s}})$ for notational simplicity. For any deterministic matrix $\mathbf{A}$ with bounded spectral norm, we define

$$\mathcal{S}_i(\mathbf{A}) = \mathbf{A} + \frac{\frac{1}{N_i}\mathrm{Tr}\boldsymbol{\Sigma}_i\boldsymbol{\Pi}_i\mathbf{A}\boldsymbol{\Pi}_i}{(1 + \frac{1}{N_i}\mathrm{Tr}\boldsymbol{\Sigma}_i\boldsymbol{\Pi}_i)^2 - \frac{1}{N_i}\mathrm{Tr}(\boldsymbol{\Sigma}_i\boldsymbol{\Pi}_i)^2}\boldsymbol{\Sigma}_i, \ i = 1, 2.$$

Moreover, when $\mathbf{A} = \mathbf{I}_M$, we denote

$$\boldsymbol{\Pi}_1' = \frac{\mathrm{d}}{\mathrm{d}z}\boldsymbol{\Pi}_1(z)\big|_{z=-\lambda_{\mathsf{t}}} = \boldsymbol{\Pi}_1\mathcal{S}_1(\mathbf{I}_M)\boldsymbol{\Pi}_1, \quad \boldsymbol{\Pi}_2' = \frac{\mathrm{d}}{\mathrm{d}z}\boldsymbol{\Pi}_2(z)\big|_{z=-\lambda_{\mathsf{s}}} = \boldsymbol{\Pi}_2\mathcal{S}_2(\mathbf{I}_M)\boldsymbol{\Pi}_2,$$

The other quantities are summarized in Table 1.

## 3.1 DETERMINISTIC REGRESSION PARAMETERS

We now state our first main result.

Table 1: Some notations used in the theoretical results

| | |
|---|---|
| $\mathbf{E}_1 = \mathbf{\Pi}_1 \mathcal{S}_1(\mathbf{\Sigma}_2)\mathbf{\Pi}_1,$ | $\mathbf{E}_2 = \mathbf{\Pi}_1 \mathcal{S}_1(\mathbf{\Pi}_2 \mathcal{S}_2(\mathbf{\Sigma}_2)\mathbf{\Pi}_2)\mathbf{\Pi}_1$ |
| $\mathbf{E}_3 = \mathbf{\Pi}_1 \mathcal{S}_1(\mathbf{\Sigma}_2 \mathbf{\Pi}_2)\mathbf{\Pi}_1,$ | $\mathbf{E}_4 = \mathbf{\Pi}_2 \mathcal{S}_2(\mathbf{\Sigma}_2)\mathbf{\Pi}_2,$ $\quad \mathbf{E}_5 = \mathbf{\Sigma}_2 \mathbf{\Pi}_2$ |

**Theorem 1.** *Let* $\boldsymbol{\gamma} = \boldsymbol{\beta}_1 - \boldsymbol{\beta}_2$. *For the deterministic vectors* $\|\boldsymbol{\beta}_1\|$ *and* $\|\boldsymbol{\beta}_2\|$, *assume that* $\|\boldsymbol{\beta}_1\|, \|\boldsymbol{\beta}_2\| \leq c$ *for some constant* $c$. *Under Assumptions 1-2, the following results hold:*

$$\mathbf{Bias} = \widehat{\mathbf{Bias}} + o_{a.s.}(1), \quad \mathbf{Var} = \widehat{\mathbf{Var}} + o_{a.s.}(1),$$

*where*

$$\begin{aligned}
\widehat{\mathbf{Bias}} = {} & \xi^2 \boldsymbol{\beta}_1^{\mathsf{T}} \big[\lambda_{\mathsf{t}}^2 \mathbf{E}_1 + \lambda_{\mathsf{s}}^2 \lambda_{\mathsf{t}}^2 \mathbf{E}_2 - 2\lambda_{\mathsf{t}}^2 \lambda_{\mathsf{s}} \mathbf{E}_3\big]\boldsymbol{\beta}_1 + 2\xi \boldsymbol{\beta}_2^{\mathsf{T}} \big[\lambda_{\mathsf{s}}^2 \mathbf{E}_4 - \lambda_{\mathsf{s}} \mathbf{E}_5\big]\boldsymbol{\gamma} \\
& + \lambda_{\mathsf{s}}^2 \boldsymbol{\beta}_2^{\mathsf{T}} \mathbf{E}_4 \boldsymbol{\beta}_2 + 2\xi \boldsymbol{\beta}_1^{\mathsf{T}} \big[\lambda_{\mathsf{t}} \lambda_{\mathsf{s}} \mathbf{\Pi}_1 \mathbf{E}_5 - \lambda_{\mathsf{t}} \lambda_{\mathsf{s}}^2 \mathbf{\Pi}_1 \mathbf{E}_4\big]\boldsymbol{\beta}_2 + \xi^2 \boldsymbol{\gamma}^{\mathsf{T}}\big[-2\lambda_{\mathsf{s}} \mathbf{E}_5 + \lambda_{\mathsf{s}}^2 \mathbf{E}_4\big]\boldsymbol{\gamma} \quad (7) \\
& + 2\xi^2 \boldsymbol{\gamma}^{\mathsf{T}}\big[\lambda_{\mathsf{s}} \lambda_{\mathsf{t}} \mathbf{E}_5 \mathbf{\Pi}_1 - \lambda_{\mathsf{t}} \lambda_{\mathsf{s}}^2 \mathbf{E}_4 \mathbf{\Pi}_1 - \lambda_{\mathsf{t}} \mathbf{\Sigma}_2 \mathbf{\Pi}_1 + \lambda_{\mathsf{t}} \lambda_{\mathsf{s}} \mathbf{E}_5 \mathbf{\Pi}_1\big]\boldsymbol{\beta}_1,
\end{aligned}$$

*and*

$$\widehat{\mathbf{Var}} = \frac{\xi^2 \sigma^2}{N_1} \mathrm{Tr}\big[\big(\mathbf{\Sigma}_2 - 2\lambda_{\mathsf{s}} \mathbf{E}_5 + \lambda_{\mathsf{s}}^2 \mathbf{E}_4\big)\big(\mathbf{\Pi}_1 - \lambda_{\mathsf{t}} \mathbf{\Pi}_1'\big)\big] + \frac{(1-\xi)^2 \sigma^2}{N_2} \mathrm{Tr}\big[\mathbf{E}_5 - \lambda_{\mathsf{s}} \mathbf{\Sigma}_2 \mathbf{\Pi}_2'\big]. \quad (8)$$

This theorem characterizes the dependence of $\mathbf{ER}(\boldsymbol{\beta}_{\mathsf{s}})$ on the geometry of $\mathbf{\Sigma}_1, \boldsymbol{\beta}_1, \mathbf{\Sigma}_2$, and $\boldsymbol{\beta}_2$. We provide an illustrative example here. Suppose that $\mathbf{\Sigma}_i$ admits the spectral decomposition $\mathbf{\Sigma}_i = \mathbf{U}_i \mathbf{\Lambda}_i \mathbf{U}_i^{\mathsf{T}}$, for $i = 1, 2$. Consider the term $\boldsymbol{\beta}_1^{\mathsf{T}} \mathbf{\Pi}_1 \mathbf{E}_5 \boldsymbol{\beta}_2$, which can be expressed as

$$\boldsymbol{\beta}_1^{\mathsf{T}}(\lambda_{\mathsf{t}} + \lambda_{\mathsf{t}} m_1 \mathbf{\Sigma}_1)^{-1}(\lambda_{\mathsf{s}} + \lambda_{\mathsf{s}} m_2 \mathbf{\Sigma}_2)^{-1} \mathbf{\Sigma}_2 \boldsymbol{\beta}_2 = (\lambda_{\mathsf{s}} \lambda_{\mathsf{t}})^{-1} \tilde{\boldsymbol{\beta}}_1^{\mathsf{T}}(1 + m_1 \mathbf{\Lambda}_1)^{-1} \mathbf{U}_1^{\mathsf{T}} \mathbf{U}_2 \tilde{\mathbf{\Lambda}}_2 \tilde{\boldsymbol{\beta}}_2, \quad (9)$$

where $\tilde{\mathbf{\Lambda}}_2$ is a diagonal matrix with entries $\tilde{\mathbf{\Lambda}}_{2,jj} = \frac{\mathbf{\Lambda}_{2,jj}}{1 + m_2 \mathbf{\Lambda}_{2,jj}}$. The vector $\tilde{\boldsymbol{\beta}}_i = \mathbf{U}_i \boldsymbol{\beta}_i$ captures the alignment between $\boldsymbol{\beta}_i$ and the eigenvectors of $\mathbf{\Sigma}_i$. The right-hand side of equation 9 explicitly reveals how the term $\boldsymbol{\beta}_1^{\mathsf{T}} \mathbf{\Pi}_1 \mathbf{E}_5 \boldsymbol{\beta}_2$ depends on $\tilde{\boldsymbol{\beta}}_i$, the eigenvalues of $\mathbf{\Sigma}_1$ and $\mathbf{\Sigma}_2$, and the eigenvector overlap $\mathbf{U}_1^{\mathsf{T}} \mathbf{U}_2$ between the two covariance matrices. In the special case where each $\boldsymbol{\beta}_i$ is aligned with an eigenvector of $\mathbf{\Sigma}_i$ – for simplicity, suppose it corresponds to the first eigenvector – the expression equation 9 further simplifies to $\boldsymbol{\beta}_1^{\mathsf{T}} \boldsymbol{\beta}_2 (\lambda_{\mathsf{s}} \lambda_{\mathsf{t}})^{-1} \frac{\mathbf{\Lambda}_{2,jj}}{(1 + m_1 \mathbf{\Lambda}_{1,11})(1 + m_2 \mathbf{\Lambda}_{2,11})}$, which depends on the eigenvalues of $\mathbf{\Sigma}_i$ and the inner product $\boldsymbol{\beta}_1^{\mathsf{T}} \boldsymbol{\beta}_2$. This observation extends the results of Hastie et al. (2022), which considers high-dimensional least squares regression within a single domain (corresponding to $\xi = 0$ in equation 3).

## 3.2 RANDOM REGRESSION PARAMETERS

In this section, we assume that the vector $\boldsymbol{\beta}_1 = \boldsymbol{\beta}_2 = \boldsymbol{\beta}$ is random, and consider the excess risk under two population covariance matrices, $\mathbf{\Sigma}_1$ and $\mathbf{\Sigma}_2$, which may be equal or distinct. Before presenting the main result, we introduce the following assumption, commonly used in the literature (Dobriban & Wager, 2018; Moniri & Hassani, 2025).

**Assumption 3.** *The regression parameter vector* $\boldsymbol{\beta} = (\beta_1, ..., \beta_M)^{\mathsf{T}} \in \mathbb{R}^M$ *is random, with each entry i.i.d., and* $\beta_1$ *satisfies*

$$\mathbb{E}\beta_1 = 0, \ \mathbb{E}\beta_1^2 = \frac{\widetilde{\sigma}^2}{M}, \ \text{and} \ \mathbb{E}|\sqrt{M}\beta_1|^p \leq C_p,$$

*for any* $p \in \mathbb{N}$, *where* $C_p$ *is a constant depending only on* $p$.

**Theorem 2.** *Suppose Assumptions 1-3 hold. Then the following asymptotic expressions hold:*

$$\begin{aligned}
\widehat{\mathbf{Bias}} = {} & \frac{\tilde{\sigma}^2}{M}\Big[\xi^2 \lambda_{\mathsf{t}}^2 \mathrm{Tr}\mathbf{\Sigma}_2 \mathbf{\Pi}_1' + 2\xi \lambda_{\mathsf{t}} \lambda_{\mathsf{s}} \mathrm{Tr}\mathbf{\Pi}_1 \mathbf{\Pi}_2 \mathbf{\Sigma}_2 + \lambda_{\mathsf{s}}^2 \mathrm{Tr}\mathbf{\Sigma}_2 \mathbf{\Pi}_2' \\
& - 2\xi^2 \lambda_{\mathsf{t}}^2 \lambda_{\mathsf{s}} \mathrm{Tr}\mathbf{\Sigma}_2 \mathbf{\Pi}_2 \mathbf{\Pi}_1' + \xi \lambda_{\mathsf{t}} \lambda_{\mathsf{s}}^2 \mathrm{Tr}\big[\mathbf{E}_4(-2\mathbf{\Pi}_1 + \xi \lambda_{\mathsf{t}} \mathbf{\Pi}_1')\big]\Big] = \mathbf{Bias} + o_{a.s.}(1),
\end{aligned}$$

*and* $\widehat{\mathbf{Var}} = \mathbf{Var} + o_{a.s.}$, *which coincides with the expression in Theorem 1.*

This theorem extends the result of Moniri & Hassani (2025), which considers the case of no covariate shift, with inputs drawn i.i.d. from $\mathcal{N}(0, \mathbf{I}_M)$ in the context of W2S generalization (i.e., when $\xi = 1$). Our framework generalizes this analysis by allowing $\xi \in \mathbb{R}$, thereby providing a more comprehensive understanding of the trade-off between learning from the teacher and from the observed labels.

Let $\underline{m}_1(z), \underline{m}_2(z)$ be the Stieltjes transforms of the standard Marchenko-Pastur law with parameters $M/N_1, M/N_2$, respectively; their explicit forms are given in equation 28. The following corollary follows immediately from Theorem 2 and the fact that $\mathbf{\Pi}_1 = \underline{m}_1 \mathbf{I}_M, \mathbf{\Pi}_2 = \underline{m}_2 \mathbf{I}_M$ (see, e.g., Alex et al. (2014)).

**Corollary 1.** *Suppose $\mathbf{\Sigma}_1 = \mathbf{\Sigma}_2 = \mathbf{I}_M$. Write $\underline{m}_1 = \underline{m}_1(-\lambda_\mathsf{t}), \underline{m}_2 = \underline{m}_2(-\lambda_\mathsf{s})$. Under Assumption 1 and Assumption 3, we have the following expressions:*

$$\mathbf{Bias} = \tilde{\sigma}^2[\xi^2\lambda_\mathsf{t}^2\underline{m}_1' + 2\xi\lambda_\mathsf{t}\lambda_\mathsf{s}\underline{m}_1\underline{m}_2 + \lambda_\mathsf{s}^2\underline{m}_2' - 2\xi^2\lambda_\mathsf{t}^2\lambda_\mathsf{s}\underline{m}_2\underline{m}_1'$$
$$- 2\xi\lambda_\mathsf{t}\lambda_\mathsf{s}^2\underline{m}_2'\underline{m}_1 + \xi^2\lambda_\mathsf{t}^2\lambda_\mathsf{s}^2\underline{m}_1'\underline{m}_2'] + o_{a.s.}(1),$$

*and*

$$\mathbf{Var} = \xi^2\sigma^2\frac{M}{N_1}\big[\underline{m}_1 - 2\lambda_\mathsf{s}\underline{m}_1\underline{m}_2 + \lambda_\mathsf{s}^2\underline{m}_1\underline{m}_2' - \lambda_\mathsf{t}\underline{m}_1' + 2\lambda_\mathsf{t}\lambda_\mathsf{s}\underline{m}_2\underline{m}_1' - \lambda_\mathsf{t}\lambda_\mathsf{s}^2\underline{m}_1'\underline{m}_2'\big]$$

$$+ (1 - \xi)^2\sigma^2\frac{M}{N_2}\big[\underline{m}_2 - \lambda_\mathsf{s}\underline{m}_2'\big] + o_{a.s.}(1),$$

*where $\underline{m}_1', \underline{m}_2'$ denote the derivatives evaluated at $z = -\lambda_\mathsf{t}$ and $z = -\lambda_\mathsf{s}$, respectively:*

$$\underline{m}_1' = \frac{\mathrm{d}}{\mathrm{d}z}\underline{m}_1(z)\big|_{z=-\lambda_\mathsf{t}}, \underline{m}_2' = \frac{\mathrm{d}}{\mathrm{d}z}\underline{m}_2(z)\big|_{z=-\lambda_\mathsf{s}}.$$

As previously noted, we do not restrict $\xi$ to the interval $[0, 1]$. It has been shown in Das & Sanghavi (2023) that the optimal value of $\xi$ may exceed 1. In Corollary 2 below, we present a toy example demonstrating that even when the input data across domains are i.i.d. and in the absence of model shift – i.e., with no domain shift – the limiting optimal value of $\xi$ can be negative.

**Corollary 2.** *Suppose the conditions in Corollary 1 hold. The limiting optimal value of $\xi < 0$ if*

$$\lambda_\mathsf{s}\lambda_\mathsf{t}\underline{m}_1\mathrm{SNR} - \frac{M}{N_2} > 0, \tag{10}$$

*where $\mathrm{SNR} = \frac{\tilde{\sigma}^2}{\sigma^2} = \frac{\|\boldsymbol{\beta}\|^2}{\sigma^2} + o_{a.s.}(1)$.*

**Remark 1.** *We call the case $\xi < 0$ **anti-learning against the teacher's supervision**, in contrast to $\xi > 1$, which Das & Sanghavi (2023) termed anti-learning the observed (possibly noisy) labels. This corollary provides insight into the selection of $\xi$: the sign of the limiting optimal value of $\xi$ depends not only on parameters ($\lambda_\mathsf{t}, \lambda_\mathsf{s}$) but also on data-related factors (SNR, data dimension, and sample sizes of both domains).*

## 3.3 RIDGELESS REGRESSION

In this section, we consider the minimum $\ell_2$ norm least squares (ridgeless) regression estimator. Specifically, the teacher model is defined by

$$\boldsymbol{\beta}_\mathsf{t} = (\mathbf{X}_1\mathbf{X}_1^\mathsf{T})^+\mathbf{X}_1\mathbf{y}_1,$$

where $(\mathbf{X}_1\mathbf{X}_1^\mathsf{T})^+$ denotes the Moore-Penrose inverse of $\mathbf{X}_1\mathbf{X}_1^\mathsf{T}$. Similarly, the ridgeless estimator of $\boldsymbol{\beta}_\mathsf{s}$ takes the form

$$\boldsymbol{\beta}_\mathsf{s}(\mathbf{X}_2\mathbf{X}_2^\mathsf{T})^+[\xi\mathbf{X}_2\mathbf{X}_2^\mathsf{T}\boldsymbol{\beta}_\mathsf{t} + (1 - \xi)\mathbf{X}_2\mathbf{y}_2].$$

**Theorem 3.** *(1) Suppose $\boldsymbol{\beta}_1, \boldsymbol{\beta}_2$ are deterministic, and Assumptions 1-2 hold. We further assume*

$$\left|\frac{M}{N_i} - 1\right| \geq \tau, \quad \tau \leq \sigma_{\min}(\mathbf{\Sigma}_i) \leq \cdots \leq \sigma_{\max}(\mathbf{\Sigma}_i) \leq \tau^{-1}, \text{ for } i = 1, 2.$$

*Define $f(\lambda) = \widehat{\mathbf{Bias}}$ and $g(\lambda) = \widehat{\mathbf{Var}}$, with $\lambda = \lambda_\mathsf{s} = \lambda_\mathsf{t}$, where the expressions for $\widehat{\mathbf{Bias}}$ and $\widehat{\mathbf{Var}}$ are provided in equation 7 and equation 8, respectively. We have*

$$\mathbf{Bias} = f(0^+) + o_{a.s.}(1), \quad \mathbf{Var} = g(0^+) + o_{a.s.}(1). \tag{11}$$

*(2) Suppose $\boldsymbol{\beta} = \boldsymbol{\beta}_1 = \boldsymbol{\beta}_2$ are random and Assumptions 1-3 hold. Then, the estimated expressions in equation 11 still holds with $f(\lambda) = \widehat{\mathbf{Bias}}$ replaced by the $\widehat{\mathbf{Bias}}$ defined in Theorem 2.*

If a matrix $\mathbf{A}$ is nonsingular, $\mathbf{A}^+ = \mathbf{A}^{-1}$. The following corollary gives the characterization of $\mathbf{ER}(\boldsymbol{\beta}_{\mathsf{s}})$ in the under-parameterized setting.

**Corollary 3.** *Suppose the conditions in Theorem 3 hold and $\frac{M}{N_1}, \frac{M}{N_2} \leq 1 - \tau$. The estimator for student model obtained by equation 1 is the averaging estimator:*

$$\boldsymbol{\beta}_{\mathsf{s}} = \xi \boldsymbol{\beta}_1^{\mathsf{OLS}} + (1-\xi)\boldsymbol{\beta}_2^{\mathsf{OLS}}, \text{ where } \boldsymbol{\beta}_i^{\mathsf{OLS}} = (\mathbf{X}_i \mathbf{X}_i^{\mathsf{T}})^{-1} \mathbf{X}_i \mathbf{y}_i, \; i = 1, 2. \tag{12}$$

*Adopting the notation $\boldsymbol{\gamma} = \boldsymbol{\beta}_1 - \boldsymbol{\beta}_2$ in Theorem 1, we have*

$$\widehat{\mathrm{Bias}} = \xi^2 \boldsymbol{\gamma}^{\mathsf{T}} \boldsymbol{\Sigma}_2 \boldsymbol{\gamma}, \quad \widehat{\mathrm{Var}} = (1-\xi)^2 \sigma^2 \frac{M}{N_2 - M} + \xi^2 \sigma^2 \frac{1}{N_1 - M} \mathrm{Tr}\boldsymbol{\Sigma}_2 \boldsymbol{\Sigma}_1^{-1}.$$

Based on the conclusions of Theorems 1-3, the high-dimensional asymptotic excess risk, regarded as a function of $\xi$, is a quadratic function. Given that the excess risk is non-negative, the quadratic function opens upwards. This observation is consistent with Pareek et al. (2024), where self-distillation is considered. Given a $\xi \in \mathbb{R}$, the gain of cross-domain KD is characterized by the reduction in excess risk, $\mathbf{ER}_0 - \mathbf{ER}(\boldsymbol{\beta}_{\mathsf{s}})$.

**Proposition 1.** *Under the conditions of Theorem 1 and Assumption A.1 for the deterministic case in Appendix B.7, there exists a value of $\xi \in \mathbb{R}$ such that*

$$\min_{\xi \in \mathbb{R}} \left( \mathbf{ER}(\boldsymbol{\beta}_{\mathsf{s}}) - \mathbf{ER}_0 \right) < 0. \; a.s. \tag{13}$$

*Moreover, under the conditions of Theorem 1 and Assumption A.2 for the random case in Appendix B.7, the inequality 13 also holds.*

**Remark 2.** *This proposition shows that, even in the presence of a significant domain discrepancy, it is possible to find a value of $\xi \in \mathbb{R}$ such that the student model outperforms the student-only baseline (i.e., training on the observed labels only). We provide further details in Appendix B.7, where we provide closed-form expressions for the optimal $\xi^*$ under several common settings and demonstrate that covariate shift can, in some cases, be beneficial for KD .*

## 3.4 NUMERICAL SIMULATIONS

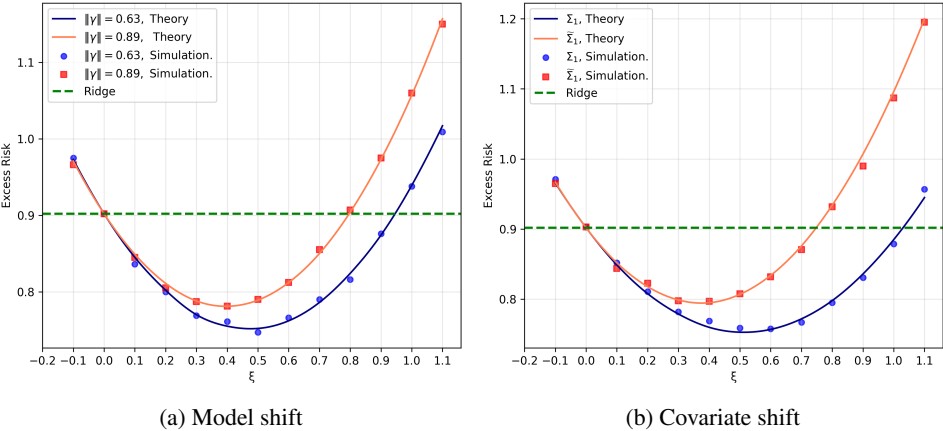

(a) Model shift

(b) Covariate shift

Figure 1: Student's excess risk in the presence of domain shift. Solid lines represent theoretical values, while scattered points denote simulation results (averaged over 100 trials). The dashed green line indicates the theoretical performance for student-only baseline, corresponding to ridge regression trained solely on the target domain data. (a) Settings: $(\lambda_{\mathsf{t}}, \lambda_{\mathsf{s}}) = (0.1, 0.5), (M, N_1, N_2) = (400, 600, 200), \boldsymbol{\Sigma}_1 = \boldsymbol{\Sigma}_2 = \mathbf{I}_M$. The vectors $\boldsymbol{\beta}_2 = (1, ..., 1)^{\mathsf{T}}/\sqrt{M}, \sigma^2 = 1$. We label the case $\|\boldsymbol{\gamma}\| = 0.63$ as $\boldsymbol{\gamma} = -(2, \ldots, 2, 0, \ldots, 0)^{\mathsf{T}}/\sqrt{M}$ with the first $M/10$ entries equal to $-2/\sqrt{M}$, and the case $\|\boldsymbol{\gamma}\| = 0.89$ with the first $M/5$ entries equal to $-2/\sqrt{M}$. (b) Settings: $(\lambda_{\mathsf{t}}, \lambda_{\mathsf{s}}) = (0.1, 0.5), \boldsymbol{\beta}_1 = \boldsymbol{\beta}_2 \sim \mathcal{N}(0, M^{-1}\mathbf{I}_M), (M, N_1, N_2) = (600, 200, 300), \boldsymbol{\Sigma}_1 = 4\mathbf{I}_M, \widetilde{\boldsymbol{\Sigma}}_1 = \mathrm{diag}(d_1, \ldots, d_M)$ with $d_i = 0.64\delta(i \leq M/2) + 0.25\delta(M/2 < i \leq M), \sigma^2 = 1$.

We plot the excess risk of the student model: (a) under model shift with identical covariate distributions, and (b) under covariate shift with identical parameter vectors, in Figure 1. All theoretical

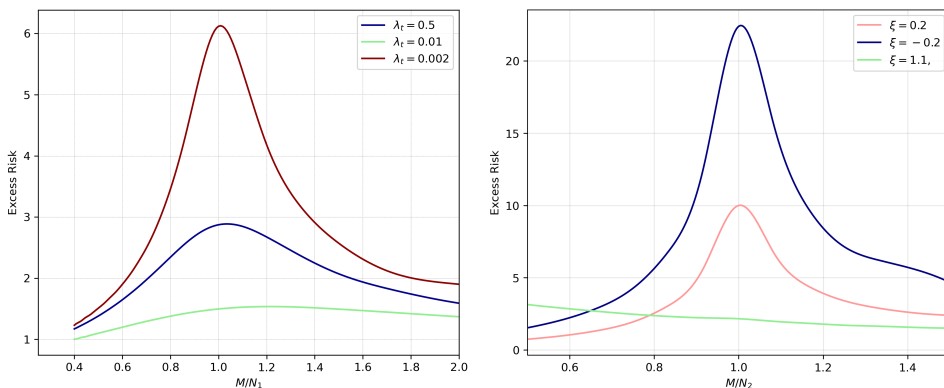

(a) Excess risk as a function of $\frac{M}{N_1}$ for varying $\lambda_{\mathsf{t}}$  (b) Excess risk as a function of $\frac{M}{N_2}$ for varying $\xi$

Figure 2: Non-monotone student excess risk curves. We set $\mathbf{\Sigma}_2 = \mathbf{I}_M$, $\mathbf{\Sigma}_1 = \mathrm{diag}(d_1, \cdots, d_M)$ where $d_i = 0.64\delta(i \leq \frac{M}{2}) + 0.25\delta(\frac{M}{2} < i \leq M)$. (a) Results are shown for fixed $M = 600$ and $\lambda_{\mathsf{s}} = 0.05$ with different $N_1$. (b) Results are shown for fixed $N_2$ and $(\lambda_{\mathsf{t}}, \lambda_{\mathsf{s}}) = (0.05, 0.001)$, with varying $M$.

values of the Stieltjes transform presented in this paper are obtained by solving equation 6. Due to space limitations, the numerical validation of Corollary 3 is provided in Appendix C.2. Simulation results, averaged over 100 independent trials, show good agreement with the theoretical predictions.

Furthermore, we present numerical simulations of $\mathbf{ER}(\boldsymbol{\beta}_{\mathsf{s}})$ as a function of $\lambda_{\mathsf{s}}$ and $\lambda_{\mathsf{t}}$ for various values of $\xi$; these results are included in Appendix C.3.

## 4 DOUBLE DESCENT OF THE EXCESS RISK

In this section, fixing $\xi, \lambda_{\mathsf{t}}$ and $\lambda_{\mathsf{s}}$, we examine the excess risk as a function of the dimension $M$ and the sample sizes $N_1$ and $N_2$. We find that the student model exhibits the double descent phenomenon, characterized by a non-monotonic behavior of the excess risk as a function of the ratio of dimension-to-sample-size. This phenomenon is consistent with findings in various linear regression settings (Hastie et al., 2022; Nakkiran et al., 2021; Belkin et al., 2020; Moniri & Hassani, 2024), and has been previously observed by Moniri & Hassani (2025) in the special case of pure teacher supervision without domain shift, where the risk was studied as a function of $\frac{M}{N_1}$.

Using our theoretical predictions from Theorem 2, we plot the excess risk of the student model, $\mathbf{ER} = \mathbf{ER}(\frac{M}{N_1})$, as a function of $\frac{M}{N_1}$ in Figure 2(a). The double descent phenomenon is evident for all three values of $\lambda_{\mathsf{t}}$. As $\lambda_{\mathsf{t}}$ decreases, the peak of the risk curve shifts towards $\frac{M}{N_1} = 1$. In Figure 2(b), we plot $\mathbf{ER} = \mathbf{ER}(\frac{M}{N_2})$ against $\frac{M}{N_2}$, while allowing $\frac{M}{N_1}$ to vary simultaneously. We consider different values of $\xi$ and observe that the double descent phenomenon is most pronounced in the regime of *anti-learning against the teacher's supervision* ($\xi < 0$). In contrast, when $\xi = 1.1$, no double descent occurs within the ratio range $[0.5, 1.5]$.

## 5 EXTENSION

### 5.1 NONLINEAR CASE

Our theoretical results are initially established for linear models; however, we anticipate that they can be extended to more general settings. To explore this extension, we conduct numerical simulations specifically for nonlinear models here. We assume the source data $\{(\mathbf{x}_j^{(1)}, y_j^{(1)})\}_{j=1}^{N_1}$ are generated i.i.d. according to $y_j^{(1)} = f(\mathbf{x}_j^{(1)}) + \varepsilon_j^{(1)}$, for $1 \leq j \leq N_1$. The target data $\{(\mathbf{x}_j^{(2)}, y_j^{(2)})\}_{j=1}^{N_2}$ are generated according to $y_j^{(2)} = \tilde{f}(\mathbf{x}_j^{(2)}) + \varepsilon_j^{(2)}$, for $1 \leq j \leq N_2$. Suppose $\mathbf{x}_j^{(1)} \sim D_1, 1 \leq j \leq N_1$

and $\mathbf{x}_j^{(2)} \sim D_2, 1 \le j \le N_2$. We refer to the case $D_1 \ne D_2$ as a covariate shift, and the case where $f \ne \tilde{f}$ as a model shift.

We consider learning the unknown function using a fully connected two-layer neural network with $n$ hidden neurons: $f_{\mathrm{NN}}(\mathbf{x}) = \mathbf{a}^\mathsf{T}\sigma(\mathbf{W}\mathbf{x})$, where $\mathbf{W} \in \mathbb{R}^{n \times M}$ is the weight matrix, and $\sigma(\cdot)$ is an activation function applied entrywise. When the random weight matrix $\mathbf{W}$ is fixed and only the second-layer weight $\mathbf{a}$ is optimized, the model reduces to a kernel regression model, where the kernel defined by $\mathbf{x} \to \sigma(\mathbf{W}\mathbf{x})$ is referred to as the conjugate kernel (Neal, 2012). The teacher model is given by $f_{\mathrm{NN}}^{\mathsf{t}}(\mathbf{x}) = \mathbf{a}_{\mathsf{t}}^\mathsf{T}\sigma(\widetilde{\mathbf{W}}_1\mathbf{x})$, with

$$\mathbf{a}_{\mathsf{t}} = \arg\min_{\mathbf{a}}\left\{\frac{1}{N_1}\|\mathbf{y}_1 - \sigma(\mathbf{X}_1^\mathsf{T}\widetilde{\mathbf{W}}_1^\mathsf{T})\mathbf{a}\|^2 + \lambda_{\mathsf{t}}\|\mathbf{a}\|^2\right\}.$$

We use $f_{\mathrm{NN}}^{\mathsf{t}}(\mathbf{x}_j^{(2)})$ together with the covariates $\{\mathbf{x}_j^{(2)}\}_{j=1}^{N_2}$ to generate predictions $\mathbf{y}_2^{\mathsf{t}}$. Then the student model is finetuned on the target domain data and $\mathbf{y}_2^{\mathsf{t}}$. The student model takes the form $f_{\mathrm{NN}}^{\mathsf{s}}(\mathbf{x}) = \mathbf{a}_{\mathsf{s}}^\mathsf{T}\sigma(\mathbf{W}_1\mathbf{x})$ with

$$\mathbf{a}_{\mathsf{s}} = \arg\min_{\mathbf{a}}\xi\left(\frac{1}{N_2}\|\mathbf{y}_2^{\mathsf{t}} - \sigma(\mathbf{X}_2^\mathsf{T}\mathbf{W}_1^\mathsf{T})\mathbf{a}\|^2\right) + (1-\xi)\left(\frac{1}{N_2}\|\mathbf{y}_2 - \sigma(\mathbf{X}_2^\mathsf{T}\mathbf{W}_1^\mathsf{T})\mathbf{a}\|^2\right) + \lambda_{\mathsf{s}}\|\mathbf{a}\|^2.$$

We also examine a setting where the teacher model is a deeper neural network. Specifically, while keeping the student model fixed, we let the teacher be a Four-layer fully connected network:

$$f_{\mathrm{NN}}^{\mathsf{t}} = \mathbf{a}_{\mathsf{t}}^\mathsf{T}\sigma(\widetilde{\mathbf{W}}_3\sigma(\widetilde{\mathbf{W}}_2\sigma(\widetilde{\mathbf{W}}_1\mathbf{x}))),$$

where

$$\mathbf{a}_{\mathsf{t}} = \arg\min_{\mathbf{a}}\frac{1}{N_1}\|\mathbf{y}_1 - [\sigma(\widetilde{\mathbf{W}}_3\sigma(\widetilde{\mathbf{W}}_2\sigma(\widetilde{\mathbf{W}}_1\mathbf{X}_1)))]^\mathsf{T}\mathbf{a}\|^2 + \lambda_{\mathsf{t}}\|\mathbf{a}\|^2.$$

We set $D_1 = \mathcal{N}(0, 4\mathbf{I}_M)$ and $D_2 = \mathcal{N}(0, \mathbf{I}_M)$. Let $f(\mathbf{x}) = (\boldsymbol{\beta}^\mathsf{T}\mathbf{x})^2 + 1, \quad \tilde{f}(\mathbf{x}) = (\boldsymbol{\beta}^\mathsf{T}\mathbf{x})^2$. Because $D_1 \ne D_2$ and $f \ne \tilde{f}$, both covariate shift and model shift are present in this setting. More details and the numerical results are provided in Appendix C.1.

## 5.2 DEPENDENCE BETWEEN DOMAINS

In this section, we consider two cases in which $\mathbf{X}_1$ and $\mathbf{X}_2$ are not fully independent. Case 1: Assume $\mathbf{X}_1$ exhibits weak dependence on $\mathbf{X}_2$ in the following sense: $\mathbf{X}_1 = \alpha\mathbf{X}_2 + \widetilde{\mathbf{X}}_1$, where $\alpha \to 0$ as $M \to \infty$, and $\widetilde{\mathbf{X}}_1$ is independent of $\mathbf{X}_2$ and takes the form $\boldsymbol{\Sigma}_1^{1/2}\mathbf{Z}_1$. It is easy to see $\mathrm{Cov}(\mathbf{x}_j^{(1)}, \mathbf{x}_j^{(2)}) = \mathrm{Cov}(\mathbf{x}_j^{(2)}, \alpha\mathbf{x}_j^{(2)}) = \alpha\boldsymbol{\Sigma}_2$. Case 2: Suppose $\mathbf{X}_1$ is a signal-plus-noise data matrix: $\mathbf{X}_1 = \mathbf{X}_2 + \mathbf{A}$, where $\mathbf{A}$ is a deterministic signal matrix with $\|\mathbf{A}\| = o(\sqrt{M})$. This model captures realistic scenarios in domain adaptation where the source and target domains share a common underlying data matrix, but differ by a small deterministic shift—such as a faint shared signal across features in source domain. In Case 1, our theoretical analysis remains valid. For Case 2, we obtain a new limiting behavior; the theoretical results and technical details are provided in Section B.9.

## 6 CONCLUSION

In this paper, we present a theoretical analysis of cross-domain KD for linear models using random matrix theory. Through the bias-variance decomposition, we precisely characterize the asymptotic expressions of excess risk for the student model in the high-dimensional setting. A surprising finding is that when the imitation parameter $\xi$ is allowed to take any real value, cross-domain KD may outperform training solely on the target domain – even in the presence of significant discrepancies between source and target domains. This highlights the potential of distillation to effectively transfer knowledge across highly heterogeneous domains.

Our work also points to several promising directions for future research. Our theoretical analysis is currently limited to linear models; extending it to more complex architectures, particularly nonlinear models, would significantly broaden its applicability. Furthermore, while we observe the double descent phenomenon using the established theoretical limits; a rigorous theoretical characterization of this behavior in nonlinear models remains an important avenue for future investigation.

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

## A  THE USE OF LARGE LANGUAGE MODELS (LLMS)

Qwen3 is used to polish the writing.

## B  PROOFS

### B.1  BASIC TOOLS

Preliminary definitions and auxiliary lemmas used in the proofs of the main results are provided in this section.

**Lemma 3.** *(Lemma B.26 in Bai & Silverstein (2010)) Let $\mathbf{C}$ be an $M \times M$ deterministic matrix and $\mathbf{z} \in \mathbb{R}^M$ be a random vector of independent entries. Assume that $\mathbb{E}x_i = 0, \mathbb{E}|x_i|^2 = 1$, and $\mathbb{E}|x_i|^\ell \le C_\ell$. Then for any $\ell \ge 1$,*

$$\mathbb{E}|\mathbf{z}^\mathsf{T}\mathbf{C}\mathbf{z} - \mathrm{Tr}\mathbf{C}|^\ell \le c_\ell\big((C_4\mathrm{Tr}\mathbf{C}\mathbf{C}^*)^{\ell/2} + C_{2\ell}\mathrm{Tr}(\mathbf{C}\mathbf{C}^*)^{\ell/2}\big),$$

*where $c_\ell$ is a constant depending on $\ell$ only.*

Before stating the subsequent results, it is convenient to introduce the notion of stochastic domination.

**Definition 1.** *Let $\chi = \chi^{(p)}$, $\zeta = \zeta^{(p)}$ be two families of $p$-dependent random variables. We say that $\chi$ is stochastically dominated by $\zeta$ if for all small $c > 0$ and large constant $\ell > 0$,*

$$\mathbb{P}\big(|\chi^{(p)}| > p^c|\zeta^{(p)}|\big) \le p^{-\ell}$$

*for all large $p$. If $\chi$ is stochastically dominated by $\zeta$, we use the notation $\chi \prec \zeta$ or $\chi = O_\prec(\zeta)$. We say an event $\mathcal{E}_p$ holds with high probability if*

$$\mathbb{P}(\mathcal{E}_p^C) \le p^{-\ell} \text{ for any fixed } \ell > 0.$$

**Lemma 4.** *(Lemma 22 in Yang et al. (2025)) Let $\mathbf{Z}$ satisfies Assumption 1(a)-(b). Suppose $\frac{M}{N} \le 1 - \tau$. Then we have*

$$(\sqrt{M} - \sqrt{N})^2 + O_\prec(\sqrt{N}) \le \lambda_{\min}(\mathbf{Z}\mathbf{Z})^\mathsf{T} \le \lambda_{\max}(\mathbf{Z}\mathbf{Z}^\mathsf{T}) \le (\sqrt{M} + \sqrt{N})^2 + O_\prec(\sqrt{N}).$$

*The upper bound on $\lambda_{\max}(\mathbf{Z}\mathbf{Z}^\mathsf{T})$ still holds without the assumption $M/N \le 1 - \tau$.*

**Lemma 5.** *(Corollary 25 in Yang et al. (2025)) Suppose $\varepsilon_1, ..., \varepsilon_t$ are independent random vectors satisfying Assumption 1(c). Then, we have that for any deterministic vector $\mathbf{v} \in \mathbb{R}^N$,*

$$|\mathbf{v}^\mathsf{T}\varepsilon_i| \prec \sigma\|\mathbf{v}\|, \ i = 1, ..., t,$$

*and for any deterministic matrix $\mathbf{B} \in \mathbb{R}^{N \times N}$,*

$$|\varepsilon_i^\mathsf{T}\mathbf{B}\varepsilon_j - \delta(i = j)\sigma^2\mathrm{Tr}\mathbf{B}| \prec \sigma^2\|\mathbf{B}\|_\mathsf{F}, \ for \ i, j \in [t].$$

*Moreover, for any deterministic vector $\mathbf{v}$, we have*

$$|\mathbf{v}^\mathsf{T}\varepsilon_i| \prec \sigma\|\mathbf{v}\|, \ i \in [t].$$

**Definition 2.** *Let $\mathbf{A}_p, \mathbf{B}_p \in \mathbb{R}^{p \times p}$ be sequences of random or deterministic symmetric real matrices. We say $\mathbf{A}_p, \mathbf{B}_p$ are equivalent, denoted by $\mathbf{A}_p \asymp \mathbf{B}_p$, if*

$$\frac{1}{p}\mathrm{Tr}\mathbf{D}_p(\mathbf{A}_p - \mathbf{B}_p) = o_{a.s.}(1) \text{ and } \mathbf{u}^\mathsf{T}(\mathbf{A}_p - \mathbf{B}_p)\mathbf{v} = o_{a.s.}(1)$$

*for any sequence of deterministic matrices $\mathbf{D}_p$ and all deterministic vectors $\mathbf{u}, \mathbf{v}$ such that*

$$\limsup_p \|\mathbf{D}_p\| < \infty, \limsup_p \max\{\|\mathbf{u}\|, \|\mathbf{v}\|\} < \infty.$$

**Lemma 6.** *(1) (Theorem 2.6 in Couillet & Liao (2022)) We denote by $\varrho$ the probability measure associated with $m$ determined in Lemma 2. Let $\mathbf{X} = \mathbf{\Sigma}^{1/2}\mathbf{Z} \in \mathbb{R}^{M \times N}$, where the entries of $\mathbf{Z}$*

*are i.i.d. centered random variables with unit variance and finite $8 + c$-th moment, for any positive constant c. Suppose Assumption 1(b) and Assumption 2 hold. Then for $z \in \mathbb{C} \backslash \mathbb{R}^+$, we have*

$$\mathbf{Q}(z) \asymp \mathbf{\Pi}(z), \quad \underline{\mathbf{Q}}(z) \asymp m(z)\mathbf{I}_M, \tag{14}$$

*where*

$$\mathbf{Q}(z) = \left(\frac{\mathbf{X}\mathbf{X}^\mathsf{T}}{N} - z\mathbf{I}_M\right)^{-1}, \quad \underline{\mathbf{Q}}(z) = \left(\frac{\mathbf{X}^\mathsf{T}\mathbf{X}}{N} - z\mathbf{I}_M\right)^{-1}, \quad \mathbf{\Pi}(z) = -(z\mathbf{I}_M + zm\mathbf{\Sigma})^{-1}.$$

*(2) When $\frac{M}{N} < 1 - \tau$, equation 14 still holds at $z = 0$.*

**Proof:** Let $\mathbf{u}, \mathbf{v}$ be two deterministic unit vectors and $f_N(\lambda) = \mathbf{u}^\mathsf{T}\mathbf{Q}(-\lambda)\mathbf{v}$ for $\lambda < 0$. Since $\lambda_{\min}(\frac{\mathbf{X}\mathbf{X}^\mathsf{T}}{N}) > \frac{1}{2}(1 - \sqrt{M/N})^2$ with high probability, we have $|f_N(\lambda)| \leq \|\mathbf{Q}(-\lambda)\| \lesssim 1$, $|f'_N(\lambda)| \leq \|\frac{\mathrm{d}}{\mathrm{d}\lambda}\mathbf{Q}(-\lambda)\| \lesssim 1$ with high probability. Therefore, $\{f_N(\lambda)\}$ is equicontinuous with high probability. By applying the Arzela-Ascol theorem, $f_N$ converges uniformly to its limit $f(\lambda) = \mathbf{u}^\mathsf{T}\mathbf{\Pi}(-\lambda)\mathbf{v}$. By the Moore-Osgood theorem, we can exchange limits and get

$$\lim_{N \to \infty} f_N(0) = \lim_{N \to \infty}\lim_{\lambda \to 0^-} f_N(\lambda) = \lim_{\lambda \to 0^-}\lim_{N \to \infty} f_N(\lambda) = \lim_{\lambda \to 0^-} \mathbf{u}^\mathsf{T}\mathbf{\Pi}(-\lambda)\mathbf{v} = \mathbf{u}^\mathsf{T}\mathbf{\Pi}(0)\mathbf{v}, \ a.s.,$$

where we use the fact that both $\mathbf{Q}$ and $\mathbf{\Pi}$ are analytic in an open neighborhood of 0 with high probability. Similarly, we can derive $\frac{1}{M}\mathrm{Tr}\mathbf{A}\mathbf{Q}(0) = \frac{1}{M}\mathrm{Tr}\mathbf{A}\mathbf{\Pi}(0) + o_{a.s.}(1)$, which completes the proof.

Lemma 6 shows that $\mathbf{\Pi}(z)$ is a deterministic equivalent of $\mathbf{Q}(z)$. For technical reasons, we further require the following result.

**Lemma 7.** *Suppose the conditions in Lemma 6 hold. $\mathbf{A}$ denotes a deterministic $M \times M$ matrix with bounded spectral norm. For any fixed complex numbers $\tilde{z}_1, \tilde{z}_2 \in \mathbb{C} \backslash \mathbb{R}^+$, we have for all deterministic vectors $\mathbf{u}, \mathbf{v}$,*

$$\mathbf{u}^\mathsf{T}\big(\mathbf{Q}(\tilde{z}_1)\mathbf{A}\mathbf{Q}(\tilde{z}_2) - \mathbf{\Pi}(\tilde{z}_1)\mathcal{S}(\mathbf{A})\mathbf{\Pi}(\tilde{z}_2)\big)\mathbf{v} = o_{a.s.}(\|\mathbf{u}\|\|\mathbf{v}\|), \tag{15}$$

*where*

$$\mathcal{S}(\mathbf{A}) = \mathbf{A} + \frac{\frac{1}{N}\mathrm{Tr}\mathbf{\Sigma}\mathbf{\Pi}(\tilde{z}_1)\mathbf{A}\mathbf{\Pi}(\tilde{z}_2)}{\big(1 + \frac{1}{N}\mathrm{Tr}\mathbf{\Sigma}\mathbf{\Pi}(\tilde{z}_2)\big)\big(1 + \frac{1}{N}\mathrm{Tr}\mathbf{\Sigma}\mathbf{\Pi}(\tilde{z}_1)\big) - \frac{1}{N}\mathrm{Tr}\mathbf{\Sigma}\mathbf{\Pi}(\tilde{z}_1)\mathbf{\Sigma}\mathbf{\Pi}(\tilde{z}_2)}\mathbf{\Sigma}. \tag{16}$$

*Moreover, for any deterministic matrix $\mathbf{C} \in \mathbb{R}^{M \times M}$ satisfying $\|\mathbf{C}\| \leq C$ for some constant C, we have*

$$\frac{1}{M}\mathrm{Tr}\mathbf{C}[\mathbf{Q}(\tilde{z}_1)\mathbf{A}\mathbf{Q}(\tilde{z}_2) - \mathbf{\Pi}_1(\tilde{z}_1)\mathcal{S}(\mathbf{A})\mathbf{\Pi}_2(\tilde{z}_2)] = o_{a.s.}(1). \tag{17}$$

The proof of this lemma is deferred to Appendix B.8.

**Remark 3.** *Lemma 7 provides the deterministic equivalent of $\mathbf{Q}(\tilde{z}_1)\mathbf{A}\mathbf{Q}(\tilde{z}_2)$. Lin & Pan (2024) established the local laws for the $\mathbf{Q}(\tilde{z}_1)\mathbf{A}\mathbf{Q}(\tilde{z}_2)$. However, their results require $\Re\tilde{z}_1, \Re\tilde{z}_2$ to be sufficiently close to $\mathrm{supp}(\varrho)$ and $\Im\tilde{z}_1, \Im\tilde{z}_2$ to be bounded below by $N^{-1+c}$, where c is any fixed constant. Lemma 7 extends the result to other regions.*

**Remark 4.** *To relax the moment assumption, we apply a standard truncation argument commonly used in random matrix theory (e.g., Yang et al. (2025)). This approach allows us to employ Lemma 4 under the weaker finite $(8 + c)$-th moment condition, introducing only a negligible additional error term that depends on $M$ but does not affect the leading-order asymptotics of our results. Moreover, a careful examination of the proofs shows that the same moment condition is also sufficient to establish Lemma 7. Consequently, all of our theoretical conclusions remain valid under this relaxed assumption.*

### B.2 PROOF OF THEOREM 1

To simplify notation, we set $z_1 = -\lambda_t, z_2 = -\lambda_s$. Recalling equation 2 and equation 3, we get

$$
\begin{aligned}
\boldsymbol{\beta}_s - \boldsymbol{\beta}_2 &= \frac{1}{N_2}\mathbf{Q}_2\big[\xi\mathbf{X}_2\mathbf{X}_2^\mathsf{T}\boldsymbol{\beta}_t + (1-\xi)\mathbf{X}_2(\mathbf{X}_2^\mathsf{T}\boldsymbol{\beta}_2 + \boldsymbol{\varepsilon}_2)\big] - \boldsymbol{\beta}_2 \\
&= \frac{1}{N_2}\mathbf{Q}_2\bigg[\frac{1}{N_1}\xi\mathbf{X}_2\mathbf{X}_2^\mathsf{T}\mathbf{Q}_1\mathbf{X}_1(\mathbf{X}_1^\mathsf{T}\boldsymbol{\beta}_1 + \boldsymbol{\varepsilon}_1) + (1-\xi)\mathbf{X}_2(\mathbf{X}_2^\mathsf{T}\boldsymbol{\beta}_2 + \boldsymbol{\varepsilon}_2)\bigg] - \boldsymbol{\beta}_2 \\
&= \xi\big[(\mathbf{I}_M + z_2\mathbf{Q}_2)(\mathbf{I}_M + z_1\mathbf{Q}_1)\boldsymbol{\beta}_1 - \boldsymbol{\beta}_2\big] + \xi\underbrace{\frac{1}{N_1}(\mathbf{I}_M + z_2\mathbf{Q}_2)\mathbf{Q}_1\mathbf{X}_1\boldsymbol{\varepsilon}_1}_{\boldsymbol{a}_5} \\
&\quad + (1-\xi)\big[(\mathbf{I}_M + z_2\mathbf{Q}_2)\boldsymbol{\beta}_2 - \boldsymbol{\beta}_2\big] + (1-\xi)\underbrace{\frac{1}{N_2}\mathbf{Q}_2\mathbf{X}_2\boldsymbol{\varepsilon}_2}_{\boldsymbol{a}_6} \\
&= \xi\boldsymbol{\gamma} + \underbrace{\xi z_2\mathbf{Q}_2\boldsymbol{\gamma}}_{\boldsymbol{a}_1} + \underbrace{\xi z_1\mathbf{Q}_1\boldsymbol{\beta}_1}_{\boldsymbol{a}_2} + \underbrace{z_2\mathbf{Q}_2\boldsymbol{\beta}_2}_{\boldsymbol{a}_3} + \underbrace{\xi z_1 z_2\mathbf{Q}_2\mathbf{Q}_1\boldsymbol{\beta}_1}_{\boldsymbol{a}_4} + \boldsymbol{a}_5 + \boldsymbol{a}_6.
\end{aligned}
\tag{18}
$$

By this, we decompose $\mathbf{ER}(\boldsymbol{\beta}_s)$ as follows:

$$
\begin{aligned}
\mathbf{ER}(\boldsymbol{\beta}_s) &= (\boldsymbol{\beta}_s - \boldsymbol{\beta}_2)^\mathsf{T}\boldsymbol{\Sigma}_2(\boldsymbol{\beta}_s - \boldsymbol{\beta}_2) \\
&= \xi^2\boldsymbol{\gamma}^\mathsf{T}\boldsymbol{\Sigma}_2\boldsymbol{\gamma} + 2\sum_{i=1}^{6}b_i + \sum_{i=1}^{6}h_{ii} + \sum_{1\leq i\neq j\leq 6}h_{ij},
\end{aligned}
\tag{19}
$$

where

$$
b_i = \xi\boldsymbol{\gamma}^\mathsf{T}\boldsymbol{\Sigma}_2\boldsymbol{a}_i, \ h_{ii} = \big\|\boldsymbol{\Sigma}_2^{1/2}\boldsymbol{a}_i\big\|^2, \ h_{ij} = \boldsymbol{a}_i^\mathsf{T}\boldsymbol{\Sigma}_2\boldsymbol{a}_j.
$$

Next, we compute the limit of each term above.

Let $n \in \mathbb{N}^+$. According to the Definition 1 and the Borel–Cantelli lemma, we have

$$
\chi(n) = o_{a.s.}(1) \text{ if } \chi(n) \prec n^{-c}
$$

for any constant $c > 0$. By this, the limits of $b_1, b_2, b_3, b_4$ can be readily obtained using Lemma 6:

$$
\begin{aligned}
b_1 &= \xi^2 z_2\boldsymbol{\gamma}^\mathsf{T}\mathbf{Q}_2\boldsymbol{\Sigma}_2\boldsymbol{\gamma} = \xi^2 z_2\boldsymbol{\gamma}^\mathsf{T}\boldsymbol{\Pi}_2\boldsymbol{\Sigma}_2\boldsymbol{\gamma} + o_{a.s.}(1), \\
b_2 &= \xi^2 z_1\boldsymbol{\gamma}^\mathsf{T}\boldsymbol{\Sigma}_2\mathbf{Q}_1\boldsymbol{\beta}_1 = \xi^2 z_1\boldsymbol{\gamma}^\mathsf{T}\boldsymbol{\Sigma}_2\boldsymbol{\Pi}_1\boldsymbol{\beta}_1 + o_{a.s.}(1), \\
b_3 &= \xi z_2\boldsymbol{\beta}_2^\mathsf{T}\mathbf{Q}_2\boldsymbol{\Sigma}_2\boldsymbol{\gamma} = \xi z_2\boldsymbol{\beta}_2^\mathsf{T}\boldsymbol{\Pi}_2\boldsymbol{\Sigma}_2\boldsymbol{\gamma} + o_{a.s.}(1), \\
b_4 &= \xi^2 z_1 z_2\boldsymbol{\gamma}^\mathsf{T}\boldsymbol{\Sigma}_2\mathbf{Q}_2\mathbf{Q}_1\boldsymbol{\beta}_1 = \xi^2 z_1 z_2\boldsymbol{\gamma}^\mathsf{T}\boldsymbol{\Sigma}_2\boldsymbol{\Pi}_2\boldsymbol{\Pi}_1\boldsymbol{\beta}_1 + o_{a.s.}(1),
\end{aligned}
$$

where the last identity is due to

$$
\boldsymbol{\gamma}^\mathsf{T}\boldsymbol{\Sigma}_2[\mathbf{Q}_2\mathbf{Q}_1 - \boldsymbol{\Pi}_2\boldsymbol{\Pi}_1]\boldsymbol{\beta}_1 = \boldsymbol{\gamma}^\mathsf{T}\boldsymbol{\Sigma}_2\big[(\mathbf{Q}_2 - \boldsymbol{\Pi}_2)\mathbf{Q}_1 + \boldsymbol{\Pi}_2(\mathbf{Q}_1 - \boldsymbol{\Pi}_1)\big]\boldsymbol{\beta}_1 = o_{a.s.}(1).
$$

We now consider the terms contributing to $\mathbf{Var}$. By Lemma 5 and the identity

$$
\frac{1}{N_i}\mathbf{Q}_i\mathbf{X}_i\mathbf{X}_i^\mathsf{T} = \mathbf{I}_M + z_1\mathbf{Q}_i, \ i = 1, 2.
\tag{20}
$$

we find that

$$
\begin{aligned}
\Big|h_{55} &- \frac{\xi^2\sigma^2}{N_1^2}\mathrm{Tr}(\mathbf{I}_M + z_2\mathbf{Q}_2)\boldsymbol{\Sigma}_2(\mathbf{I}_M + z_2\mathbf{Q}_2)\mathbf{Q}_1\mathbf{X}_1\mathbf{X}_1^\mathsf{T}\mathbf{Q}_1\Big| \\
&\prec \frac{1}{M^2}\|(\mathbf{I}_M + z_2\mathbf{Q}_2)\boldsymbol{\Sigma}_2(\mathbf{I}_M + z_2\mathbf{Q}_2)(\mathbf{Q}_1 + z_1\mathbf{Q}_1^2)\|_\mathsf{F} \lesssim \frac{1}{\sqrt{M}}.
\end{aligned}
\tag{21}
$$

For any deterministic matrix $\mathbf{C} \in \mathbb{R}^{M\times M}$ satisfying $\|\mathbf{C}\|$ is bounded, and having the spectral decomposition

$$
\mathbf{C} = \sum_{i=1}^{M}\lambda_i\mathbf{u}_i\mathbf{v}_i^\mathsf{T},
$$

we have by Lemma 6 that

$$
\begin{aligned}
\frac{1}{M}\mathrm{Tr}\mathbf{C}\mathbf{Q}_1\mathbf{Q}_2 &= \frac{1}{M}\sum_{i=1}^{M}\lambda_i\mathrm{Tr}\mathbf{u}_i\mathbf{v}_i^\mathsf{T}\mathbf{Q}_1\mathbf{Q}_2 \\
&= \frac{1}{M}\sum_{i=1}^{M}\lambda_i\mathbf{v}_i^\mathsf{T}\mathbf{Q}_1\mathbf{Q}_2\mathbf{u}_i \\
&= \frac{1}{M}\sum_{i=1}^{M}\lambda_i\mathbf{v}_i^\mathsf{T}\mathbf{\Pi}_1\mathbf{\Pi}_2\mathbf{u}_i + o_{a.s.}(1) \\
&= \frac{1}{M}\mathrm{Tr}\mathbf{C}\mathbf{\Pi}_1\mathbf{\Pi}_2 + o_{a.s.}(1).
\end{aligned}
\tag{22}
$$

Similarly, by recalling the notation $\mathbf{\Pi}_i\mathcal{S}_i(\mathbf{I}_M)\mathbf{\Pi}_i = \mathbf{\Pi}_i'$ for $i = 1, 2$, one may check by Lemma 7 that

$$
\begin{aligned}
\frac{1}{M}\mathrm{Tr}\mathbf{C}\mathbf{Q}_2\mathbf{Q}_1^2 &= \frac{1}{M}\mathrm{Tr}\mathbf{C}\mathbf{\Pi}_2\mathbf{\Pi}_1' + o_{a.s.}(1), \\
\frac{1}{M}\mathrm{Tr}\mathbf{C}\mathbf{Q}_1\mathbf{Q}_2^2 &= \frac{1}{M}\mathrm{Tr}\mathbf{C}\mathbf{\Pi}_1\mathbf{\Pi}_2', \\
\frac{1}{M}\mathrm{Tr}\mathbf{Q}_2\mathbf{C}\mathbf{Q}_2\mathbf{Q}_1^2 &= \frac{1}{M}\mathrm{Tr}\mathbf{\Pi}_2\mathcal{S}_2(\mathbf{C})\mathbf{\Pi}_2\mathbf{\Pi}_1' + o_{a.s.}(1), \\
\frac{1}{M}\mathrm{Tr}\mathbf{Q}_1\mathbf{C}\mathbf{Q}_1\mathbf{Q}_2^2 &= \frac{1}{M}\mathrm{Tr}\mathbf{\Pi}_1\mathcal{S}_1(\mathbf{C})\mathbf{\Pi}_1\mathbf{\Pi}_2' + o_{a.s.}(1).
\end{aligned}
\tag{23}
$$

Then by Lemma 6, Lemma 7, equation 20 and equation 21, for $\xi \neq 0$, we have

$$
\begin{aligned}
\frac{h_{55}}{\xi^2\sigma^2} &= \frac{1}{N_1}\mathrm{Tr}(\mathbf{I}_M + z_2\mathbf{Q}_2)\mathbf{\Sigma}_2(\mathbf{I}_M + z_2\mathbf{Q}_2)(\mathbf{Q}_1 + z_1\mathbf{Q}_1^2) \\
&= \frac{1}{N_1}\mathrm{Tr}\big[\mathbf{\Sigma}_2 + z_2\mathbf{\Sigma}_2\mathbf{Q}_2 + z_2\mathbf{Q}_2\mathbf{\Sigma}_2 + z_2^2\mathbf{Q}_2\mathbf{\Sigma}_2\mathbf{Q}_2\big]\big[\mathbf{\Pi}_1 + z_1\mathbf{\Pi}_1'\big] + o_{a.s.}(1) \\
&= \frac{1}{N_1}\mathrm{Tr}\big[\mathbf{\Sigma}_2 + 2z_2\mathbf{\Sigma}_2\mathbf{\Pi}_2 + z_2^2\mathbf{\Pi}_2\mathcal{S}_2(\mathbf{\Sigma}_2)\mathbf{\Pi}_2\big]\big[\mathbf{\Pi}_1 + z_1\mathbf{\Pi}_1'\big] \\
&\quad + o_{a.s.}(1).
\end{aligned}
$$

Likewise, we have by Lemma 6 that

$$
\begin{aligned}
h_{66} &= (1-\xi)^2\sigma^2\frac{1}{N_2}\mathrm{Tr}\mathbf{\Sigma}_2(\mathbf{I}_M + z_2\mathbf{Q}_2)\mathbf{Q}_2 \\
&= (1-\xi)^2\sigma^2\frac{1}{N_2}\mathrm{Tr}\mathbf{\Sigma}_2(\mathbf{\Pi}_2 + z_2\mathbf{\Pi}_2') + o_{a.s.}(1).
\end{aligned}
\tag{24}
$$

Let $d = \min\big\{\mathrm{dist}\big(z_1, \mathbb{R}^+\big), \mathrm{dist}\big(z_2, \mathbb{R}^+\big)\big\}$. According to Lemma 5 and the fact following from equation 20 that

$$
\frac{1}{\sqrt{N_i}}\|\mathbf{Q}_i\mathbf{X}_i\| = \sqrt{\|\mathbf{Q}_i + z_i\mathbf{Q}_i^2\|} \leq \sqrt{d^{-1} + d^{-2}|z_i|} \lesssim 1,
$$

one has for $j = 1, 2, 3, 4$,

$$
|h_{5j}| = |h_{j5}| \prec \frac{\sigma}{M}\|\mathbf{X}_1^\mathsf{T}\mathbf{Q}_1(\mathbf{I}_M + z_1\mathbf{Q}_2)\mathbf{\Sigma}_2\boldsymbol{a}_j\| \lesssim \frac{1}{\sqrt{M}},
$$

$$
|h_{6j}| = |h_{j6}| \prec \frac{\sigma}{M}\|\mathbf{X}_2^\mathsf{T}\mathbf{Q}_2\mathbf{\Sigma}_2\boldsymbol{a}_j\| \lesssim \frac{1}{\sqrt{M}},
$$

and

$$
|b_5 + b_6| \prec \frac{1}{\sqrt{M}}.
$$

Using Lemma 5 again, it can be shown that

$$|h_{65}| = |h_{56}| \prec \frac{\sigma^2}{M^2}\|\mathbf{X}_2^\mathsf{T}\mathbf{Q}_2\mathbf{\Sigma}_2(\mathbf{I}_M + z_2\mathbf{Q}_2)\mathbf{Q}_1\mathbf{X}_1\|_\mathsf{F}$$

$$\lesssim \frac{\sigma^2}{M}\sqrt{\frac{N_2\|\mathbf{X}_2^\mathsf{T}\mathbf{Q}_2\mathbf{\Sigma}_2(\mathbf{I}_M + z_2\mathbf{Q}_2)\mathbf{Q}_1\mathbf{X}_1\|^2}{M^2}} \lesssim \frac{1}{\sqrt{M}}.$$

Therefore, we get

$$b_5 + b_6 + h_{65} + h_{56} + \sum_{j=1}^{4}(h_{5j} + h_{j5} + h_{6j} + h_{j6}) = o_{a.s.}(1).$$

We now turn to the terms $h_{ii}, i = 1, 2, 3, 4$. By Lemma 7, we have

$$h_{11} = \xi^2 z_2^2 \boldsymbol{\gamma}^\mathsf{T}\mathbf{Q}_2\mathbf{\Sigma}_2\mathbf{Q}_2\boldsymbol{\gamma}$$
$$= \xi^2 z_2^2 \boldsymbol{\gamma}^\mathsf{T}\mathbf{\Pi}_2\mathcal{S}_2(\mathbf{\Sigma}_2)\mathbf{\Pi}_2\boldsymbol{\gamma} + o_{a.s.}(1),$$
$$h_{22} = \xi^2 z_1^2 \boldsymbol{\beta}_1^\mathsf{T}\mathbf{Q}_1\mathbf{\Sigma}_2\mathbf{Q}_1\boldsymbol{\beta}_1$$
$$= \xi^2 z_1^2 \boldsymbol{\beta}_1^\mathsf{T}\mathbf{\Pi}_1\mathcal{S}_1(\mathbf{\Sigma}_2)\mathbf{\Pi}_1\boldsymbol{\beta}_1 + o_{a.s.}(1),$$
$$h_{33} = z_2^2 \boldsymbol{\beta}_2^\mathsf{T}\mathbf{Q}_2\mathbf{\Sigma}_2\mathbf{Q}_2\boldsymbol{\beta}_2$$
$$= z_2^2 \boldsymbol{\beta}_2^\mathsf{T}\mathbf{\Pi}_2\mathcal{S}_2(\mathbf{\Sigma}_2)\mathbf{\Pi}_2\boldsymbol{\beta}_2 + o_{a.s.}(1),$$
$$h_{44} = \xi^2 z_1^2 z_2^2 \boldsymbol{\beta}_1^\mathsf{T}\mathbf{Q}_1\mathbf{Q}_2\mathbf{\Sigma}_2\mathbf{Q}_2\mathbf{Q}_1\boldsymbol{\beta}_1$$
$$= \xi^2 z_1^2 z_2^2 \mathbb{E}\boldsymbol{\beta}_1^\mathsf{T}\mathbf{Q}_1\mathbf{\Pi}_2\mathcal{S}_2(\mathbf{\Sigma}_2)\mathbf{\Pi}_2\mathbf{Q}_1\boldsymbol{\beta}_1 + o_{a.s.}(1)$$
$$= \xi^2 z_1^2 z_2^2 \boldsymbol{\beta}_1^\mathsf{T}\mathbf{\Pi}_1\mathcal{S}_1(\mathbf{\Pi}_2\mathcal{S}_2(\mathbf{\Sigma}_2)\mathbf{\Pi}_2)\mathbf{\Pi}_1\boldsymbol{\beta}_1 + o_{a.s.}(1).$$

Similarly, one can obtain the limits of the remaining terms in $h_{ij}, 1 \le i, j \le 6$:

$$h_{12} = h_{21} = \xi^2 z_1 z_2 \boldsymbol{\gamma}^\mathsf{T}\mathbf{Q}_2\mathbf{\Sigma}_2\mathbf{Q}_1\boldsymbol{\beta}_1$$
$$= \xi^2 z_1 z_2 \boldsymbol{\gamma}^\mathsf{T}\mathbf{\Pi}_2\mathbf{\Sigma}_2\mathbf{\Pi}_1\boldsymbol{\beta}_1 + o_{a.s.}(1),$$
$$h_{13} = h_{31} = \xi z_2^2 \boldsymbol{\beta}_2^\mathsf{T}\mathbf{Q}_2\mathbf{\Sigma}_2\mathbf{Q}_2\boldsymbol{\gamma}$$
$$= \xi z_2^2 \boldsymbol{\beta}_2^\mathsf{T}\mathbf{\Pi}_2\mathcal{S}_2(\mathbf{\Sigma}_2)\mathbf{\Pi}_2\boldsymbol{\gamma} + o_{a.s.}(1),$$
$$h_{14} = h_{41} = \xi^2 z_1 z_2^2 \boldsymbol{\gamma}^\mathsf{T}\mathbf{Q}_2\mathbf{\Sigma}_2\mathbf{Q}_2\mathbf{Q}_1\boldsymbol{\beta}_1$$
$$= \xi^2 z_1 z_2^2 \boldsymbol{\gamma}^\mathsf{T}\mathbf{\Pi}_2\mathcal{S}_2(\mathbf{\Sigma}_2)\mathbf{\Pi}_2\mathbf{\Pi}_1\boldsymbol{\beta}_1 + o_{a.s.}(1),$$
$$h_{23} = h_{32} = \xi z_1 z_2 \boldsymbol{\beta}_1^\mathsf{T}\mathbf{Q}_1\mathbf{\Sigma}_2\mathbf{Q}_2\boldsymbol{\beta}_2$$
$$= \xi z_1 z_2 \boldsymbol{\beta}_1^\mathsf{T}\mathbf{\Pi}_1\mathbf{\Sigma}_2\mathbf{\Pi}_2\boldsymbol{\beta}_2 + o_{a.s.}(1),$$
$$h_{24} = h_{42} = \xi^2 z_1^2 z_2 \boldsymbol{\beta}_1^\mathsf{T}\mathbf{Q}_1\mathbf{\Sigma}_2\mathbf{Q}_2\mathbf{Q}_1\boldsymbol{\beta}_1$$
$$= \xi^2 z_1^2 z_2 \boldsymbol{\beta}_1^\mathsf{T}\mathbf{Q}_1\mathbf{\Sigma}_2\mathbf{\Pi}_2\mathbf{Q}_1\boldsymbol{\beta}_1 + o_{a.s.}(1)$$
$$= \xi^2 z_1^2 z_2 \boldsymbol{\beta}_1^\mathsf{T}\mathbf{\Pi}_1\mathcal{S}_1(\mathbf{\Sigma}_2\mathbf{\Pi}_2)\mathbf{\Pi}_1\boldsymbol{\beta}_1 + o_{a.s.}(1),$$
$$h_{34} = h_{43} = \xi z_1 z_2^2 \boldsymbol{\beta}_2^\mathsf{T}\mathbf{Q}_2\mathbf{\Sigma}_2\mathbf{Q}_2\mathbf{Q}_1\boldsymbol{\beta}_1$$
$$= \xi z_1 z_2^2 \boldsymbol{\beta}_2^\mathsf{T}\mathbf{\Pi}_2\mathcal{S}_2(\mathbf{\Sigma}_2)\mathbf{\Pi}_2\mathbf{\Pi}_1\boldsymbol{\beta}_1 + o_{a.s.}(1).$$

Combining the above estimates, we conclude the proof of Theorem 1.

### B.3 PROOF OF THEOREM 2

We use the same notation as in Appendix B.2. Note that $\boldsymbol{\gamma} = \boldsymbol{\beta}_1 - \boldsymbol{\beta}_2 = 0$. Denoting

$$\mathbf{H} = \xi z_1 \mathbf{Q}_1 + z_2 \mathbf{Q}_2 + \xi z_1 z_2 \mathbf{Q}_2 \mathbf{Q}_1, \tag{25}$$

by equation 18, we have

$$\boldsymbol{\beta}_\mathsf{s} - \boldsymbol{\beta} = \mathbf{H}\boldsymbol{\beta} + \boldsymbol{a}_5 + \boldsymbol{a}_6.$$

Hence, the excess risk becomes

$$\mathbf{ER}(\boldsymbol{\beta}_{\mathsf{s}}) = \|\boldsymbol{\Sigma}_2^{1/2}(\boldsymbol{\beta}_{\mathsf{s}} - \boldsymbol{\beta})\|^2$$
$$= \boldsymbol{\beta}^{\mathsf{T}}\mathbf{H}^{\mathsf{T}}\boldsymbol{\Sigma}_2\mathbf{H}\boldsymbol{\beta} + 2\sum_{i=5,6}\boldsymbol{\beta}^{\mathsf{T}}\mathbf{H}^{\mathsf{T}}\boldsymbol{\Sigma}_2\boldsymbol{a}_i + \sum_{i=5,6}h_{ii}.$$

Using Lemma 5, by Assumption 3 we have

$$\boldsymbol{\beta}^{\mathsf{T}}\mathbf{H}^{\mathsf{T}}\boldsymbol{\Sigma}_2\mathbf{H}\boldsymbol{\beta} - \frac{\tilde{\sigma}^2}{M}\mathrm{Tr}\mathbf{H}^{\mathsf{T}}\boldsymbol{\Sigma}_2\mathbf{H} \prec \frac{\tilde{\sigma}^2}{M}\|\mathbf{H}^{\mathsf{T}}\boldsymbol{\Sigma}_2\mathbf{H}\|_{\mathsf{F}} \lesssim \frac{1}{\sqrt{M}}.$$

By equation 25, we have

$$\frac{1}{M}\mathrm{Tr}\boldsymbol{\Sigma}_2\mathbf{H}\mathbf{H}^{\mathsf{T}} = \frac{1}{M}\bigg[\xi^2 z_1^2\mathrm{Tr}\boldsymbol{\Sigma}_2\mathbf{Q}_1^2 + \xi z_1 z_2\mathrm{Tr}\boldsymbol{\Sigma}_2[\mathbf{Q}_1\mathbf{Q}_2 + \mathbf{Q}_2\mathbf{Q}_1]$$
$$+ z_2^2\mathrm{Tr}\boldsymbol{\Sigma}_2\mathbf{Q}_2^2 + \xi^2 z_1^2 z_2\mathrm{Tr}\boldsymbol{\Sigma}_2[\mathbf{Q}_2\mathbf{Q}_1^2 + \mathbf{Q}_1^2\mathbf{Q}_2]$$
$$+ 2\xi z_1 z_2^2\mathrm{Tr}\mathbf{Q}_2\boldsymbol{\Sigma}_2\mathbf{Q}_2\mathbf{Q}_1 + \xi^2 z_1^2 z_2^2\mathrm{Tr}\mathbf{Q}_2\boldsymbol{\Sigma}_2\mathbf{Q}_2\mathbf{Q}_1^2\bigg] \qquad (26)$$
$$= \sum_{i=1}^{6}t_i,$$

where

$$t_1 = \frac{1}{M}\xi^2 z_1^2\mathrm{Tr}\boldsymbol{\Sigma}_2\mathbf{Q}_1^2, t_2 = \frac{2}{M}\xi z_1 z_2\mathrm{Tr}\mathbf{Q}_1\mathbf{Q}_2, t_3 = \frac{z_2^2}{M}\mathrm{Tr}\boldsymbol{\Sigma}_2\mathbf{Q}_2^2,$$
$$t_4 = 2\frac{\xi^2}{M}z_1^2 z_2\mathrm{Tr}\boldsymbol{\Sigma}_2\mathbf{Q}_2\mathbf{Q}_1^2, t_5 = \frac{2\xi z_1 z_2^2}{M}\mathrm{Tr}\mathbf{Q}_2\boldsymbol{\Sigma}_2\mathbf{Q}_2\mathbf{Q}_1, t_6 = \frac{\xi^2 z_1^2 z_2^2}{M}\mathrm{Tr}\mathbf{Q}_2\boldsymbol{\Sigma}_2\mathbf{Q}_2\mathbf{Q}_1^2.$$

We next consider the terms $t_i, i = 1, ..., 6$. In the subsequent proof, we shall make use of Lemma 6, Lemma 7 and the property that $\boldsymbol{\Sigma}_2\boldsymbol{\Pi}_2 = \boldsymbol{\Pi}_2\boldsymbol{\Sigma}_2$.

By equation 22, we have

$$t_2 = \frac{2\xi z_1 z_2}{M}\mathrm{Tr}\boldsymbol{\Pi}_1\boldsymbol{\Pi}_2\boldsymbol{\Sigma}_2 + o_{a.s.}(1).$$

$$t_1 = \xi^2 z_1^2\frac{1}{M}\mathrm{Tr}\boldsymbol{\Sigma}_2\boldsymbol{\Pi}_1' + o_{a.s.}(1).$$

The limits of $t_3, t_4, t_5, t_6$ can be derived by equation 23:

$$t_3 = \frac{z_2^2}{M}\mathrm{Tr}\boldsymbol{\Sigma}_2\boldsymbol{\Pi}_2' + o_{a.s.}(1),$$

$$t_4 = 2\frac{\xi^2 z_1^2 z_2}{M}\mathrm{Tr}\boldsymbol{\Sigma}_2\boldsymbol{\Pi}_2\boldsymbol{\Pi}_1' + o_{a.s.}(1),$$

$$t_5 = \frac{2\xi z_1 z_2^2}{M}\mathrm{Tr}\boldsymbol{\Pi}_2\mathcal{S}_2(\boldsymbol{\Sigma}_2)\boldsymbol{\Pi}_2\boldsymbol{\Pi}_1 + o_{a.s.}(1),$$

and

$$t_6 = \frac{\xi^2 z_1^2 z_2^2}{M}\mathrm{Tr}\boldsymbol{\Pi}_2\mathcal{S}_2(\boldsymbol{\Sigma}_2)\boldsymbol{\Pi}_2\boldsymbol{\Pi}_1' + o_{a.s.}(1).$$

Using Lemma 5, we find

$$|\boldsymbol{\beta}^{\mathsf{T}}\mathbf{H}^{\mathsf{T}}\boldsymbol{\Sigma}_2\boldsymbol{a}_5| \prec \frac{1}{M^{3/2}}\|\mathbf{H}^{\mathsf{T}}\boldsymbol{\Sigma}_2(\mathbf{I}_M + z_2\mathbf{Q}_2)\mathbf{Q}_1\mathbf{X}_1\|_{\mathsf{F}} \lesssim \frac{1}{\sqrt{M}},$$

$$|\boldsymbol{\beta}^{\mathsf{T}}\mathbf{H}^{\mathsf{T}}\boldsymbol{\Sigma}_2\boldsymbol{a}_6| \prec \frac{1}{M^{3/2}}\|\mathbf{Q}_2\mathbf{X}_2\|_{\mathsf{F}} \lesssim \frac{1}{\sqrt{M}}.$$

Therefore, the terms $\boldsymbol{\beta}^{\mathsf{T}}\mathbf{H}^{\mathsf{T}}\boldsymbol{\Sigma}_2\boldsymbol{a}_i, i = 5, 6$ are ignorable. The proof is now complete.

### B.4 PROOF OF COROLLARY 3

Letting $\lambda_{\mathsf{t}} = \lambda_{\mathsf{s}} = 0$, by equation 3, we obtain

$$
\begin{aligned}
\boldsymbol{\beta}_{\mathsf{s}} &= \xi \boldsymbol{\beta}_1^{\mathsf{OLS}} + (1 - \xi) \boldsymbol{\beta}_2^{\mathsf{OLS}} \\
&= \xi (\mathbf{X}_1 \mathbf{X}_1^{\mathsf{T}})^{-1} \mathbf{X}_1 (\mathbf{X}_1^{\mathsf{T}} \boldsymbol{\beta}_1 + \boldsymbol{\varepsilon}_1) + (1 - \xi) (\mathbf{X}_2 \mathbf{X}_2^{\mathsf{T}})^{-1} \mathbf{X}_2 (\mathbf{X}_2^{\mathsf{T}} \boldsymbol{\beta}_2 + \boldsymbol{\varepsilon}_2) \\
&= \xi \boldsymbol{\beta}_1 + (1 - \xi) \boldsymbol{\beta}_2 + \xi (\mathbf{X}_1 \mathbf{X}_1^{\mathsf{T}})^{-1} \mathbf{X}_1 \boldsymbol{\varepsilon}_1 + (1 - \xi) (\mathbf{X}_2 \mathbf{X}_2^{\mathsf{T}})^{-1} \mathbf{X}_2 \boldsymbol{\varepsilon}_2 .
\end{aligned}
$$

Plugging this into $\mathbf{ER}(\boldsymbol{\beta}_{\mathsf{s}})$, one may obtain that

$$
\begin{aligned}
\mathbf{ER}(\boldsymbol{\beta}_{\mathsf{s}}) &= \| \boldsymbol{\Sigma}_2^{1/2} (\boldsymbol{\beta}_2 - \boldsymbol{\beta}_{\mathsf{s}}) \|^2 \\
&= \left\| \boldsymbol{\Sigma}_2^{1/2} [\xi \boldsymbol{\gamma} + \xi (\mathbf{X}_1 \mathbf{X}_1^{\mathsf{T}})^{-1} \mathbf{X}_1 \boldsymbol{\varepsilon}_1 + (1 - \xi) (\mathbf{X}_2 \mathbf{X}_2^{\mathsf{T}})^{-1} \mathbf{X}_2 \boldsymbol{\varepsilon}_2] \right\|^2 \\
&= \widehat{\mathbf{Bias}} + h_1 + h_2 + 2 h_3 + 2 h_4 + 2 h_5 ,
\end{aligned}
$$

where

$$
\begin{aligned}
h_1 &= \xi^2 \boldsymbol{\varepsilon}_1^{\mathsf{T}} \mathbf{X}_1^{\mathsf{T}} (\mathbf{X}_1 \mathbf{X}_1^{\mathsf{T}})^{-1} \boldsymbol{\Sigma}_2 (\mathbf{X}_1 \mathbf{X}_1^{\mathsf{T}})^{-1} \mathbf{X}_1 \boldsymbol{\varepsilon}_1 , \\
h_2 &= (1 - \xi)^2 \boldsymbol{\varepsilon}_2^{\mathsf{T}} \mathbf{X}_2^{\mathsf{T}} (\mathbf{X}_2 \mathbf{X}_2^{\mathsf{T}})^{-1} \boldsymbol{\Sigma}_2 (\mathbf{X}_2 \mathbf{X}_2^{\mathsf{T}})^{-1} \mathbf{X}_2 \boldsymbol{\varepsilon}_2 , \\
h_3 &= \xi^2 \boldsymbol{\gamma}^{\mathsf{T}} \boldsymbol{\Sigma}_2 (\mathbf{X}_1 \mathbf{X}_1^{\mathsf{T}})^{-1} \mathbf{X}_1 \boldsymbol{\varepsilon}_1 , \\
h_4 &= \xi (1 - \xi) \boldsymbol{\gamma}^{\mathsf{T}} \boldsymbol{\Sigma}_2 (\mathbf{X}_2 \mathbf{X}_2^{\mathsf{T}})^{-1} \mathbf{X}_2 \boldsymbol{\varepsilon}_2 , \\
h_5 &= \xi (1 - \xi) \boldsymbol{\varepsilon}_1^{\mathsf{T}} \mathbf{X}_1^{\mathsf{T}} (\mathbf{X}_1 \mathbf{X}_1^{\mathsf{T}})^{-1} \boldsymbol{\Sigma}_2 (\mathbf{X}_2 \mathbf{X}_2^{\mathsf{T}})^{-1} \mathbf{X}_2 \boldsymbol{\varepsilon}_2 .
\end{aligned}
$$

By Lemmas 4-5, we have with high probability,

$$
\begin{aligned}
\left| h_2 - (1 - \xi)^2 \sigma^2 \mathrm{Tr}(\mathbf{X}_2 \mathbf{X}_2^{\mathsf{T}})^{-1} \boldsymbol{\Sigma}_2 \right| &= \left| h_2 - (1 - \xi)^2 \sigma^2 \mathrm{Tr}(\mathbf{Z}_2 \mathbf{Z}_2)^{\mathsf{T}} \right| \\
&\prec (1 - \xi)^2 \sigma^2 \| (\mathbf{Z}_2 \mathbf{Z}_2^{\mathsf{T}})^{-1} \|_{\mathsf{F}} \\
&= (1 - \xi)^2 \sigma^2 \sqrt{ \sum_{i=1}^{M} \lambda_i^{-2} (\mathbf{Z}_2 \mathbf{Z}_2^{\mathsf{T}}) } \\
&\lesssim (1 - \xi)^2 \sigma^2 \frac{1}{\sqrt{M}} \mathrm{Tr}(\mathbf{Z}_2 \mathbf{Z}_2^{\mathsf{T}})^{-1} \\
&\lesssim \frac{1}{\sqrt{M}} .
\end{aligned} \tag{27}
$$

Lemma 6 implies that with high probability,

$$
\mathrm{Tr}(\mathbf{Z}_2 \mathbf{Z}_2^{\mathsf{T}})^{-1} = \frac{M}{N_2 - M} + o_{a.s.}(1).
$$

Combining this with equation 27, we obtain with high probability,

$$
h_2 = (1 - \xi)^2 \sigma^2 \frac{M}{N_2 - M} \big( 1 + o_{a.s.}(1) \big).
$$

Similarly, one may derive with high probability,

$$
\begin{aligned}
|h_5| &\prec \sigma^2 \| \mathbf{X}_1^{\mathsf{T}} (\mathbf{X}_1 \mathbf{X}_1^{\mathsf{T}})^{-1} \boldsymbol{\Sigma}_2 (\mathbf{X}_2 \mathbf{X}_2^{\mathsf{T}})^{-1} \mathbf{X}_2 \|_{\mathsf{F}} \\
&= \sigma^2 \sqrt{ \mathrm{Tr}(\mathbf{X}_1 \mathbf{X}_1^{\mathsf{T}})^{-1} \boldsymbol{\Sigma}_2 (\mathbf{X}_1 \mathbf{X}_1^{\mathsf{T}})^{-1} } \lesssim \frac{1}{\sqrt{M}} \widehat{\mathbf{Var}} .
\end{aligned}
$$

Using Lemmas 4-5, the following estimate holds with high probability,

$$
\begin{aligned}
\left| h_1 - \xi^2 \sigma^2 \mathrm{Tr}(\mathbf{X}_1 \mathbf{X}_1^{\mathsf{T}})^{-1} \boldsymbol{\Sigma}_2 \right| &\prec \| \boldsymbol{\Sigma}_2 (\mathbf{X}_1 \mathbf{X}_1^{\mathsf{T}})^{-1} \|_{\mathsf{F}} \\
&\lesssim \sqrt{ M \| \boldsymbol{\Sigma}_2 \|^2 \| \boldsymbol{\Sigma}_1 \|^{-2} \| (\mathbf{Z}_1 \mathbf{Z}_1^{\mathsf{T}})^{-1} \|^2 } \lesssim \frac{1}{\sqrt{M}} .
\end{aligned}
$$

Then by Lemma 4 and Lemma 6, one has with high probability,

$$
\begin{aligned}
\mathrm{Tr}(\mathbf{X}_1 \mathbf{X}_1^{\mathsf{T}})^{-1} \boldsymbol{\Sigma}_2 &= \frac{1}{N_1} \mathrm{Tr} \left( \frac{1}{N_1} \mathbf{Z}_1 \mathbf{Z}_1^{\mathsf{T}} \right)^{-1} \boldsymbol{\Sigma}_1^{-1/2} \boldsymbol{\Sigma}_2 \boldsymbol{\Sigma}_1^{-1/2} \\
&= \frac{1}{N_1} \frac{N_1}{N_1 - M} \mathrm{Tr} \boldsymbol{\Sigma}_2 \boldsymbol{\Sigma}_1^{-1} + o_{a.s.}(1).
\end{aligned}
$$

Therefore, for $\xi \neq 0$, we get with high probability,

$$\frac{1}{\xi^2 \sigma^2} h_1 = \text{Tr}(\mathbf{X}_1 \mathbf{X}_1^{\mathsf{T}})^{-1} \mathbf{\Sigma}_2 + o_{a.s.}(1) = \frac{1}{N_1 - M} \text{Tr} \mathbf{\Sigma}_2 \mathbf{\Sigma}_1^{-1} + o_{a.s.}(1).$$

We note that

$$\|\boldsymbol{\gamma}\|^2 \gtrsim \widehat{\mathbf{Bias}} = \xi^2 \|\mathbf{\Sigma}_2^{1/2} \boldsymbol{\gamma}\|^2 \gtrsim \lambda_{\min}(\mathbf{\Sigma}_2) \|\boldsymbol{\gamma}\|^2 \gtrsim \|\boldsymbol{\gamma}\|^2.$$

Since $\sigma_M(\mathbf{\Sigma}_1) \lesssim 1, \sigma_1(\mathbf{\Sigma}_2) \lesssim 1$, it is easy to see $\widehat{\mathbf{Var}} \sim 1$. Using Lemmas 4-5, we get with high probability

$$|h_3| \prec \xi^2 \sigma \|\boldsymbol{\gamma}^{\mathsf{T}} \mathbf{\Sigma}_2 (\mathbf{X}_1 \mathbf{X}_1^{\mathsf{T}})^{-1} \mathbf{X}_1\|$$

$$\leq \xi^2 \sigma \sqrt{\widehat{\mathbf{Bias}}} \|\mathbf{\Sigma}_2^{1/2}\| \|(\mathbf{X}_1 \mathbf{X}_1^{\mathsf{T}})^{-1} \mathbf{X}_1\|$$

$$\lesssim \frac{\sqrt{\widehat{\mathbf{Bias}}}}{M^{1/4}} \frac{1}{M^{1/4}} \leq \frac{\widehat{\mathbf{Bias}}}{\sqrt{M}} + \frac{1}{\sqrt{M}}$$

$$\lesssim \frac{1}{\sqrt{M}} (\widehat{\mathbf{Bias}} + \widehat{\mathbf{Var}}).$$

Similarly, we can estimate with high probability

$$|h_4| \prec \frac{1}{\sqrt{M}} (\widehat{\mathbf{Bias}} + \widehat{\mathbf{Var}}).$$

Combining the above estimates on $h_i, i = 1, 2, 3, 4, 5$, the proof of Corollary 3 is completed.

## B.5 PROOF OF THEOREM 3

For simplicity, we present the proof only for deterministic $\boldsymbol{\beta}_1$ and $\boldsymbol{\beta}_2$; the extension to the random case follows by similar reasoning and is therefore omitted. Denote $\mathbf{P}_{X_1}$ and $\mathbf{P}_{X_2}$ by

$$\mathbf{P}_{X_1} = (\mathbf{X}_1 \mathbf{X}_1^{\mathsf{T}})^+ \mathbf{X}_1 \mathbf{X}_1^{\mathsf{T}}, \quad \mathbf{P}_{X_2} = (\mathbf{X}_2 \mathbf{X}_2^{\mathsf{T}})^+ \mathbf{X}_2 \mathbf{X}_2^{\mathsf{T}}.$$

Note that for any rectangular matrix $\mathbf{A}$ and compatible $\mathbf{B}$,

$$(\mathbf{A} \mathbf{A}^{\mathsf{T}})^+ \mathbf{B} = \lim_{\lambda \to 0^+} (\mathbf{A} \mathbf{A}^{\mathsf{T}} + \lambda \mathbf{I}_M)^{-1} \mathbf{B}.$$

We can apply this to $\mathbf{A}_1 = \frac{1}{\sqrt{N_i}} \mathbf{X}_i$ for $i = 1, 2$ and rewrite the bias as

$$\mathbf{Bias} = \lim_{\lambda \to 0^+} f_M(\lambda),$$

where

$$f_M(\lambda) = \xi^2 \boldsymbol{\gamma}^{\mathsf{T}} \mathbf{\Sigma}_2 \boldsymbol{\gamma} + 2 \sum_{i=1}^{6} b_i + \sum_{i=1}^{4} h_{ii} + 2 \sum_{1 \leq i \neq j \leq 4} h_{ij},$$

and all terms on the right-hand side are given in Section B.2, under the setting $\lambda_{\mathsf{t}} = \lambda_{\mathsf{s}} = \lambda$. It is straightforward to see that $|f_M(\lambda)| \lesssim 1$. Now we consider $f_M'(\lambda)$. Let $\lambda_{\min}^+(\cdot)$ denote the smallest positive eigenvalue. Lemma 4 implies that for $i = 1, 2$,

$$\frac{1}{N_i} \lambda_{\max}(\mathbf{X}_i \mathbf{X}_i^{\mathsf{T}}) \leq 2\sigma_1^i \left(1 + \sqrt{\frac{M}{N_i}}\right)^2, \quad \frac{1}{N_i} \lambda_{\min}^+(\mathbf{X}_i \mathbf{X}_i^{\mathsf{T}}) \geq \frac{1}{2} \sigma_M^i \left(1 - \sqrt{\frac{M}{N_i}}\right)^2, \quad a.s.$$

Recall that $\|\boldsymbol{\beta}_1\|, \|\boldsymbol{\beta}_2\| \leq c$. Then, by equation 18, we have with high probability

$$\left|\frac{\mathrm{d}}{\mathrm{d}\lambda} h_{22}\right| = \left|\frac{\mathrm{d}}{\mathrm{d}\lambda} \xi^2 \lambda^2 \boldsymbol{\beta}_1^{\mathsf{T}} \mathbf{Q}_1 \mathbf{\Sigma}_2 \mathbf{Q}_1 \boldsymbol{\beta}_1\right|$$

$$= 2\xi^2 \left|\lambda \boldsymbol{\beta}_1^{\mathsf{T}} \mathbf{Q}_1^2 \frac{1}{N_1} \mathbf{X}_1 \mathbf{X}_1^{\mathsf{T}} \mathbf{\Sigma}_2 \mathbf{Q}_1 \boldsymbol{\beta}_1\right|$$

$$\leq 2\xi^2 \|\boldsymbol{\beta}_1\|^2 \|\lambda \mathbf{Q}_1\| \|\mathbf{\Sigma}_2\| \left\|\mathbf{Q}_1^2 \frac{1}{N_1} \mathbf{X}_1 \mathbf{X}_1^{\mathsf{T}}\right\|$$

$$\leq C_\xi \frac{\lambda_{\max}(\mathbf{X}_1 \mathbf{X}_1^{\mathsf{T}}/N_1)}{(\lambda_{\min}(\mathbf{X}_1 \mathbf{X}_1^{\mathsf{T}}/N_1) + \lambda)^2} \lesssim 1.$$

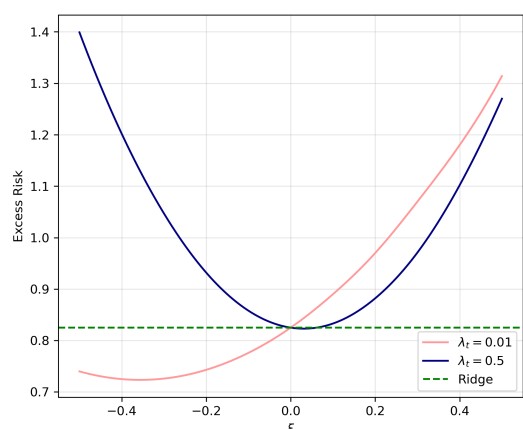

Figure 3: Theoretical excess risk for different $\lambda_t$. Settings: $(M, N_1, N_2) = (200, 200, 600)$, $\Sigma_1 = \Sigma_2 = \mathbf{I}_M$, $\lambda_s = 0.5$, SNR=4, $\beta_1 = \beta_2 \sim \mathcal{N}(0, \frac{4}{M})$, $\sigma^2 = 1$.

The remaining terms in $f'_M(\lambda)$ can be bounded in a similar manner, and hence $|f'_M(\lambda)| \lesssim 1$ almost surely. Therefore, $\{f_M(x)\}$ is equicontinuous almost surely. By the Arzela-Ascoli theorem, $f_M$ converges uniformly to its limit $f$, almost surely. By the Moore-Osgood theorem, we can exchange limits and get,

$$\lim_{M \to \infty} \lim_{\lambda \to 0^+} f_M(\lambda) = \lim_{\lambda \to 0^+} \lim_{M \to \infty} f_M(\lambda) = f(0^+), \quad a.s.$$

Similarly, letting $g_M(\lambda) = h_{55} + h_{66}$ with $h_{55}$ and $h_{66}$ as defined in Section B.2 under the setting $\lambda_t = \lambda_s = \lambda$, we get $g_M$ converges uniformly to its limit, $g$, and

$$\lim_{M \to \infty} \lim_{\lambda \to 0^+} g_M(\lambda) = \lim_{\lambda \to 0^+} \lim_{M \to \infty} g_M(\lambda) = g(0^+), \quad a.s.$$

By Theorem 1, $f = \widehat{\mathbf{Bias}}$ and $g = \widehat{\mathbf{Var}}$ under the setting $\lambda = \lambda_t = \lambda_s$. It is easy to verify that the remaining terms appearing in 19 are asymptotically negligible, and that $f, g$ are right-continuous. The proof is completed.

### B.6 PROOF OF COROLLARY 2

The Stieltjes transform of the Marchenko-Pastur distribution is given by

$$\underline{m}_i(z) = \int \frac{d\varrho_{\mathrm{MP},i}(x)}{x - z} = \frac{\left(1 - \frac{M}{N_i}\right) - z - \sqrt{\left(z - 1 - \frac{M}{N_i}\right)^2 - 4\frac{M}{N_i}}}{2\frac{M}{N_i}z}. \tag{28}$$

We take the derivative of $\mathbf{ER}(\beta_s)$ with respect to $\xi$, and evaluate it at $\xi = 0$:

$$\frac{\partial}{\partial \xi} \mathbf{ER}(\beta_s)\Big|_{\xi=0} = \sigma^2 \left(\lambda_t \lambda_s \underline{m}_1 \mathrm{SNR} - \frac{M}{N_2}\right) \frac{d}{dz}\left(z\underline{m}_2(z)\right)\Big|_{z=-\lambda_s} + o_{a.s.}(1).$$

Since

$$z\underline{m}_2(z) = \int \frac{z}{x - z} d\varrho_{\mathrm{MP},2}(x) = -1 + \int \frac{x}{x - z} d\varrho_{\mathrm{MP},2}(x),$$

we hence get that

$$\frac{d}{dz}\left(z\underline{m}_2(z)\right) > 0 \big|_{z=-\lambda_s}. \tag{29}$$

Therefore, $\frac{\partial}{\partial \xi} \mathbf{ER}(\beta_s)|_{\xi=0}$ and $\lambda_t \lambda_s \mathrm{SNR} - \frac{M}{N_2}$ share the same sign almost surely. That is, the limiting optimal value of $\xi$ is negative when equation 10 holds, which establishes Corollary 2.

We provide an example in Figure 3 to illustrate this corollary. Specifically, when $\lambda_t = 0.5$, the limiting optimal value of $\xi$ is positive, whereas when $\lambda_t = 0.01$, it becomes negative.

### B.7 DETAILS FOR PROPOSITION 1 AND REMARK 2

Recall that $\mathbf{ER}_0$ denotes the excess risk of the ridge regression model trained solely on the target domain data.

**Assumption A.1.** *When $\boldsymbol{\beta}_1, \boldsymbol{\beta}_2$ are deterministic, we assume that :*

$$
\Big| \boldsymbol{\beta}_1^{\mathsf{T}} \lambda_{\mathsf{t}} \lambda_{\mathsf{s}} \big[ \boldsymbol{\Pi}_1 \boldsymbol{\Sigma}_2 \boldsymbol{\Pi}_2 - \lambda_{\mathsf{s}} \boldsymbol{\Pi}_1 \boldsymbol{\Pi}_2 \mathcal{S}_2(\boldsymbol{\Sigma}_2) \boldsymbol{\Pi}_2 \big] \boldsymbol{\beta}_2 - \frac{\sigma^2}{N_2} \mathrm{Tr}[\boldsymbol{\Sigma}_2 (\boldsymbol{\Pi}_2 - \lambda_{\mathsf{s}} \boldsymbol{\Pi}_2')]
$$

$$
+ \boldsymbol{\beta}_2^{\mathsf{T}} [\lambda_{\mathsf{s}}^2 \boldsymbol{\Pi}_2 \mathcal{S}_2(\boldsymbol{\Sigma}_2) \boldsymbol{\Pi}_2 - \lambda_{\mathsf{s}} \boldsymbol{\Pi}_2 \boldsymbol{\Sigma}_2] \boldsymbol{\gamma} \Big| > c,
$$

*where $c$ is a positive constant.*

**Assumption A.2.** *If $\boldsymbol{\beta} = \boldsymbol{\beta}_1 = \boldsymbol{\beta}_2$ is random, we assume that*

$$
\Big| \frac{\tilde{\sigma}^2}{M} \Big[ \lambda_{\mathsf{t}} \lambda_{\mathsf{s}} \mathrm{Tr} \boldsymbol{\Pi}_1 \boldsymbol{\Pi}_2 \boldsymbol{\Sigma}_2 - \lambda_{\mathsf{t}} \lambda_{\mathsf{s}}^2 \mathrm{Tr} \boldsymbol{\Pi}_2 \mathcal{S}_2(\boldsymbol{\Sigma}_2) \boldsymbol{\Pi}_2 \boldsymbol{\Pi}_1 \Big] - \frac{\sigma^2}{N_2} \mathrm{Tr}[\boldsymbol{\Sigma}_2 (\boldsymbol{\Pi}_2 - \lambda_{\mathsf{s}} \boldsymbol{\Pi}_2')] \Big| > c,
$$

*where $c$ is a positive constant.*

**Proof of Proposition 1:** (i) Suppose the conditions in Theorem 1 hold. Note that

$$
\frac{\partial}{\partial \xi} \mathbf{ER}(\boldsymbol{\beta}_{\mathsf{s}}) \big|_{\xi=0} = 2 \boldsymbol{\beta}_1^{\mathsf{T}} \big[ \lambda_{\mathsf{t}} \lambda_{\mathsf{s}} \boldsymbol{\Pi}_1 \boldsymbol{\Sigma}_2 \boldsymbol{\Pi}_2 - \lambda_{\mathsf{t}} \lambda_{\mathsf{s}}^2 \boldsymbol{\Pi}_1 \boldsymbol{\Pi}_2 \mathcal{S}_2(\boldsymbol{\Sigma}_2) \boldsymbol{\Pi}_2 \big] \boldsymbol{\beta}_2
$$

$$
+ 2 \boldsymbol{\beta}_2^{\mathsf{T}} [\lambda_{\mathsf{s}}^2 \boldsymbol{\Pi}_2 \mathcal{S}_2(\boldsymbol{\Sigma}_2) \boldsymbol{\Pi}_2 - \lambda_{\mathsf{s}} \boldsymbol{\Pi}_2 \boldsymbol{\Sigma}_2] \boldsymbol{\gamma} - \frac{2\sigma^2}{N_2} \mathrm{Tr}[\boldsymbol{\Sigma}_2 (\boldsymbol{\Pi}_2 - \lambda_{\mathsf{s}} \boldsymbol{\Pi}_2')] + o_{a.s.}(1).
$$

Under the conditions in Theorem 1 and Assumption A.1, the asymptotic excess risk is a quadratic function whose minimizer is bounded away from 0. Therefore, $\min_\xi \mathbf{ER}(\boldsymbol{\beta}_{\mathsf{s}})$ is strictly less than $\mathbf{ER}_0$ almost surely.

(ii) Similarly, suppose Assumption A.2, under the conditions of Theorem 2, the inequality equation 13 holds by noticing that

$$
\frac{\partial}{\partial \xi} \mathbf{ER}(\boldsymbol{\beta}_{\mathsf{s}}) \big|_{\xi=0} = \frac{2\tilde{\sigma}^2}{M} \Big[ \lambda_{\mathsf{t}} \lambda_{\mathsf{s}} \mathrm{Tr} \boldsymbol{\Pi}_1 \boldsymbol{\Pi}_2 \boldsymbol{\Sigma}_2 - \lambda_{\mathsf{t}} \lambda_{\mathsf{s}}^2 \mathrm{Tr} \boldsymbol{\Pi}_2 \mathcal{S}_2(\boldsymbol{\Sigma}_2) \boldsymbol{\Pi}_2 \boldsymbol{\Pi}_1 \Big] - \frac{2\sigma^2}{N_2} \mathrm{Tr}[\boldsymbol{\Sigma}_2 (\boldsymbol{\Pi}_2 - \lambda_{\mathsf{s}} \boldsymbol{\Pi}_2')]
$$

$$
+ o_{a.s.}(1).
$$

**Further discussion on Remark 2**: To clarify the dependence of Assumption A.1 on the geometry of $\boldsymbol{\Sigma}_1, \boldsymbol{\Sigma}_2, \boldsymbol{\beta}_1, \boldsymbol{\beta}_2$ and the noise strength $\sigma^2$, we consider a simple example in which $\boldsymbol{\Sigma}_2 = \mathbf{I}_M$. Then we have

$$
\frac{\partial}{\partial \xi} \mathbf{ER}(\boldsymbol{\beta}_{\mathsf{s}}) \big|_{\xi=0} = \lambda_{\mathsf{s}} \lambda_{\mathsf{t}} (\underline{m}_2 - \lambda_{\mathsf{s}} \underline{m}_2') \boldsymbol{\beta}_1^{\mathsf{T}} \boldsymbol{\Pi}_1 \boldsymbol{\beta}_2 - \frac{\sigma^2 M}{N_2} (\underline{m}_2 - \lambda_{\mathsf{s}} \underline{m}_2') - \lambda_{\mathsf{s}} (\underline{m}_2 - \lambda_{\mathsf{s}} \underline{m}_2') \boldsymbol{\beta}_2^{\mathsf{T}} \boldsymbol{\gamma}
$$

$$
+ o_{a.s.}(1)
$$

$$
= \big( \underbrace{\lambda_{\mathsf{s}} \lambda_{\mathsf{t}} \boldsymbol{\beta}_1^{\mathsf{T}} \boldsymbol{\Pi}_1 \boldsymbol{\beta}_2 - \frac{\sigma^2 M}{N_2} - \lambda_{\mathsf{s}} \boldsymbol{\beta}_2^{\mathsf{T}} \boldsymbol{\gamma}}_{e} \big) \frac{\mathrm{d}}{\mathrm{d}z} (z \underline{m}_2(z)) \big|_{z=-\lambda_{\mathsf{s}}} + o_{a.s.}(1),
$$

(30)

where $\underline{m}_2(z)$ is defined in equation 28. Recalling equation 29, we have $|\frac{\partial}{\partial \xi} \mathbf{ER}(\boldsymbol{\beta}_{\mathsf{s}})|_{\xi=0}| > c$ if $|e| > C$ for some constant $C$. Below, we discuss two cases, when $\boldsymbol{\Sigma}_1 = \mathbf{I}_M$ and when $\boldsymbol{\Sigma}_1 \neq \mathbf{I}_M$:

- $\boldsymbol{\Sigma}_1 = \mathbf{I}_M$. The term $e$ becomes

$$
e = \lambda_{\mathsf{s}} \lambda_{\mathsf{t}} \underline{m}_1 \boldsymbol{\beta}_1^{\mathsf{T}} \boldsymbol{\beta}_2 - \frac{\sigma^2 M}{N_2} - \lambda_{\mathsf{s}} \boldsymbol{\beta}_2^{\mathsf{T}} (\boldsymbol{\beta}_1 - \boldsymbol{\beta}_2).
$$

  Recall that the limiting ridge risk is minimized at $\lambda_{\mathsf{s}}^* = \frac{\sigma^2 M}{N_2 \|\boldsymbol{\beta}_2\|^2}$, with asymptotic excess risk $\sigma^2 \frac{M}{N_2} \underline{m}_2(-\lambda_{\mathsf{s}}^*)$ (Hastie et al., 2022). Taking $\lambda_{\mathsf{s}} = \lambda_{\mathsf{s}}^*$, we have

$$
e = \lambda_{\mathsf{s}}^* (\lambda_{\mathsf{t}} \underline{m}_1 - 1) \boldsymbol{\beta}_1^{\mathsf{T}} \boldsymbol{\beta}_2.
$$

Note that

$$\lambda_t \underline{m}_1 - 1 = -\int \frac{x}{x + \lambda_t} \mathrm{d}\varrho_{\mathsf{MP},1}(x) < 0.$$

Therefore, in a small neighborhood containing 0, $\mathbf{ER}(\boldsymbol{\beta}_\mathsf{s})$ is monotonic in $\xi$, indicating that the teacher's supervision is helpful – even outperforming the optimal ridge regression – provided that $\boldsymbol{\beta}_1$ and $\boldsymbol{\beta}_2$ are not asymptotically orthogonal.

- $\boldsymbol{\Sigma}_1 \neq \mathbf{I}_M$. By taking $\lambda_\mathsf{s} = \lambda_\mathsf{s}^*$, $e$ becomes

$$e = \lambda_\mathsf{s}^*(\lambda_t \boldsymbol{\beta}_1^\mathsf{T} \boldsymbol{\Pi}_1 \boldsymbol{\beta}_2 - \boldsymbol{\beta}_2^\mathsf{T} \boldsymbol{\beta}_1) = -\lambda_\mathsf{s}^* \sum_{i=1}^M \frac{m_1 \sigma_i}{1 + m_1 \sigma_i} \boldsymbol{\beta}_1^\mathsf{T} \mathbf{u}_i \mathbf{u}_i^\mathsf{T} \boldsymbol{\beta}_2, \tag{31}$$

where $m_1$ is determined by Lemma 2 and $\boldsymbol{\Sigma}_1 = \sum_{i=1}^M \sigma_i \mathbf{u}_i \mathbf{u}_i^\mathsf{T}$ represents the spectral decomposition of $\boldsymbol{\Sigma}_1$. By equation 31, the alignment of $\boldsymbol{\beta}_i$ ($i = 1, 2$) with the eigenvectors of $\boldsymbol{\Sigma}_1$ determines whether Assumption A.1 is satisfied. Therefore, given $\lambda_\mathsf{s} = \lambda_\mathsf{s}^*$, under the "help" of covariate shift, even if $\boldsymbol{\beta}_1^\mathsf{T} \boldsymbol{\beta}_2 = 0$, it may still be possible to find a $\xi$ such that $\mathbf{ER}(\boldsymbol{\beta}_\mathsf{s}) < \mathbf{ER}_0$, a.s. By comparing with the case where $\boldsymbol{\Sigma}_1 = \mathbf{I}_M$, we find that the presence of covariate shift can, in some cases, be beneficial.

Note that, up to asymptotically negligible terms, $\mathbf{ER}$ can be expressed as a quadratic function of $\xi : \mathbf{ER}(\boldsymbol{\beta}_\mathsf{s}, \xi) = A\xi^2 + B\xi + C$. Below we provide closed-form expressions for the asymptotic optimal $\xi^* = -\frac{B}{2A}$ under several common settings.

(1) When $\boldsymbol{\gamma} = \boldsymbol{\beta}_1 - \boldsymbol{\beta}_2$, $\frac{M}{N_1}, \frac{M}{N_2} < (1 + \tau)^{-1}$,

$$\xi^* = \left( \boldsymbol{\gamma}^\mathsf{T} \boldsymbol{\Sigma}_2 \boldsymbol{\gamma} + \sigma^2 \frac{M}{N_2 - M} + \frac{\sigma^2}{N_1 - M} \mathrm{Tr} \boldsymbol{\Sigma}_2 \boldsymbol{\Sigma}_1^{-1} \right)^{-1} \frac{\sigma^2 M}{N_2 - M} \in (0, 1).$$

(2) When $\boldsymbol{\beta} = \boldsymbol{\beta}_1 = \boldsymbol{\beta}_2$ is random, and $\boldsymbol{\Sigma}_1 = \boldsymbol{\Sigma}_2 = \mathbf{I}_M$,

$$\xi^* = \frac{\frac{M}{N_2}(m_2 - \lambda_s m_2') - \frac{\tilde{\sigma}^2}{\sigma^2} \lambda_t \lambda_s (m_1 m_2 - \lambda_s m_2' m_1)}{A_1 + A_2 + A_3},$$

where

$$A_1 = \frac{\tilde{\sigma}^2}{\sigma^2} \left( \lambda_t^2 \underline{m}_1' - 2\lambda_t^2 \lambda_s \underline{m}_2 \underline{m}_1' + \lambda_t^2 \lambda_s^2 \underline{m}_1' \underline{m}_2' \right),$$

$$A_2 = \frac{M}{N_1} \left( \underline{m}_1 - 2\lambda_s \underline{m}_1 \underline{m}_2 + \lambda_s^2 \underline{m}_1 \underline{m}_2' - \lambda_t \underline{m}_1' + 2\lambda_t \lambda_s \underline{m}_2 \underline{m}_1' - \lambda_t \lambda_s^2 \underline{m}_1' \underline{m}_2' \right),$$

$$A_3 = \frac{M}{N_2} \left( \underline{m}_2 - \lambda_s \underline{m}_2' \right).$$

(3) When $\boldsymbol{\beta} = \boldsymbol{\beta}_1 = \boldsymbol{\beta}_2$, $\boldsymbol{\Sigma}_2 = \mathbf{I}_M$,

$$\xi^* = \frac{\boldsymbol{\beta}^\mathsf{T}[\lambda_t \lambda_\mathsf{s}^2 \underline{m}_2' \boldsymbol{\Pi}_1 - \lambda_t \lambda_\mathsf{s} \underline{m}_2 \boldsymbol{\Pi}_1] \boldsymbol{\beta} + \frac{\sigma^2 M}{N_2}(\underline{m}_2' - \lambda_\mathsf{s} \underline{m}_2')}{\boldsymbol{\beta}^\mathsf{T} \lambda_t^2 [1 + \lambda_\mathsf{s}^2 \underline{m}_2' - 2\lambda_\mathsf{s} \underline{m}_2] \boldsymbol{\Pi}_1' \boldsymbol{\beta} + \frac{\sigma^2(1 - 2\lambda_\mathsf{s} \underline{m}_2 + \lambda_\mathsf{s}^2 \underline{m}_2')}{N_1} \mathrm{Tr}[\boldsymbol{\Pi}_1 - \lambda_t \boldsymbol{\Pi}_1'] + \frac{\sigma^2 M}{N_2}(\underline{m}_2 - \lambda_\mathsf{s} \underline{m}_2')}.$$

### B.8 Proof of Lemma 7

The following result, which is an immediate consequence of Lemma 2, will be used in the proof below:

$$-zm = \left( 1 + \frac{1}{N} \mathrm{Tr} \boldsymbol{\Sigma} \boldsymbol{\Pi}(z) \right)^{-1}. \tag{32}$$

We abuse notation by writing $z_1$ and $z_2$ for $\tilde{z}_1$ and $\tilde{z}_2$, respectively, whenever there is no risk of ambiguity. Without loss of generality, we assume $\|\mathbf{u}\| = \|\mathbf{v}\| = 1$ and $z_1, z_2$ lie on the negative real axis, as the other cases follow by analogous arguments.

Using standard techniques of martingale decomposition (see, e.g., Bai & Silverstein (2010)), we can prove the almost sure convergence of the random part:

$$\mathbf{u}^\mathsf{T}\mathbf{Q}(z_1)\mathbf{A}\mathbf{Q}(z_2)\mathbf{v} = \mathbf{u}^\mathsf{T}\mathbb{E}\mathbf{Q}(z_1)\mathbf{A}\mathbf{Q}(z_2)\mathbf{v} + o_{a.s.}(1). \tag{33}$$

Therefore, it suffices to consider the term $\mathbf{u}^\mathsf{T}\mathbb{E}\mathbf{Q}(z_1)\mathbf{A}\mathbf{Q}(z_2)\mathbf{v}$. Let $\sigma_1 \geq \cdots \geq \sigma_M$ denote the eigenvalues of $\mathbf{\Sigma}$. For the sequence of deterministic matrices, we denote $\mathbf{A}_M = o(1)$ if $\|\mathbf{A}_M\| \to 0$. Since

$$\mathbf{Q}(z_1)\mathbf{A}\mathbf{Q}(z_2) = \mathbf{Q}(z_1)\mathbf{A}\mathbf{\Pi}(z_2) + \mathbf{Q}(z_1)\mathbf{A}\big(\mathbf{Q}(z_2) - \mathbf{\Pi}(z_2)\big), \tag{34}$$

we obtain by Lemma 6 that

$$\begin{aligned}
\mathbf{u}^\mathsf{T}\mathbb{E}\mathbf{Q}(z_1)\mathbf{A}\mathbf{Q}(z_2)\mathbf{v} &= \mathbf{u}^\mathsf{T}\mathbb{E}\mathbf{Q}(z_1)\mathbf{A}\mathbf{\Pi}(z_2)\mathbf{v} + \mathbf{u}^\mathsf{T}\mathbb{E}\mathbf{Q}(z_1)\mathbf{A}(\mathbf{Q}(z_2) - \mathbf{\Pi}(z_2))\mathbf{v} \\
&= \mathbf{u}^\mathsf{T}\mathbf{\Pi}(z_1)\mathbf{A}\mathbf{\Pi}(z_2)\mathbf{v} + \mathbf{u}^\mathsf{T}\mathbb{E}\mathbf{Q}(z_1)\mathbf{A}(\mathbf{Q}(z_2) - \mathbf{\Pi}(z_2))\mathbf{v} + o(1),
\end{aligned} \tag{35}$$

where the second identity follows from Lemma 6, the Dominated Convergence Theorem and the fact that

$$\|\mathbf{\Pi}(z_1)\| = \max_i \big|z_1 + z_1 m(z_1)\sigma_i\big|^{-1} \leq |z_1|^{-1}, \ \|\mathbf{A}\mathbf{\Pi}(z_2)\mathbf{v}\| \leq \|\mathbf{A}\|\|\mathbf{\Pi}(z_2)\| \leq |z_2|^{-1}\|\mathbf{A}\|.$$

Therefore, our task reduces to finding the deterministic equivalent of

$$\mathbb{E}\mathbf{Q}(z_1)\mathbf{A}\big(\mathbf{Q}(z_2) - \mathbf{\Pi}(z_2)\big).$$

Denote

$$\mathbf{X}_{-k} = \mathbf{X} - \mathbf{x}_k\mathbf{e}_k^\mathsf{T}, \ \mathbf{Q}_{-k}(z) = \left(\frac{\mathbf{X}_{-k}\mathbf{X}_{-k}^\mathsf{T}}{N} - z\mathbf{I}_M\right)^{-1}.$$

By Sherman-Morrison formula, one may easily check that

$$\begin{aligned}
\mathbf{Q}(z) &= \mathbf{Q}_{-k}(z) - \frac{\frac{1}{N}\mathbf{Q}_{-k}(z)\mathbf{x}_k\mathbf{x}_k^\mathsf{T}\mathbf{Q}_{-k}(z)}{1 + \frac{1}{N}\mathbf{x}_k^\mathsf{T}\mathbf{Q}_{-k}(z)\mathbf{x}_k}, \\
\mathbf{Q}(z)\mathbf{x}_k &= \frac{\mathbf{Q}_{-k}(z)\mathbf{x}_k}{1 + \frac{1}{N}\mathbf{x}_k^\mathsf{T}\mathbf{Q}_{-k}(z)\mathbf{x}_k}.
\end{aligned} \tag{36}$$

We show here the following result for future use:

$$\frac{1}{N}\mathbb{E}\mathrm{Tr}\mathbf{C}\mathbf{Q}_{-1}(z_1)\mathbf{A}\mathbf{Q}_{-1}(z_2) = \frac{1}{N}\mathbb{E}\mathrm{Tr}\mathbf{C}\mathbf{Q}(z_1)\mathbf{A}\mathbf{Q}(z_2) + o(1), \tag{37}$$

where $\mathbf{C} \in \mathbb{R}^{M \times M}$ is a deterministic matrix with $\|\mathbf{C}\| \leq C$ for some constant $C$. We decompose

$$\begin{aligned}
&\mathbf{Q}(z_1)\mathbf{A}\mathbf{Q}(z_2) - \mathbf{Q}_{-1}(z_1)\mathbf{A}\mathbf{Q}_{-1}(z_2) \\
=&[\mathbf{Q}(z_1) - \mathbf{Q}_{-1}(z_1)]\mathbf{A}\mathbf{Q}(z_2) + \mathbf{Q}_{-1}(z_1)\mathbf{A}[\mathbf{Q}_{-1}(z_2) - \mathbf{Q}_{-1}(z_2)].
\end{aligned}$$

Applying the identity

$$\mathbf{A}^{-1} - \mathbf{B}^{-1} = \mathbf{B}^{-1}(\mathbf{B} - \mathbf{A})\mathbf{A}^{-1}, \tag{38}$$

we have for $i = 1, 2$, and $\widetilde{\mathbf{C}} \in \mathbb{R}^{M \times M}$ with finite spectral norm (where $\widetilde{\mathbf{C}}$ may be a deterministic matrix, or a random matrix that is either dependent on or independent of $\mathbf{X}$),

$$\frac{1}{N}|\mathrm{Tr}[\mathbf{Q}(z_i) - \mathbf{Q}_{-1}(z_i)]\widetilde{\mathbf{C}}| = \frac{1}{N^2}|\mathbf{x}_1^\mathsf{T}\mathbf{Q}(z_i)\widetilde{\mathbf{C}}\mathbf{Q}_{-1}(z_1)\mathbf{x}_1| \leq \frac{C}{N^2}\|\mathbf{x}_1\|^2 = o_{a.s.}(1).$$

We denote $d = \min\{\mathrm{dist}(z_1, \mathbb{R}^+), \mathrm{dist}(z_2, \mathbb{R}^+)\}$. One may easily check that $d \sim 1$. Then by

$$\frac{1}{N}|\mathrm{Tr}[\mathbf{Q}(z_i) - \mathbf{Q}_{-1}(z_i)]\widetilde{\mathbf{C}}| \leq \frac{M}{N}(\|\mathbf{Q}(z_i)\widetilde{\mathbf{C}}\| + \|\mathbf{Q}_{-1}(z_i)\widetilde{\mathbf{C}}\|) \leq \frac{2M}{dN}, \text{ for } i = 1, 2,$$

and the Dominated Convergence Theorem, we obtain equation 37. By similar arguments, we get for any deterministic unit vectors $\mathbf{u}, \mathbf{v}$,

$$\begin{aligned}
\mathbf{u}^\mathsf{T}\mathbb{E}\widetilde{\mathbf{C}}\mathbf{Q}(z_i)\mathbf{C}\mathbf{v} &= \mathbf{u}^\mathsf{T}\mathbb{E}\widetilde{\mathbf{C}}\mathbf{Q}_{-k}(z_i)\mathbf{C}\mathbf{v} + o(1) \\
&= \mathbf{u}^\mathsf{T}\widetilde{\mathbf{C}}\mathbf{\Pi}(z_i)\mathbf{C}\mathbf{v} + o(1), \ i = 1, 2,
\end{aligned} \tag{39}$$

$$\mathbf{u}^\mathsf{T}\mathbb{E}\widetilde{\mathbf{C}}\mathbf{Q}(z_1)\mathbf{A}\mathbf{Q}(z_2)\mathbf{C}\mathbf{v} = \mathbf{u}^\mathsf{T}\mathbb{E}\widetilde{\mathbf{C}}\mathbf{Q}_{-k}(z_1)\mathbf{A}\mathbf{Q}_{-k}(z_2)\mathbf{C}\mathbf{v} + o(1),$$

where $\widetilde{\mathbf{C}}$ and $\mathbf{C}$ are deterministic $M \times M$ matrices with finite spectral norms.

We denote
$$b_k = \frac{1}{N}\mathbf{x}_k^\mathsf{T}\mathbf{Q}_{-k}(z_2)\mathbf{x}_k, \; \tilde{b} = \frac{1}{N}\mathbb{E}\mathbf{x}_k^\mathsf{T}\mathbf{Q}_{-k}(z_2)\mathbf{x}_k,$$

$$\mathsf{b}_k = \frac{1}{N}\mathbf{x}_k^\mathsf{T}\mathbf{Q}_{-k}(z_1)\mathbf{x}_k, \; \tilde{\mathsf{b}} = \frac{1}{N}\mathbb{E}\mathbf{x}_k^\mathsf{T}\mathbf{Q}_{-k}(z_1)\mathbf{x}_k.$$

It follows directly from the proof of equation 37 that
$$\tilde{b} = \frac{1}{N}\mathbb{E}\mathrm{Tr}\boldsymbol{\Sigma}\mathbf{Q}_{-1}(z_2) = \frac{1}{N}\mathrm{Tr}\boldsymbol{\Sigma}\boldsymbol{\Pi}(z_2) + o(1),$$

$$\tilde{\mathsf{b}} = \frac{1}{N}\mathbb{E}\mathrm{Tr}\boldsymbol{\Sigma}\mathbf{Q}_{-1}(z_2) = \frac{1}{N}\mathrm{Tr}\boldsymbol{\Sigma}\boldsymbol{\Pi}(z_2) + o(1). \tag{40}$$

Recalling equation 32, we rewrite $\mathbb{E}\mathbf{Q}(z_1)\mathbf{A}\big(\mathbf{Q}(z_2) - \boldsymbol{\Pi}(z_2)\big)$ as
$$\begin{aligned}
\mathbb{E}\mathbf{Q}(z_1)\mathbf{A}\big(\mathbf{Q}(z_2) - \boldsymbol{\Pi}(z_2)\big) &= \mathbb{E}\mathbf{Q}(z_1)\mathbf{A}\mathbf{Q}(z_2)\big(\mathbf{I}_M - \mathbf{Q}^{-1}(z_2)\boldsymbol{\Pi}(z_2)\big)\\
&= \mathbb{E}\mathbf{Q}(z_1)\mathbf{A}\mathbf{Q}(z_2)\big(\boldsymbol{\Pi}^{-1}(z_2) - \mathbf{Q}^{-1}(z_2)\big)\boldsymbol{\Pi}(z_2)\\
&= \mathbb{E}\mathbf{Q}(z_1)\mathbf{A}\mathbf{Q}(z_2)\Big(-\frac{1}{N}\mathbf{X}\mathbf{X}^\mathsf{T} - z_2 m\boldsymbol{\Sigma}\Big)\boldsymbol{\Pi}(z_2)\\
&= \mathbb{E}\mathbf{Q}(z_1)\mathbf{A}\mathbf{Q}(z_2)\frac{\boldsymbol{\Sigma}\boldsymbol{\Pi}(z_2)}{1 + \frac{1}{N}\mathrm{Tr}\boldsymbol{\Sigma}\boldsymbol{\Pi}(z_2)} - \frac{1}{N}\mathbb{E}\mathbf{Q}(z_1)\mathbf{A}\mathbf{Q}(z_2)\mathbf{X}\mathbf{X}^\mathsf{T}\boldsymbol{\Pi}(z_2).
\end{aligned} \tag{41}$$

An application of equation 36 yields that
$$\begin{aligned}
\frac{1}{N}\sum_{k=1}^N \mathbb{E}\mathbf{Q}(z_1)\mathbf{A}\mathbf{Q}(z_2)\mathbf{x}_k\mathbf{x}_k^\mathsf{T} &= \frac{1}{N}\sum_{k=1}^N \mathbb{E}\mathbf{Q}(z_1)\mathbf{A}\frac{\mathbf{Q}_{-k}(z_2)\mathbf{x}_k\mathbf{x}_k^\mathsf{T}}{1 + b_k}\\
&= \frac{1}{N(1+\tilde{b})}\sum_{k=1}^N \mathbb{E}\mathbf{Q}(z_1)\mathbf{A}\mathbf{Q}_{-k}(z_2)\mathbf{x}_k\mathbf{x}_k^\mathsf{T}\Big[1 + \frac{\tilde{b} - b_k}{(1+b_k)}\Big]\\
&= \frac{1}{N(1+\tilde{b})}\Big[\sum_{k=1}^N \mathbb{E}\mathbf{Q}(z_1)\mathbf{A}\mathbf{Q}_{-k}(z_2)\mathbf{x}_k\mathbf{x}_k^\mathsf{T} + \mathbb{E}\mathbf{Q}(z_1)\mathbf{A}\mathbf{Q}(z_2)\mathbf{X}\mathbf{B}\mathbf{X}^\mathsf{T}\Big]\\
&= \frac{1}{1+\tilde{b}}\big(\mathbb{E}\mathbf{F}_1 + \mathbb{E}\mathbf{F}_2\big),
\end{aligned} \tag{42}$$

where $\mathbf{B} = \mathrm{diag}\big(\tilde{b} - b_1, ..., \tilde{b} - b_N\big)$, and
$$\mathbf{F}_1 = \frac{1}{N}\sum_{k=1}^N \mathbf{Q}(z_1)\mathbf{A}\mathbf{Q}_{-k}(z_2)\mathbf{x}_k\mathbf{x}_k^\mathsf{T}, \; \mathbf{F}_2 = \frac{1}{N}\mathbf{Q}(z_1)\mathbf{A}\mathbf{Q}(z_2)\mathbf{X}\mathbf{B}\mathbf{X}^\mathsf{T}.$$

We now bound the spectral norm of $\mathbf{F}_2$. Define the event
$$\mathcal{E} = \Big\{\frac{1}{N}\|\mathbf{Z}\mathbf{Z}^\mathsf{T}\| \le 2\big(1 + \sqrt{\tfrac{M}{N}}\big)^2\Big\}.$$

We then have
$$\begin{aligned}
\|\mathbb{E}\mathbf{F}_2\| \le \mathbb{E}\|\mathbf{F}_2\| &\le \frac{1}{N}\frac{\|\mathbf{A}\|}{d^2}\mathbb{E}\|\mathbf{X}\mathbf{B}\mathbf{X}^\mathsf{T}\|\\
&\le \frac{\|\mathbf{A}\|\|\boldsymbol{\Sigma}\|}{d^2}\Big[4(1+\sqrt{\phi})^2\mathbb{E}\|\mathbf{B}\|\delta(\mathcal{E}) + \frac{1}{N}\mathbb{E}\|\mathbf{B}\|\|\mathbf{Z}\mathbf{Z}^\mathsf{T}\|\delta(\mathcal{E}^C)\Big]\\
&\le C\mathbb{E}\max_k|\tilde{b} - b_k| + \frac{1}{N}\sqrt{\mathbb{E}\max_k|\tilde{b} - b_k|^2\mathbb{E}\|\mathbf{Z}\mathbf{Z}^\mathsf{T}\|^2\delta(\mathcal{E}^C)}.
\end{aligned}$$

By using the inequality that (see e.g. Bai & Silverstein (2010))
$$\mathbb{P}(\mathcal{E}^C) \le N^{-\ell} \text{ for any } \ell > 0,$$

we have

$$\mathbb{E}\|\mathbf{Z}\mathbf{Z}^\mathsf{T}\|^2\delta(\mathcal{E}^C) \leq \mathbb{E}\|\mathbf{Z}\mathbf{Z}^\mathsf{T}\|_\mathsf{F}^2\delta(\mathcal{E}^C) \leq \sqrt{\mathbb{E}\|\mathbf{Z}\mathbf{Z}^\mathsf{T}\|_\mathsf{F}^4 \mathbb{P}(\mathcal{E}^C)}$$
$$\leq N^{100}o(N^{-101}) = o(N^{-1}). \tag{43}$$

It can be shown by Lemma 3 that for $\ell \geq 1$,

$$\mathbb{P}(|\tilde{b} - b_k| > t) \leq \frac{\mathbb{E}|\mathbf{z}_k^\mathsf{T}\mathbf{\Sigma}^{1/2}\mathbf{Q}_{-k}\mathbf{\Sigma}^{1/2}\mathbf{z}_k - \mathrm{Tr}\mathbf{\Sigma}\mathbf{Q}_{-k}(z)|^\ell}{(Nt)^\ell}$$

$$= t^{-\ell}\frac{\mathbb{E}[\mathbb{E}_{-k}|\mathbf{z}_k^\mathsf{T}\mathbf{\Sigma}^{1/2}\mathbf{Q}_{-k}\mathbf{\Sigma}^{1/2}\mathbf{z}_k - \mathrm{Tr}\mathbf{\Sigma}\mathbf{Q}_{-k}(z)|^\ell]}{N^\ell}$$

$$\leq t^{-\ell}C\frac{\mathbb{E}[(\mathrm{Tr}\mathbf{Q}_{-k}^2)^{\ell/2} + \mathrm{Tr}(\mathbf{Q}_{-k})^\ell]}{N^\ell}$$

$$\leq Ct^{-\ell}N^{-\ell/2},$$

where we use the fact that

$$\mathrm{Tr}\big(\mathbf{Q}_{-k}(z_2)\big)^\ell \leq M\|\mathbf{Q}_{-k}(z_2)\|^\ell \leq \frac{M}{d^\ell}.$$

By taking a large enough $\ell$, we have

$$\mathbb{E}\max_k|\tilde{b} - b_k| = \left(\int_{t \leq N^{-1/4}} + \int_{t > N^{-1/4}}\right)\mathbb{P}(\max_k|\tilde{b} - b_k| > t)\mathrm{d}t$$

$$\leq N^{-1/4} + \int_{t > N^{-1/4}}\sum_{k=1}^N \mathbb{P}(|\tilde{b} - b_k| > t)\mathrm{d}t \tag{44}$$

$$\leq 2N^{-1/4}.$$

Similarly, one may obtain

$$\mathbb{E}\max_k|\tilde{b} - b_k|^2 = o(1). \tag{45}$$

This, along with equation 43 and equation 44, implies that

$$\|\mathbb{E}\mathbf{F}_2\| = o(1).$$

By using equation 36, we rewrite

$$\mathbb{E}\mathbf{F}_1 = \frac{1}{N}\sum_{k=1}^N \mathbb{E}\mathbf{Q}(z_1)\mathbf{A}\mathbf{Q}_{-k}(z_2)\mathbf{x}_k\mathbf{x}_k^\mathsf{T}$$

$$= \frac{1}{N}\sum_{k=1}^N \mathbb{E}\big[\mathbf{Q}_{-k}(z_1) - \frac{1}{N}\frac{\mathbf{Q}_{-k}(z_1)\mathbf{x}_k\mathbf{x}_k^\mathsf{T}\mathbf{Q}_{-k}(z_1)}{1 + \mathsf{b}_k}\big]\mathbf{A}\mathbf{Q}_{-k}(z_2)\mathbf{x}_k\mathbf{x}_k^\mathsf{T}$$

$$= \frac{1}{N}\sum_{k=1}^N \mathbb{E}\mathbf{Q}_{-k}(z_1)\mathbf{A}\mathbf{Q}_{-k}(z_2)\mathbf{x}_k\mathbf{x}_k^\mathsf{T} - \frac{1}{N}\sum_{k=1}^N \mathbb{E}\frac{\frac{1}{N}\mathbf{Q}_{-k}(z_1)\mathbf{x}_k\mathbf{x}_k^\mathsf{T}\mathbf{Q}_{-k}(z_1)}{1 + \mathsf{b}_k}\mathbf{A}\mathbf{Q}_{-k}(z_2)\mathbf{x}_k\mathbf{x}_k^\mathsf{T}$$

$$= \mathbb{E}\mathbf{Q}_{-1}(z_1)\mathbf{A}\mathbf{Q}_{-1}(z_2)\mathbf{\Sigma} - \frac{1}{(1+\tilde{\mathsf{b}})N}\sum_{k=1}^N \mathbb{E}\frac{1}{N}\mathbf{Q}_{-k}(z_1)\mathbf{x}_k\mathbf{x}_k^\mathsf{T}\mathbf{Q}_{-k}(z_1)\mathbf{A}\mathbf{Q}_{-k}(z_2)\mathbf{x}_k\mathbf{x}_k^\mathsf{T}$$

$$- \frac{1}{(1+\tilde{\mathsf{b}})N^2}\sum_{k=1}^N \mathbb{E}\mathbf{Q}_{-k}(z_1)\mathbf{x}_k\mathbf{x}_k^\mathsf{T}\mathbf{Q}_{-k}(z_1)\mathbf{A}\mathbf{Q}_{-k}(z_2)\mathbf{x}_k\mathbf{x}_k^\mathsf{T}\frac{(\mathsf{b}_k - \tilde{\mathsf{b}})}{1 + \mathsf{b}_k}$$

$$= \mathbb{E}\mathbf{Q}_{-1}(z_1)\mathbf{A}\mathbf{Q}_{-1}(z_2)\mathbf{\Sigma} - \frac{1}{1+\tilde{\mathsf{b}}}\big(\mathbb{E}\mathsf{F}_1 + \mathbb{E}\mathsf{F}_2\big), \tag{46}$$

where

$$\mathsf{F}_1 = \frac{1}{N}\mathbf{Q}_{-1}(z_1)\mathbf{x}_1\mathbf{x}_1^\mathsf{T}\mathbf{Q}_{-1}(z_2)\mathbf{A}\mathbf{Q}_{-1}(z_2)\mathbf{x}_1\mathbf{x}_1^\mathsf{T},$$

$$\mathsf{F}_2 = \frac{1}{N}\mathbf{Q}_{-1}(z_1)\mathbf{x}_1\mathbf{x}_1^\mathsf{T}\mathbf{Q}_{-1}(z_2)\mathbf{A}\mathbf{Q}_{-1}(z_2)\mathbf{x}_1\mathbf{x}_1^\mathsf{T}\frac{(\mathsf{b}_1 - \tilde{\mathsf{b}})}{1 + \mathsf{b}_1}.$$

We first consider $\mathbb{E}\mathsf{F}_2$. Let $\tilde{\mathbf{u}}, \tilde{\mathbf{v}}$ denote a pair of unit vectors satisfying

$$\tilde{\mathbf{u}}, \tilde{\mathbf{v}} = \arg \max_{\|\tilde{\mathbf{u}}\|=\|\tilde{\mathbf{v}}\|=1} |\tilde{\mathbf{u}}^{\mathsf{T}} \mathbb{E}\mathsf{F}_2 \tilde{\mathbf{v}}|,$$

and let $\mathbf{y} = \mathbf{Q}_{-1}(z_1)\tilde{\mathbf{u}} = (y_1, ..., y_M)^{\mathsf{T}}$. Using the Burkholder's inequality (Burkholder, 1973), we have

$$\mathbb{E}|\mathbf{y}^{\mathsf{T}}\mathbf{x}_1|^4 = \mathbb{E}\left|\sum_{i=1}^{M} y_i x_{i1}\right|^4 \le c\mathbb{E}\left|\sum_{i=1}^{M} y_i^2\right|^2 + c\mathbb{E}\sum_{i=1}^{M} |y_i x_{i1}|^4$$

$$\le C\mathbb{E}\|\mathbf{y}\|^4 + C\mathbb{E}\sum_{i=1}^{M} y_i^4 \lesssim 1,$$

where we use the inequality

$$\sum_{i=1}^{M} y_i^4 \le \left(\sum_{i=1}^{M} y_i^2\right)^2 \le \|\mathbf{y}\|^4.$$

Likewise, we have $\mathbb{E}|\mathbf{x}_1^{\mathsf{T}}\tilde{\mathbf{v}}|^4 \lesssim 1$. It follows from Lemma 3 that

$$\mathbb{E}|\mathsf{b}_1 - \tilde{\mathsf{b}}|^{\ell} \le \frac{c}{N^{\ell}}[(\mathrm{Tr}\mathbf{Q}_{-1}^2(z_1))^{\ell/2} + \mathrm{Tr}\mathbf{Q}_{-1}^{\ell}(z_1)] \le \frac{C}{N^{\ell/2}},$$

and

$$\mathbb{E}|\mathbf{x}_1^{\mathsf{T}}\mathbf{Q}_{-1}(z_1)\mathbf{x}_1|^{\ell} \le C\mathbb{E}|\mathbf{x}_1^{\mathsf{T}}\mathbf{Q}_{-1}(z_1)\mathbf{x}_1 - \mathbb{E}\mathrm{Tr}\mathbf{\Sigma}\mathbf{Q}_{-1}(z_1)|^{\ell} + C|\mathbb{E}\mathrm{Tr}\mathbf{\Sigma}\mathbf{Q}_{-1}(z_1)|^{\ell} \lesssim N^{\ell}.$$

Since $\mathsf{b}_k > 1$, we can bound the spectral norm of $\mathbb{E}\mathsf{F}_2$ as

$$\|\mathbb{E}\mathsf{F}_2\| = |\tilde{\mathbf{u}}^{\mathsf{T}}\mathbb{E}\mathsf{F}_2\tilde{\mathbf{v}}| \le \mathbb{E}|\tilde{\mathbf{u}}^{\mathsf{T}}\mathsf{F}_2\tilde{\mathbf{v}}|$$

$$\le \frac{1}{N}\mathbb{E}|\tilde{\mathbf{u}}^{\mathsf{T}}\mathbf{Q}_{-1}(z_1)\mathbf{x}_1\mathbf{x}_1^{\mathsf{T}}\mathbf{Q}_{-1}(z_1)\mathbf{A}\mathbf{Q}_{-1}(z_2)\mathbf{x}_1\mathbf{x}_1^{\mathsf{T}}\tilde{\mathbf{v}}||\mathsf{b}_1 - \tilde{\mathsf{b}}|$$

$$\le \frac{1}{N}\sqrt{\mathbb{E}|\mathbf{y}^{\mathsf{T}}\mathbf{x}_1\mathbf{x}_1^{\mathsf{T}}\tilde{\mathbf{v}}|^2\mathbb{E}|\mathbf{x}_1^{\mathsf{T}}\mathbf{Q}_{-1}(z_1)\mathbf{A}\mathbf{Q}_{-1}(z_2)\mathbf{x}_1(\mathsf{b}_1 - \tilde{\mathsf{b}})|^2}$$

$$\le \frac{1}{N}\sqrt{\sqrt{\mathbb{E}|\mathbf{y}^{\mathsf{T}}\mathbf{x}_1|^4\mathbb{E}|\mathbf{x}_1^{\mathsf{T}}\tilde{\mathbf{v}}|^4}\sqrt{\mathbb{E}|\mathbf{x}_1^{\mathsf{T}}\mathbf{Q}_{-1}(z_1)\mathbf{A}\mathbf{Q}_{-1}(z_2)\mathbf{x}_1|^4\mathbb{E}|\mathsf{b}_1 - \tilde{\mathsf{b}}|^4}}$$

$$\le C\frac{1}{N}o(N) = o(1).$$

Therefore, it suffices to find the deterministic equivalent of $\mathbb{E}\mathsf{F}_1$. We recall the definition above equation 34 that $\mathbf{A}_M = o(1)$ if $\|\mathbf{A}_M\| = o(1)$. Let $\mathbb{E}_{-1}(\cdot) = \mathbb{E}[\cdot|\mathbf{x}_2, ..., \mathbf{x}_N]$. We have

$$\begin{aligned}
\mathbb{E}\mathsf{F}_1 &= \frac{1}{N}\mathbb{E}\mathbf{Q}_{-1}(z_1)\mathbf{x}_1\mathbf{x}_1^{\mathsf{T}}\mathbf{Q}_{-1}(z_1)\mathbf{A}\mathbf{Q}_{-1}(z_2)\mathbf{x}_1\mathbf{x}_1^{\mathsf{T}} \\
&= \frac{1}{N}\mathbb{E}\mathbf{Q}_{-1}(z_1)\big[\mathbb{E}_{-1}\mathbf{x}_1\mathbf{x}_1^{\mathsf{T}}\mathbf{Q}_{-1}(z_1)\mathbf{A}\mathbf{Q}_{-1}(z_2)\mathbf{x}_1\mathbf{x}_1^{\mathsf{T}}\big] \\
&= \frac{1}{N}\mathbb{E}\mathbf{Q}_{-1}(z_1)\mathbf{\Sigma}^{1/2}\mathbb{E}_{-1}[\mathbf{z}_1\mathbf{z}_1^{\mathsf{T}}\mathbf{\Sigma}^{1/2}\mathbf{Q}_{-1}(z_1)\mathbf{A}\mathbf{Q}_{-1}(z_2)\mathbf{\Sigma}^{1/2}\mathbf{z}_1\mathbf{z}_1^{\mathsf{T}}]\mathbf{\Sigma}^{1/2} \\
&= \frac{1}{N}\mathbb{E}\mathbf{Q}_{-1}(z_1)\big[\mathrm{Tr}\mathbf{\Sigma}\mathbf{Q}_{-1}(z_1)\mathbf{A}\mathbf{Q}_{-1}(z_2)\big]\mathbf{\Sigma} \\
&\quad + \frac{1}{N}\mathbb{E}\mathbf{Q}_{-1}(z_1)\mathbf{\Sigma}\big[\mathbf{Q}_{-1}(z_1)\mathbf{A}\mathbf{Q}_{-1}(z_2) + \mathbf{Q}_{-1}(z_2)\mathbf{A}\mathbf{Q}_{-1}(z_1)\big]\mathbf{\Sigma} \\
&\quad + \frac{1}{N}(\mathbb{E}z_{11}^4 - 3)\mathbb{E}\mathbf{Q}_{-1}(z_1)\mathbf{\Sigma}^{1/2}\mathrm{diag}(\mathbf{\Sigma}^{1/2}\mathbf{Q}_{-1}(z_1)\mathbf{A}\mathbf{Q}_{-1}(z_2)\mathbf{\Sigma}^{1/2})\mathbf{\Sigma}^{1/2} \\
&= \frac{1}{N}\mathbb{E}\big[\mathrm{Tr}\mathbf{\Sigma}\mathbf{Q}_{-1}(z_1)\mathbf{A}\mathbf{Q}_{-1}(z_2)\big]\mathbf{Q}_{-1}(z_1)\mathbf{\Sigma} + o(1) \\
&= \frac{1}{N}\big[\mathbb{E}\mathrm{Tr}\mathbf{\Sigma}\mathbf{Q}(z_1)\mathbf{A}\mathbf{Q}(z_2)\big]\mathbf{\Pi}(z_1)\mathbf{\Sigma} + o(1),
\end{aligned} \tag{47}$$

where the last identity is due to equation 37, equation 39 and

$$\frac{1}{N}\mathbb{E}\big[\mathrm{Tr}\boldsymbol{\Sigma}\mathbf{Q}_{-1}(z_1)\mathbf{A}\mathbf{Q}_{-1}(z_2)\big]\mathbf{Q}_{-1}(z_1)\boldsymbol{\Sigma}$$

$$=\frac{1}{N}\mathbb{E}\big[\mathrm{Tr}\boldsymbol{\Sigma}\mathbf{Q}_{-1}(z_1)\mathbf{A}\mathbf{Q}_{-1}(z_2) - \mathbb{E}\mathrm{Tr}\boldsymbol{\Sigma}\mathbf{Q}_{-1}(z_1)\mathbf{A}\mathbf{Q}_{-1}(z_2)\big]\mathbf{Q}_{-1}(z_1)\boldsymbol{\Sigma}$$

$$+\frac{1}{N}[\mathbb{E}\mathrm{Tr}\boldsymbol{\Sigma}\mathbf{Q}_{-1}(z_1)\mathbf{A}\mathbf{Q}_{-1}(z_2)]\mathbf{Q}_{-1}(z_1)\boldsymbol{\Sigma}$$

$$=\frac{1}{N}\mathbb{E}[\mathrm{Tr}\boldsymbol{\Sigma}\mathbf{Q}(z_1)\mathbf{A}\mathbf{Q}(z_2)]\boldsymbol{\Pi}(z_1)\boldsymbol{\Sigma} + o(1).$$

By equation 40, equation 42, equation 46 and equation 47 and the fact that $\|\boldsymbol{\Pi}(z_2)\|$ is bounded, we have

$$\frac{1}{N}\mathbb{E}\mathbf{Q}(z_1)\mathbf{A}\mathbf{Q}(z_2)\mathbf{X}\mathbf{X}^\mathsf{T}\boldsymbol{\Pi}(z_2)$$

$$=\frac{1}{(1+\tilde{b})}\mathbb{E}\mathbf{F}_1\boldsymbol{\Pi}(z_2) + o(1)$$

$$=\frac{1}{1+\tilde{b}}\left[\mathbb{E}\mathbf{Q}_{-1}(z_1)\mathbf{A}\mathbf{Q}_{-1}(z_2)\boldsymbol{\Sigma}\boldsymbol{\Pi}(z_2) - \frac{1}{1+\tilde{b}}\mathbb{E}\mathbf{F}_1\boldsymbol{\Pi}(z_2)\right] + o(1)$$

$$=\frac{\mathbb{E}\mathbf{Q}(z_1)\mathbf{A}\mathbf{Q}(z_2)\boldsymbol{\Sigma}\boldsymbol{\Pi}(z_2)}{1+\frac{1}{N}\mathrm{Tr}\boldsymbol{\Sigma}\boldsymbol{\Pi}(z_2)} - \frac{\frac{1}{N}[\mathbb{E}\mathrm{Tr}\boldsymbol{\Sigma}\mathbf{Q}(z_1)\mathbf{A}\mathbf{Q}(z_2)]\boldsymbol{\Pi}(z_1)\boldsymbol{\Sigma}\boldsymbol{\Pi}(z_2)}{(1+\frac{1}{N}\mathrm{Tr}\boldsymbol{\Sigma}\boldsymbol{\Pi}(z_2))(1+\frac{1}{N}\mathrm{Tr}\boldsymbol{\Sigma}\boldsymbol{\Pi}(z_1))} + o(1).$$

This, along with equation 34, equation 41, leads to

$$\mathbb{E}\mathbf{Q}(z_1)\mathbf{A}\mathbf{Q}(z_2)$$

$$=\boldsymbol{\Pi}(z_1)\mathbf{A}\boldsymbol{\Pi}(z_2) + \frac{\frac{1}{N}[\mathbb{E}\mathrm{Tr}\boldsymbol{\Sigma}\mathbf{Q}(z_1)\mathbf{A}\mathbf{Q}(z_2)]\boldsymbol{\Pi}(z_1)\boldsymbol{\Sigma}\boldsymbol{\Pi}(z_2)}{(1+\frac{1}{N}\mathrm{Tr}\boldsymbol{\Sigma}\boldsymbol{\Pi}(z_2))(1+\frac{1}{N}\mathrm{Tr}\boldsymbol{\Sigma}\boldsymbol{\Pi}(z_1))} + o(1). \tag{48}$$

Multiplying both sides of the above equation on the left by $\boldsymbol{\Sigma}$, and taking the trace, we obtain

$$\frac{1}{N}\mathbb{E}\mathrm{Tr}\boldsymbol{\Sigma}\mathbf{Q}(z_1)\mathbf{A}\mathbf{Q}(z_2)$$

$$=\frac{1}{N}\mathrm{Tr}\boldsymbol{\Sigma}\boldsymbol{\Pi}(z_1)\mathbf{A}\boldsymbol{\Pi}(z_2) + \frac{\frac{1}{N}[\mathbb{E}\mathrm{Tr}\boldsymbol{\Sigma}\mathbf{Q}(z_1)\mathbf{A}\mathbf{Q}(z_2)]\frac{1}{N}\mathrm{Tr}\boldsymbol{\Sigma}\boldsymbol{\Pi}(z_1)\boldsymbol{\Sigma}\boldsymbol{\Pi}(z_2)}{(1+\frac{1}{N}\mathrm{Tr}\boldsymbol{\Sigma}\boldsymbol{\Pi}(z_2))(1+\frac{1}{N}\mathrm{Tr}\boldsymbol{\Sigma}\boldsymbol{\Pi}(z_1))} + o(1).$$

It follows that

$$\frac{1}{N}\mathbb{E}\mathrm{Tr}\boldsymbol{\Sigma}\mathbf{Q}(z_1)\mathbf{A}\mathbf{Q}(z_2)$$

$$=\left(1 - \frac{\frac{1}{N}\mathrm{Tr}\boldsymbol{\Sigma}\boldsymbol{\Pi}(z_1)\boldsymbol{\Sigma}\boldsymbol{\Pi}(z_2)}{(1+\frac{1}{N}\mathrm{Tr}\boldsymbol{\Sigma}\boldsymbol{\Pi}(z_2))(1+\frac{1}{N}\mathrm{Tr}\boldsymbol{\Sigma}\boldsymbol{\Pi}(z_1))}\right)^{-1}\frac{1}{N}\mathrm{Tr}\boldsymbol{\Sigma}\boldsymbol{\Pi}(z_1)\mathbf{A}\boldsymbol{\Pi}(z_2) + o(1). \tag{49}$$

Plugging equation 49 into equation 48, we get

$$\mathbb{E}\mathbf{Q}(z_1)\mathbf{A}\mathbf{Q}(z_2) = \boldsymbol{\Pi}(z_1)\mathbf{A}\boldsymbol{\Pi}(z_2)$$

$$+\frac{\frac{1}{N}\mathrm{Tr}\boldsymbol{\Sigma}\boldsymbol{\Pi}(z_1)\mathbf{A}\boldsymbol{\Pi}(z_2)}{\big(1+\frac{1}{N}\mathrm{Tr}\boldsymbol{\Sigma}\boldsymbol{\Pi}(z_2)\big)\big(1+\frac{1}{N}\mathrm{Tr}\boldsymbol{\Sigma}\boldsymbol{\Pi}(z_1)\big) - \frac{1}{N}\mathrm{Tr}\boldsymbol{\Sigma}\boldsymbol{\Pi}(z_1)\boldsymbol{\Sigma}\boldsymbol{\Pi}(z_2)}\boldsymbol{\Pi}(z_1)\boldsymbol{\Sigma}\boldsymbol{\Pi}(z_2) + o(1). \tag{50}$$

The result equation 15 follows by combining the equation 50 with equation 33. Now we prove equation 17. Using a proof analogous to that of equation 33, we can obtain that

$$\frac{1}{M}\mathrm{Tr}\mathbf{C}\big[\mathbf{Q}(z_1)\mathbf{A}\mathbf{Q}_2(z_2) - \mathbb{E}\mathbf{Q}(z_1)\mathbf{A}\mathbf{Q}(z_2)\big] = o_{a.s.}(1). \tag{51}$$

We denote the spectral decomposition of $\mathbf{C}$ by

$$\mathbf{C} = \sum_{i=1}^{M}\lambda_i\mathbf{u}_i\mathbf{v}_i^\mathsf{T}.$$

By equation 50, we have

$$\frac{1}{M}\text{Tr}\mathbf{C}\mathbb{E}\mathbf{Q}(z_1)\mathbf{A}\mathbf{Q}(z_2) = \frac{1}{M}\text{Tr}\sum_{i=1}^{M}\lambda_i\mathbf{u}_i\mathbf{v}_i^{\mathsf{T}}\mathbb{E}\mathbf{Q}(z_1)\mathbf{A}\mathbf{Q}(z_2)$$

$$= \frac{1}{M}\sum_{i=1}^{M}\lambda_i\mathbf{v}_i^{\mathsf{T}}\mathbb{E}\mathbf{Q}(z_1)\mathbf{A}\mathbf{Q}(z_2)\mathbf{u}_i$$

$$= \frac{1}{M}\sum_{i=1}^{M}\lambda_i\mathbf{v}_i^{\mathsf{T}}\mathbf{\Pi}(z_1)\mathcal{S}(\mathbf{A})\mathbf{\Pi}(z_2)\mathbf{u}_i + o(1)$$

$$= \frac{1}{M}\text{Tr}\mathbf{C}\mathbf{\Pi}(z_1)\mathcal{S}(\mathbf{A})\mathbf{\Pi}(z_2) + o(1).$$

This, along with equation 51, establishes equation 17.

### B.9 DETAILS OF SECTION 5.2

In Case 1 of Section 5.2, where $\mathbf{X}_1 = \alpha\mathbf{X}_2 + \widetilde{\mathbf{X}}_1$, the following result holds.

**Proposition 2.** *Suppose that* $\widetilde{\mathbf{X}}_1, \mathbf{X}_2, \varepsilon_1$ *and* $\varepsilon_2$ *satisfy Assumptions 1-2. Then Theorem 1 continues to hold. Moreover, if we additionally impose Assumption 3, then Theorem 2 remains valid.*

**Proof**: We recall that $z_1 = -\lambda_{\mathsf{t}}, z_2 = -\lambda_{\mathsf{s}}$. We only consider $h_{55}$ in equation 19 here and the remaining terms can be handled analogously. By equation 21, it suffices to estimate

$$\frac{1}{N}\text{Tr}(\mathbf{I}_M + z_2\mathbf{Q}_2)\mathbf{\Sigma}_2(\mathbf{I}_M + z_2\mathbf{Q}_2)(\mathbf{Q}_1 + z_1\mathbf{Q}_1^2).$$

Since

$$\mathbf{Q}_1 = \left(\frac{1}{N}\mathbf{X}_1\mathbf{X}_1^{\mathsf{T}} - z_1\mathbf{I}_M\right)^{-1} = \left(\frac{1}{N}\widetilde{\mathbf{X}}_1\widetilde{\mathbf{X}}_1^{\mathsf{T}} - z_1\mathbf{I}_M + \mathbf{\Delta}\right)^{-1}, \quad \mathbf{\Delta} = \frac{\alpha}{N}(\widetilde{\mathbf{X}}_1\mathbf{X}_2^{\mathsf{T}} + \mathbf{X}_2\widetilde{\mathbf{X}}_1^{\mathsf{T}} + \alpha\mathbf{X}_2\mathbf{X}_2^{\mathsf{T}}).$$

We denote $\widetilde{\mathbf{Q}}_1 = \left(\frac{1}{N}\widetilde{\mathbf{X}}_1\widetilde{\mathbf{X}}_1^{\mathsf{T}} - z_1\mathbf{I}_M\right)^{-1}$. Applying equation 38 and Lemma 4, we have with high probability,

$$\|\mathbf{E}\| = \|\widetilde{\mathbf{Q}}_1 - \mathbf{Q}_1\| = \|\widetilde{\mathbf{Q}}_1\mathbf{\Delta}\mathbf{Q}_1\| \leq \frac{1}{|z_1|^2}\|\mathbf{\Delta}\| \lesssim \alpha = o(1). \tag{52}$$

Then we obtain

$$\frac{z_2}{N}\text{Tr}\mathbf{Q}_2\mathbf{\Sigma}_2\mathbf{Q}_1^2 = \frac{z_2}{N}\text{Tr}\mathbf{Q}_2\mathbf{\Sigma}_2(\widetilde{\mathbf{Q}}_1^2 + \underbrace{\mathbf{E}^2 + \mathbf{E}\widetilde{\mathbf{Q}}_1 + \widetilde{\mathbf{Q}}_1\mathbf{E}}_{\widehat{\mathbf{E}}})$$

$$= \frac{z_2}{N}\text{Tr}\mathbf{Q}_2\mathbf{\Sigma}_2\widetilde{\mathbf{Q}}_1^2 + o_{a.s.}(1),$$

where we use the fact that

$$\frac{1}{N}\text{Tr}\mathbf{Q}_2\mathbf{\Sigma}_2\widehat{\mathbf{E}} \leq \frac{M}{N}\|\mathbf{Q}_2\mathbf{\Sigma}_2\widehat{\mathbf{E}}\| \lesssim \|\widehat{\mathbf{E}}\| = o_{a.s.}(1).$$

By similar argument, we have

$$h_{55} = \frac{\xi^2\sigma^2}{N^2}\text{Tr}(\mathbf{I}_M + z_2\mathbf{Q}_2)\mathbf{\Sigma}_2(\mathbf{I}_M + z_2\mathbf{Q}_2)(\widetilde{\mathbf{Q}}_1 + z_1\widetilde{\mathbf{Q}}_1^2) + o_{a.s.}(1).$$

The proof is completed.

For Case 2 in Section 5.2, we have the following proposition, which also covers the setting of self-distillation.

**Proposition 3.** *Suppose* $\mathbf{X}_1 = \mathbf{X}_2 + \mathbf{A}$ *is a signal-plus-noise data matrix, with* $\|\mathbf{A}\| = o(\sqrt{M})$. *The regression parameter vector* $\boldsymbol{\beta} = \boldsymbol{\beta}_1 = \boldsymbol{\beta}_2$ *satisfies Assumption 3. When* $\lambda_{\mathsf{s}} \neq \lambda_{\mathsf{t}}$, *we have*

$$\widehat{\mathbf{Bias}} = \frac{a}{M}\text{Tr}\mathbf{\Sigma}_2[\mathbf{\Pi}_2(-\lambda_{\mathsf{t}}) - \mathbf{\Pi}_2(-\lambda_{\mathsf{s}})] + \frac{b}{M}\text{Tr}\mathbf{\Sigma}_2\mathbf{\Pi}_2'(-\lambda_{\mathsf{t}}) + \frac{c}{M}\text{Tr}\mathbf{\Sigma}_2\mathbf{\Pi}_2'(-\lambda_{\mathsf{s}}),$$

*and*

$$\widehat{\mathrm{Var}} = \frac{\xi^2\sigma^2}{N_1}\left(d\mathrm{Tr}\boldsymbol{\Sigma}_2[\boldsymbol{\Pi}_2(-\lambda_\mathsf{t}) - \boldsymbol{\Pi}_2(-\lambda_\mathsf{s})] + e\mathrm{Tr}\boldsymbol{\Sigma}_2\boldsymbol{\Pi}_2'(-\lambda_\mathsf{t}) + f\mathrm{Tr}\boldsymbol{\Sigma}_2\boldsymbol{\Pi}_2'(-\lambda_\mathsf{s})\right)$$

$$+ (1-\xi)^2\sigma^2\frac{1}{N_2}\mathrm{Tr}\boldsymbol{\Sigma}_2[\boldsymbol{\Pi}_2(-\lambda_\mathsf{s}) - \lambda_\mathsf{s}\boldsymbol{\Pi}_2'(-\lambda_\mathsf{s})],$$

*where*

$$a = \frac{2\xi\lambda_1\lambda_2}{\lambda_\mathsf{s} - \lambda_\mathsf{t}} + \frac{2\xi\lambda_\mathsf{t}\lambda_\mathsf{s}(\xi\lambda_\mathsf{t} - \lambda_\mathsf{s})}{(\lambda_\mathsf{s} - \lambda_\mathsf{t})^2} - \frac{2\xi\lambda_\mathsf{s}^2\lambda_\mathsf{t}^2}{(\lambda_\mathsf{s} - \lambda_\mathsf{s})^3}, \; b = \xi^2\lambda_\mathsf{t}^2 - \frac{2\xi^2\lambda_\mathsf{t}^2\lambda_\mathsf{s}}{\lambda_\mathsf{t} - \lambda_\mathsf{s}},$$

$$c = \lambda_\mathsf{s}^2 - \frac{2\xi\lambda_\mathsf{t}\lambda_\mathsf{s}^2}{\lambda_\mathsf{t} - \lambda_\mathsf{s}}, \; d = \frac{2\lambda_\mathsf{s}}{\lambda_\mathsf{t} - \lambda_\mathsf{s}} + \frac{\lambda_\mathsf{s}^2}{(\lambda_\mathsf{t} - \lambda_\mathsf{s})^2} + \frac{\lambda_\mathsf{t}\lambda_\mathsf{s}^2}{(\lambda_\mathsf{s} - \lambda_\mathsf{t})^3} + \frac{2\lambda_\mathsf{s}\lambda_\mathsf{t}}{(\lambda_\mathsf{t} - \lambda_\mathsf{s})^2},$$

$$e = -\lambda_\mathsf{t} + \frac{2\lambda_\mathsf{s}\lambda_\mathsf{t}}{(\lambda_\mathsf{t} - \lambda_\mathsf{s})^2} - \frac{\lambda_\mathsf{t}\lambda_\mathsf{s}^2}{(\lambda_\mathsf{s} - \lambda_\mathsf{t})^2}, \quad f = \frac{\lambda_\mathsf{s}^2}{\lambda_\mathsf{s} - \lambda_\mathsf{t}} - \frac{\lambda_\mathsf{t}\lambda_\mathsf{s}^2}{(\lambda_\mathsf{s} - \lambda_\mathsf{t})^2}.$$

*When $\lambda = \lambda_\mathsf{s} = \lambda_\mathsf{t}$, $\widehat{\mathrm{Bias}}$ is given in equation 54 and*

$$\widehat{\mathrm{Var}} = (1-\xi)^2\sigma^2\frac{1}{N_2}\mathrm{Tr}\boldsymbol{\Sigma}_2(\boldsymbol{\Pi}_2 + \lambda\boldsymbol{\Pi}_2') + \frac{\xi^2\sigma^2}{N_1}\mathrm{Tr}\boldsymbol{\Sigma}_2[\boldsymbol{\Pi}_2 - 3\lambda\boldsymbol{\Pi}_2' + 3\lambda^2\boldsymbol{\Pi}_2^{(2)} - \lambda^3\boldsymbol{\Pi}_2^{(3)}],$$

*with $\boldsymbol{\Pi}_2^{(k)} = \frac{\mathrm{d}^k\boldsymbol{\Pi}_2(z)}{\mathrm{d}z^k}\big|_{z=-\lambda}$.*

**Proof**: By an argument analogous to that used for equation 52, one may readily verify that

$$\|\mathbf{Q}_1(z) - \mathbf{Q}_2(z)\| = o(1).$$

Then equation 26 becomes

$$\frac{1}{M}\mathrm{Tr}\boldsymbol{\Sigma}_2\mathbf{H}\mathbf{H}^\mathsf{T} = \frac{1}{M}\left[\xi^2 z_1^2\mathrm{Tr}\boldsymbol{\Sigma}_2\mathbf{Q}_2^2(z_1) + 2\xi z_1 z_2\underbrace{\mathrm{Tr}\boldsymbol{\Sigma}_2\mathbf{Q}_2(z_1)\mathbf{Q}_2(z_2)}_{t_1} + 2\xi^2 z_1^2 z_2\underbrace{\mathrm{Tr}\boldsymbol{\Sigma}_2\mathbf{Q}_2(z_2)\mathbf{Q}_2^2(z_1)}_{t_2}\right.$$

$$+ z_2^2\mathrm{Tr}\boldsymbol{\Sigma}_2\mathbf{Q}_2^2(z_2) + 2\xi z_1 z_2^2\underbrace{\mathrm{Tr}[\mathbf{Q}_2(z_2)\boldsymbol{\Sigma}_2\mathbf{Q}_2(z_2)\mathbf{Q}_2(z_1)]}_{t_3}$$

$$\left.+ \xi^2 z_1^2 z_2^2\underbrace{\mathrm{Tr}[\mathbf{Q}_2(z_2)\boldsymbol{\Sigma}_2\mathbf{Q}_2(z_2)\mathbf{Q}_2^2(z_1)]}_{t_4}\right] + o(1).$$

We note that when $z_1 \neq z_2$,

$$\mathbf{Q}_2(z_1) - \mathbf{Q}_2(z_2) = (z_1 - z_2)\mathbf{Q}_2(z_1)\mathbf{Q}(z_2), \quad \mathbf{Q}_2(z_1)\mathbf{Q}_2(z_2) = \mathbf{Q}_2(z_2)\mathbf{Q}_2(z_1).$$

Then we have

$$\frac{1}{M}t_1 = \frac{2\xi z_1 z_2}{M}\mathrm{Tr}\boldsymbol{\Sigma}_2\frac{\mathbf{Q}_2(z_1) - \mathbf{Q}_2(z_2)}{z_1 - z_2} = \frac{2\xi z_1 z_2}{M(z_1 - z_2)}\mathrm{Tr}\boldsymbol{\Sigma}_2[\boldsymbol{\Pi}_2(z_1) - \boldsymbol{\Pi}_2(z_2)] + o_{a.s.}(1),$$

$$\frac{1}{M}t_2 = \frac{2\xi^2 z_1^2 z_2}{M(z_1 - z_2)}\mathrm{Tr}\boldsymbol{\Sigma}_2[\mathbf{Q}_2(z_1) - \mathbf{Q}_2(z_2)]\mathbf{Q}_2(z_1)$$

$$= -\frac{2\xi^2 z_1^2 z_2}{M(z_1 - z_2)}\mathrm{Tr}\boldsymbol{\Sigma}_2\boldsymbol{\Pi}_2'(z_1) - \frac{2\xi^2 z_1^2 z_2}{M(z_1 - z_2)^2}\mathrm{Tr}\boldsymbol{\Sigma}_2[\boldsymbol{\Pi}_2(z_1) - \boldsymbol{\Pi}_2(z_2)] + o_{a.s.}(1),$$

$$\frac{1}{M}t_3 = \frac{2\xi z_1 z_2^2}{M(z_1 - z_2)}\mathrm{Tr}\boldsymbol{\Sigma}_2[\mathbf{Q}_2(z_1) - \mathbf{Q}_2(z_2)]\mathbf{Q}_2(z_2)$$

$$= -\frac{2\xi z_1 z_2^2}{M(z_1 - z_2)}\mathrm{Tr}\boldsymbol{\Sigma}_2\boldsymbol{\Pi}_2'(z_2) + \frac{2\xi z_1 z_2^2}{M(z_1 - z_2)^2}\mathrm{Tr}\boldsymbol{\Sigma}_2[\boldsymbol{\Pi}_2(z_1) - \boldsymbol{\Pi}_2(z_2)] + o_{a.s.}(1),$$

$$\frac{1}{M}t_4 = \frac{\xi^2 z_1^2 z_2^2}{M(z_1 - z_2)^2}\mathrm{Tr}\boldsymbol{\Sigma}_2[\mathbf{Q}_2(z_1) - \mathbf{Q}_2(z_2)]^2$$

$$= \frac{\xi z_1^2 z_2^2}{M(z_1 - z_2)^2}\mathrm{Tr}\boldsymbol{\Sigma}_2\left[\mathbf{Q}_2^2(z_2) + \mathbf{Q}_2^2(z_1) - 2\frac{\mathbf{Q}_2(z_1) - \mathbf{Q}_2(z_2)}{z_1 - z_2}\right]$$

$$= \frac{\xi^2 z_1^2 z_2^2}{M(z_1 - z_2)^2}\mathrm{Tr}\boldsymbol{\Sigma}_2[\boldsymbol{\Pi}_2'(z_1) + \boldsymbol{\Pi}_2'(z_2)] - \frac{2\xi z_1^2 z_2^2}{M(z_1 - z_2)^3}\mathrm{Tr}\boldsymbol{\Sigma}_2[\boldsymbol{\Pi}_2(z_1) - \boldsymbol{\Pi}_2(z_2)] + o_{a.s.}(1).$$

Based on above results, we have $\frac{1}{M}\mathrm{Tr}\boldsymbol{\Sigma}_2\mathbf{H}\mathbf{H}^\mathsf{T} = \widehat{\mathbf{Bias}} + o_{a.s.}(1)$. As for the variance, one may check that

$$
\begin{aligned}
h_{55} = \frac{\xi^2\sigma^2}{N_1}\mathrm{Tr}\Bigg[ &\boldsymbol{\Sigma}_2[\boldsymbol{\Pi}_2(z_1) + z_1\boldsymbol{\Pi}_2'(z_1)] + 2z_2(z_2 - z_1)^{-1}\boldsymbol{\Sigma}_2[\boldsymbol{\Pi}_2(z_2) - \boldsymbol{\Pi}_2(z_1)] \\
&+ z_2^2(z_2 - z_1)^{-1}\boldsymbol{\Sigma}_2\boldsymbol{\Pi}_2'(z_2) + z_2^2(z_1 - z_2)^{-2}\boldsymbol{\Sigma}_2[\boldsymbol{\Pi}_2(z_1) - \boldsymbol{\Pi}_2(z_2)] - 2z_1z_2(z_2 - z_1)^{-2}\mathrm{Tr}\boldsymbol{\Sigma}_2\boldsymbol{\Pi}_2'(z_1) \\
&+ 2z_1z_2(z_1 - z_2)^{-2}\mathrm{Tr}\boldsymbol{\Sigma}_2[\boldsymbol{\Pi}_2(z_2) - \boldsymbol{\Pi}_2(z_1)] \\
&+ z_1z_2^2(z_1 - z_2)^{-2}\mathrm{Tr}\boldsymbol{\Sigma}_2[\boldsymbol{\Pi}_2'(z_1) + \boldsymbol{\Pi}'(z_2)] - z_1z_2^2(z_1 - z_2)^3\mathrm{Tr}[\boldsymbol{\Pi}_2(z_1) - \boldsymbol{\Pi}_2(z_2)]\Bigg] + o_{a.s.}(1),
\end{aligned}
\tag{53}
$$

and the limit of $h_{66}$ is the same as that in equation 24, where $h_{55}, h_{66}$ are given in Appendix B.2. When $\lambda = \lambda_\mathsf{t} = \lambda_\mathsf{s}$, denoting $\mathbf{Q}_2 = \mathbf{Q}_2(-\lambda)$, we have

$$
\begin{aligned}
\frac{1}{M}\mathrm{Tr}\boldsymbol{\Sigma}_2\mathbf{H}\mathbf{H}^\mathsf{T} &= \frac{1}{M}\Big[ (1 + \xi)^2\lambda^2\mathrm{Tr}\boldsymbol{\Sigma}_2\mathbf{Q}_2^2 - (2\xi + \xi^2)\lambda^3\mathrm{Tr}\boldsymbol{\Sigma}_2\mathbf{Q}_2^3 + \xi^2\lambda^4\mathrm{Tr}\boldsymbol{\Sigma}_2\mathbf{Q}_2^4 \Big] \\
&= \underbrace{\frac{1}{M}\Big[ (1 + \xi)^2\lambda^2\mathrm{Tr}\boldsymbol{\Sigma}_2\boldsymbol{\Pi}_2' - (2\xi + \xi^2)\lambda^3\mathrm{Tr}\boldsymbol{\Sigma}_2\boldsymbol{\Pi}_2^{(2)} + \xi^4\lambda^4\mathrm{Tr}\boldsymbol{\Sigma}_2\boldsymbol{\Pi}_2^{(3)} \Big]}_{\widehat{\mathbf{Bias}}} + o_{a.s.}(1),
\end{aligned}
\tag{54}
$$

where we use Vitali's convergence theorem. Similarly, we have

$$
h_{55} = \frac{\xi^2\sigma^2}{N_1}\mathrm{Tr}\boldsymbol{\Sigma}_2[\boldsymbol{\Pi}_2 + 3z\boldsymbol{\Pi}_2' + 3z^2\boldsymbol{\Pi}_2^{(2)} + z^3\boldsymbol{\Pi}_2^{(3)}] + o_{a.s.}(1),
$$

and the limit of $h_{66}$ coincides with the one given in equation 24. The proof is completed.

## C   ADDITIONAL EXPERIMENTAL DETAILS

### C.1   NONLINEAR MODELS

When considering more complex models, we fix the student model and let the teacher model be a deeper fully connected neural network:

$$
f_{\mathrm{NN}}^\mathsf{t} = \mathbf{a}_\mathsf{t}^\mathsf{T}\sigma(\widetilde{\mathbf{W}}_3\sigma(\widetilde{\mathbf{W}}_2\sigma(\widetilde{\mathbf{W}}_1\mathbf{x}))),
$$

where

$$
\mathbf{a}_\mathsf{t} = \arg\min_{\mathbf{a}}\frac{1}{N_1}\|\mathbf{y}_1 - [\sigma(\widetilde{\mathbf{W}}_3\sigma(\widetilde{\mathbf{W}}_2\sigma(\widetilde{\mathbf{W}}_1\mathbf{X}_1)))]^\mathsf{T}\mathbf{a}\|^2 + \lambda_\mathsf{t}\|\mathbf{a}\|^2.
$$

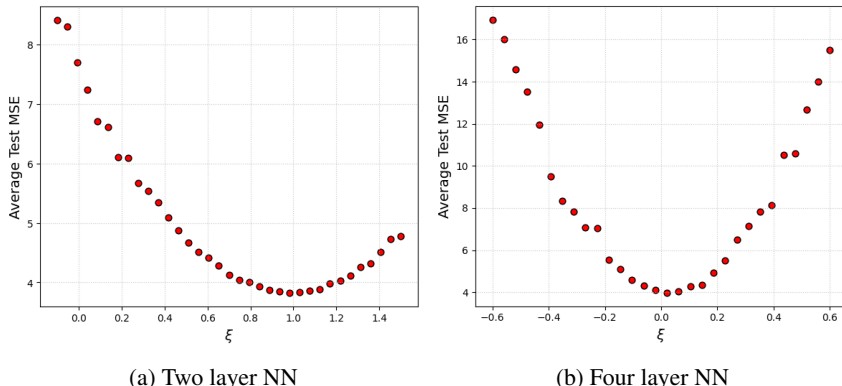

(a) Two layer NN  (b) Four layer NN

Figure 4: Excess risk estimated over 30 trials. We set $(M, N_1, N_2) = (50, 200, 100)$. (a) Settings: $(\lambda_t, \lambda_s) = (0.5, 0.2)$, $\sigma(x) = x^3$. The weight matrices $\widetilde{\mathbf{W}} \in \mathbb{R}^{n_1 \times M}$ and $\mathbf{W} \in \mathbb{R}^{n \times M}$ have i.i.d. centered Gaussian entries with variance $M^{-1}$, where $(n, n_1) = (100, 200)$. (b) Settings: $\lambda_t = \lambda_s = 0.2$, $(n_0, n_1, n_2, n_3) = (M, 600, 400, 200)$. The weight matrices $\widetilde{\mathbf{W}}_i \in \mathbb{R}^{n_i \times n_{i-1}}$ have i.i.d. centered Gaussian random variables with variance $n_{i-1}^{-1}$. We use the Leaky ReLU activation: $\sigma(x) = 0.01 x \delta(x \leq 0) + x \delta(x > 0)$.

## C.2 DEMONSTRATION OF COROLLARY 3

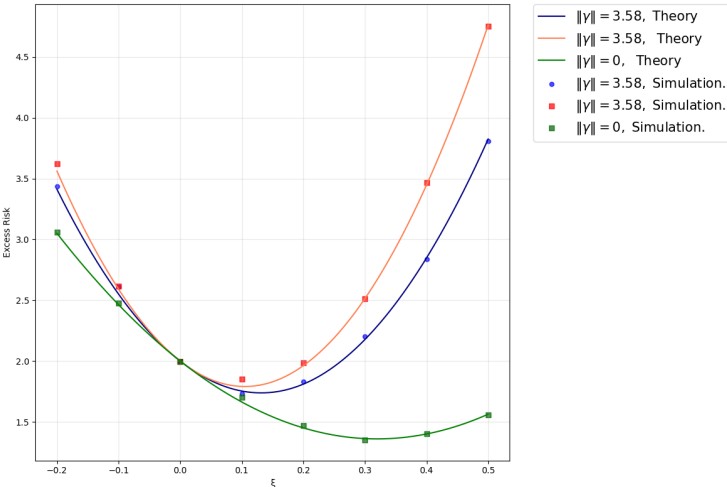

Figure 5: Theoretical predictions (solid curves) versus simulation results (scatter points, averaged over 100 independent trials) for ridgeless regression. We set $(M, N_1, N_2) = (400, 600, 600)$, $\boldsymbol{\beta}_2 = \frac{4}{\sqrt{M}}(1, ..., 1)^{\mathsf{T}}$, $\sigma^2 = 1$ and $\boldsymbol{\Sigma}_2 = \mathbf{I}_M$. We label the case $\|\boldsymbol{\gamma}\| = 3.58$ as $\boldsymbol{\gamma} = \frac{-8}{\sqrt{M}}(1, ..., 1, 0, ..., 0)^{\mathsf{T}}$ with the first $M/5$ entries equal to 1. The orange and green curves correspond to the setting where $\boldsymbol{\Sigma}_1 = \text{diag}(4, ..., 4, \frac{1}{4}, ..., \frac{1}{4})$, with the first half of the diagonal entries equal to 4 and the second half equal to $\frac{1}{4}$. The dark blue curve corresponds to the setting where $\boldsymbol{\Sigma}_1 = 4\mathbf{I}_M$.

Figure 5 presents empirical results that support 3. The gap between the orange and green curves quantifies the impact of model shift on the excess risk. Furthermore, the gap between the dark blue and orange curves reflects the role of the term $\text{Tr}\boldsymbol{\Sigma}_1^{-1}\boldsymbol{\Sigma}_2$ as characterized in Corollary 3.

## C.3 IMPACT OF REGULARIZATION PARAMETERS

To examine the impact of the regularization parameters $\lambda_t, \lambda_s$, we plot the empirical excess risk of the student model for $(\lambda_t, \lambda_s) \in [0.01, 0.5]^2$ in Figures 6-8 (averaged over 5 trials), correspond-

ing to $\xi = 0.5, -0.5$ and $1.5$, respectively. We set $\boldsymbol{\beta}_1 = \boldsymbol{\beta}_2 \sim \mathcal{N}(0, \frac{1}{M}\mathbf{I}_M), (M, N_1, N_2) = (400, 300, 200), \sigma^2 = 1$. We set $\boldsymbol{\Sigma}_2 = \mathbf{I}_M$ in the absence of covariate shift. Under covariate shift, we set $\boldsymbol{\Sigma}_1 = \mathrm{diag}(d_1, ..., d_M)$, where

$$d_i = 0.64\delta(i \leq M/2) + 0.25\delta(M/2 < i \leq M).$$

From these figures, we observe that when $\xi > 1$, the influence of $\lambda_t$ becomes large. In contrast, in the case $\xi = -0.5$, $\lambda_s$ almost dominates the variation of the excess risk, reflecting a weaker impact of the teacher's guidance (anti-learning against the teacher's supervision).

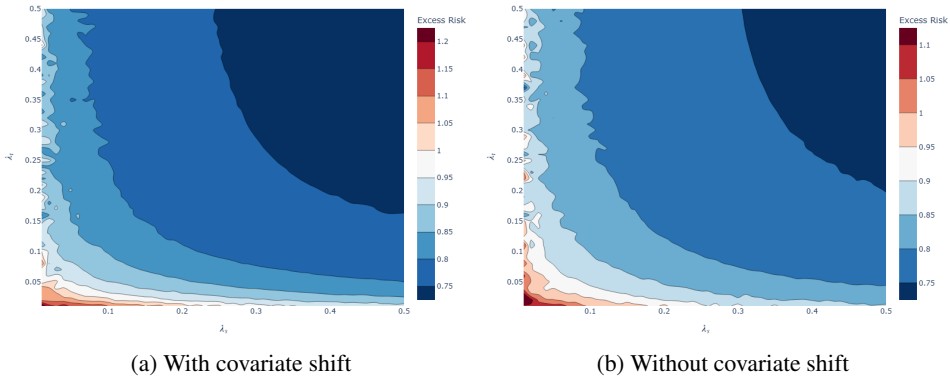

(a) With covariate shift        (b) Without covariate shift

Figure 6: Excess risk when $\xi = 0.5$.

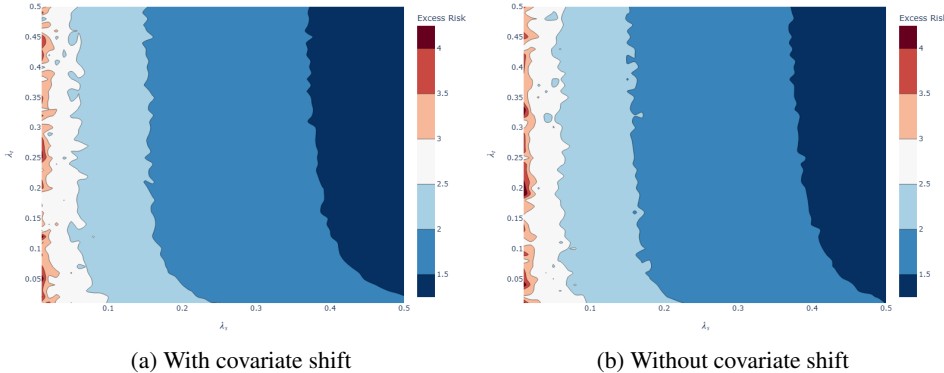

(a) With covariate shift        (b) Without covariate shift

Figure 7: Excess risk when $\xi = -0.5$.

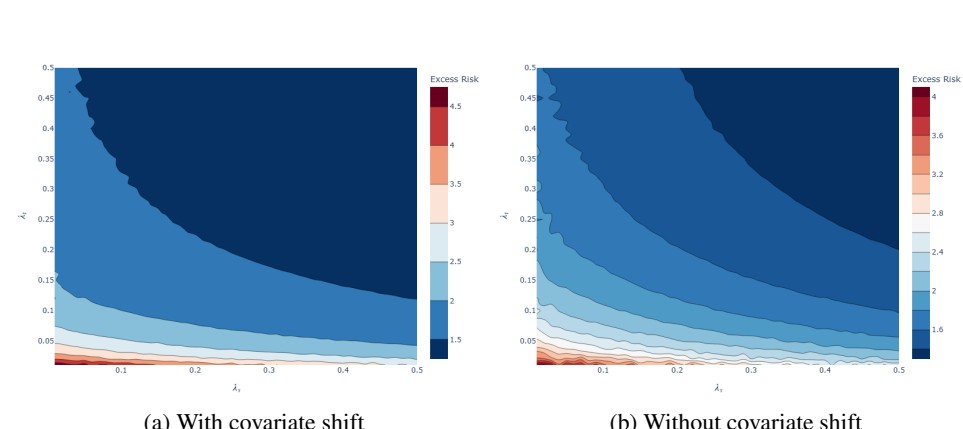

(a) With covariate shift          (b) Without covariate shift

Figure 8: Excess risk when $\xi = 1.5$.

