# OpenReview forum: "Quantifying Cross-Domain Knowledge Distillation in the Presence of Domain shift"
_ICLR.cc/2026/Conference — Submitted to ICLR 2026_

### Official Review · Reviewer_rJMR · 2025-10-30

**Soundness:** 3
**Presentation:** 3
**Contribution:** 3
**Rating:** 4
**Confidence:** 3

**Summary:**

This paper builds upon high-dimensional random matrix theory to analytically study cross-domain knowledge distillation (KD) in teacher–student linear regression settings under both covariate and model shifts. It derives asymptotic expressions for the excess risk through bias–variance decomposition, covering both regularized (ridge) and unregularized (ridgeless) regimes with deterministic and random parameters. The analysis includes an exploration of the imitation parameter $\xi$, showing that the optimal $\xi$ may lie outside the interval [0,1] (even negative), and discusses the conditions under which double descent arises.

**Strengths:**

1. Clear, testable theoretical claims: The paper presents three main results that express the excess risk of cross-domain KD as explicit matrix functions involving Stieltjes transforms. The derivations are mathematically sound and well-grounded in the theory of random matrices.
2. Beyond single-domain baselines: The work demonstrates the existence of an imitation coefficient $\xi$ that allows the student model to outperform pure supervised learning (Proposition 1).
3. Unified explanation of observed phenomena: The analysis connects double descent behavior in KD with the interplay among $\xi$, teacher/student regularization, and the dimensional ratio, offering a coherent theoretical interpretation.

**Weaknesses:**

1. Limited practical applicability beyond linear models: All results are derived for linear regression; implications for nonlinear or deep KD remain speculative. A discussion or small-scale experiment validating the theoretical predictions in nonlinear settings would strengthen the paper.
2. Empirical validation is light: The experiments are limited to synthetic data, verifying asymptotic formulas.
3. Strong assumptions and limited robustness analysis: Theoretical results rely on independence, bounded moments, and concentration assumptions.

**Questions:**

See Weakness.

---

> ### Author Response · Authors · 2025-11-21
>
> Thank you for your valuable comments and suggestions.  Your recognition of our clear, testable theoretical claims, the advancement beyond single-domain baselines, and the unified explanation of observed phenomena is greatly encouraging. We address your concerns below:
>
> > Limited practical applicability beyond linear models: All results are derived for linear regression; implications for nonlinear or deep KD remain speculative. A discussion or small-scale experiment validating the theoretical predictions in nonlinear settings would strengthen the paper.
>
> We anticipate that the key phenomena revealed by our theory may persist beyond the linear regime. To support this claim, we have added a new Section 5.1 describing the experimental setup for nonlinear models. The corresponding empirical results are presented in Figure 4 of Appendix C.1 in the revised manuscript. These simulations provide strong evidence that the mechanisms identified in the linear regime are robust and likely to generalize to more complex nonlinear settings. For your convenience, we summarize the experimental setup below; this same description appears in the newly added Section 5.
>
> ---
> *We assume the source data $\\{(\mathbf{x}_j^{(1)},{y}_j^{(1)} )\\}\_{j=1}^{N_1} $ are generated i.i.d. according to*
>
> \\[
> y_j^{(1)} =f(\mathbf{x}_j^{(1)} )+\varepsilon^{(1)}_j,\text{ for } 1\le j\le N_1.
> \\]
>
> *The target data $\\{(\mathbf{x}_j^{(2)},y_j^{(2)})\\} \_{j=1}^{N_2} $ are generated according to*
> \\[
> y_j^{(2)} =\tilde{f}(\mathbf{x}_j^{(2)} )+\varepsilon_j^{(2)},\text{ for } 1\le j\le N_2
> \\]
> *Suppose $\mathbf{x}_j^{(1)}\sim D_1,1\le j\le N_1$ and $\mathbf{x}_j^{(2)}\sim D_2,1\le j\le N_2. $ We refer to the case $D_1\neq D_2$ as a covariate shift, and the case where $f\neq \tilde{f}$ as a model shift.*
>
> *We consider learning the unknown function using a fully connected two-layer neural network with $n$ hidden neurons:
> $f_{\mathrm{NN}}(\mathbf{x})=\mathbf{a}^{\mathsf{T}}\sigma(\mathbf{Wx}),$ where $\mathbf{W}\in \mathbb{R}^{n\times M} $ is the  weight matrix, and $\sigma(\cdot)$ is an activation function applied entrywise.
> When the random weight matrix $\mathbf{W}$ is fixed and only the second-layer weight $\mathbf{a}$ is  optimized, the model reduces to a kernel regression model, where the kernel defined by $\mathbf{x}\to \sigma(\mathbf{Wx})$ is referred to as the conjugate kernel. The teacher model is given by*
>
> \\[
> f\_{\mathrm{NN}}^{\mathsf{t}}(\mathbf{x})=\mathbf{a}\_\mathsf{t}^{\mathsf{T}}\sigma(\widetilde{\mathbf{W}}\_1\mathbf{x}),
> \\]
> *with*
> $$
> \mathbf{a}\_{\mathsf{t}}=\arg\min\_{\mathbf{a}}\bigg\\{\frac{1}{N\_1}\\|\mathbf{y}\_1-\sigma(\mathbf{X}\_1^{\mathsf{T}}\widetilde{\mathbf{W}}\_1^{\mathsf{T}}) \mathbf{a}\\|^2+\lambda\_{\mathsf{t}} \\|\mathbf{a}\\|^2  \bigg\\}.
> $$
>
> *We use $f\_{\mathrm{NN}}^{\mathsf{t}}(\mathbf{x}\_j^{(2)})$ together with the covariates $\\{\mathbf{x}\_j^{(2)}\\}\_{j=1}^{N\_2}$ to generate predictions $\mathbf{y}\_2^{\mathsf{t}}$. Then the student model is finetuned on the target domain data and $\mathbf{y}\_2^{\mathsf{t}}$. The student model takes the form
> $f\_{\mathrm{NN}}^{\mathsf{s}}(\mathbf{x})=\mathbf{a}\_{\mathsf{s}}^{\mathsf{T}}\sigma(\mathbf{W}\_1\mathbf{x})$
> with*
> $$
> \mathbf{a}\_{\mathsf{s}}=\arg\min\_{\mathbf{a}}\xi\bigg(\frac{1}{N\_2} \\| \mathbf{y}\_2^{\mathsf{t}}-\sigma(\mathbf{X}\_2^{\mathsf{T}}\mathbf{W}\_1^{\mathsf{T}} )\mathbf{a} \\|^2 \bigg)+(1-\xi)\bigg( \frac{1}{N\_2} \\|\mathbf{y}_2-\sigma(\mathbf{X}\_2^{\mathsf{T}}\mathbf{W}\_1^{\mathsf{T}} )\mathbf{a} \\|^2\bigg)+\lambda\_{\mathsf{s}} \\|\mathbf{a} \\|^2.
> $$
> *We also examine a setting where the teacher model is a deeper neural network. Specifically, while keeping the student model fixed, we let the teacher be a 4-layer fully connected network:*
>
> $$
> f\_{\mathrm{NN}}^{\mathsf{t}}=\mathbf{a}\_{\mathsf{t}}^{\mathsf{T}}\sigma(\widetilde{\mathbf{W}}\_3\sigma(\widetilde{\mathbf{W}}\_2\sigma(\widetilde{\mathbf{W}}\_1\mathbf{x}))),
> $$
>
> *where*
>
> $$
> \mathbf{a}\_{\mathsf{t}}=\arg\min\_{\mathbf{a}}\frac{1}{N\_1}\\|\mathbf{y}_1-[\sigma(\widetilde{\mathbf{W}}\_3\sigma(\widetilde{\mathbf{W}}\_2\sigma(\widetilde{\mathbf{W}}\_1\mathbf{X}\_1))]^{\mathsf{T}}\mathbf{a}\\|^2+\lambda\_{\mathsf{t}}\\|\mathbf{a}\\|^2.
> $$
>
> *We set $D_1=\mathcal{N}(0,4\mathbf{I}_M)$ and $D_2=\mathcal{N}(0,\mathbf{I}_M)$. Let
> $$
> f(\mathbf{x})= (\boldsymbol{\beta}^{\mathsf{T}}\mathbf{x} )^2+1,\quad \tilde{f}(\mathbf{x})=(\boldsymbol{\beta}^{\mathsf{T}}\mathbf{x})^2.
> $$
> Because $D_1\neq D_2$ and $f\neq \tilde{f}$, both covariate shift and model shift are present in this setting. More details and the
> numerical results are provided in Appendix C.1.*
>
> ---

---

> ### Author Response · Authors · 2025-11-21
>
> > Empirical validation is light: The experiments are limited to synthetic data, verifying asymptotic formulas.
>
> We  agree that empirical evaluation on real-world data is essential to assess practical relevance. Following the reviewer's suggestion, we now apply our approach to the Human Activity Recognition Using Smartphones dataset from the commonly used UCI repository (available at https://archive.ics.uci.edu/dataset/240/human+activity+recognition+using+smartphones). This dataset contains 561 dimensional sensor signals collected from 30 individuals performing various activities.  We are interested in a binary classification task, where the **Active** class includes walking, walking upstairs, walking downstairs, and standing, while the **Rest** class includes sitting and laying.
>
> We treat the data from Subject 25 (the largest individual dataset) as the target domain. The target dataset is divided into a training set (the last 70\%, containing 286 samples) and a testing set (the first 30\%, containing 123 samples). The remaining 29 subjects (a total of 9,890 samples) constitute the source domain.  We set $\lambda\_t=\lambda\_s=0.1$. Denote the training data by $( {X}\_2^{\mathsf{T}}, {y})$ and the test data by $(X\_{t,2}^{\mathsf{T}} ,y\_t )$, where $ {X}\_2=( {x}\_1,..., {x}\_{N\_2})$, $X\_{t,2}=(x\_{t,1},...,x_{t,n\_2})$ with $N\_2=286, n\_2=123$.  To obtain soft labels in [0,1], we apply the sigmoid function to teacher's predictions:
> $$ {y}\_2^{{t}}=\text{sigmoid}( {\beta}\_{t}^{\mathsf{T}}  {X}_{2} ),$$
> where the sigmoid function is applied entrywise.
> The student estimator $ {\beta}\_{\mathsf{s}}$ then takes the form:
> $$
>  {\beta}\_{s}=\frac{1}{N_2} {Q}\_2(\xi {X}\_2 {y}_2^{t}+(1-\xi) {X}\_2 {y} ).
> $$
> For each test sample, we apply the sigmoid function to the predicted output and assign label 1 if the value exceeds 0.5, and 0 otherwise. The results for different values of $\xi$ are summarized in Table 1 below, which indicate that **anti-learning against the teacher’s supervision $(\xi<0)$ may lead
> to better generalization ability**. Remarkably, the student achieves perfect classification accuracy when $\xi=-1$ or $\xi=-0.849$.  This finding provides empirical evidence supporting the necessity of optimizing $\xi$ over $\mathbb{R}$.
>
> Table 1: Test accuracy vs. imitation parameter $\xi$.
>
> | $\xi$     | Accuracy | $\xi$     | Accuracy |
> |-----------|----------|-----------|----------|
> | -1.000    | 1.0000   | 0.206     | 0.9918   |
> | -0.849    | 1.0000   | 0.357     | 0.9836   |
> | -0.698    | 0.9918   | 0.508     | 0.9754   |
> | -0.548    | 0.9918   | 0.658     | 0.9590   |
> | -0.397    | 0.9918   | 0.809     | 0.9016   |
> | -0.246    | 0.9918   | 0.960     | 0.7623   |
> | -0.095    | 0.9918   | 1.111     | 0.5492   |
> | 0.055     | 0.9918   | 1.261     | 0.5492   |
>
> > Strong assumptions and limited robustness analysis: Theoretical results rely on independence, bounded moments, and concentration assumptions.
>
> We provide a detailed response to the three key concerns below.
>
> + **Concern 1: Independence between domains.**  Following your suggestion, we have extended our theoretical analysis in the revised manuscript to accommodate dependent source and target domains, which more accurately reflect practical KD scenarios. Specifically, we have introduced two forms of domain dependence: Case 1, corresponding to weak dependence, and Case 2, corresponding to strong dependence. For your convenience, we summarize the corresponding setups and key findings below:
>
> ---
> ***Case 1 (Weak dependence)**: Assume that $\mathbf{X}_1$ exhibits weak dependence on  $\mathbf{X}_2$ in the following sense: $\mathbf{X}_1=\alpha\mathbf{X}_2+\widetilde{\mathbf{X}}_1 $, where $\alpha\to 0$ as $M\to \infty$, and $\widetilde{\mathbf{X}}_1$ is independent of $\mathbf{X}_2$ with the form $\widetilde{\mathbf{X}}_1=\mathbf{\Sigma}_1^{1/2}\mathbf{Z}_1$. It follows that $\mathrm{Cov}(\mathbf{x}_j^{(1)},\mathbf{x}_j^{(2)} )=\mathrm{Cov}(\mathbf{x}_j^{(2)},\alpha\mathbf{x}_j^{(2)} )=\alpha\mathbf{\Sigma}_2$. We refer to this regime as weak dependence between domain covariates.*
>
> ***Proposition 1** Suppose that $\widetilde{\mathbf{X}}_{1},\mathbf{X}_2,\boldsymbol{\varepsilon}_1$ and $\boldsymbol{\varepsilon}_2$ satisfy Assumptions 1-2. Then, Theorem 1 continues to hold. Moreover, if we additionally impose Assumption 3, then Theorem 2  remains valid.*
>
>
> ---

---

> ### Author Response · Authors · 2025-11-21
>
> ---
> ***Case 2 (Strong dependence):** Suppose the matrix $\mathbf{X}_1$ follows a signal-plus-noise structure: $\mathbf{X}_1=\mathbf{X}_2+\mathbf{A}$, where $\mathbf{A}$ is a deterministic signal matrix satisfying $\\|\mathbf{A}\\|=o(\sqrt{M}).$ This model captures realistic domain adaptation scenarios in which the source and target domains share a common underlying data matrix but differ by a small deterministic shift -- such as a weak shared signal across features in source domain.*
>
> ***Proposition 3**  Suppose $\mathbf{X}_1=\mathbf{X}_2+\mathbf{A}$ is a signal-plus-noise data matrix, with $\\|\mathbf{A}\\|=o(\sqrt{M})$. The regression parameter vector $\boldsymbol{\beta}=\boldsymbol{\beta}\_1=\boldsymbol{\beta}\_2$  satisfies Assumption 3.
> When $\lambda\_{\mathsf{s}}\neq \lambda\_{\mathsf{t}}$, we have*
> $$
> \widehat{\mathbf{Bias}}=\frac{a}{M}\mathrm{Tr}\mathbf{\Sigma}\_2[\mathbf{\Pi}\_2(-\lambda\_{\mathsf{t}} )-\mathbf{\Pi}\_2(-\lambda\_{\mathsf{s}})]+\frac{b}{M}\mathrm{Tr}\mathbf{\Sigma}\_2\mathbf{\Pi}\_2'(-\lambda\_{\mathsf{t}})+\frac{c}{M}\mathrm{Tr}\mathbf{\Sigma}\_2\mathbf{\Pi}\_2'(-\lambda\_{\mathsf{s}}),
> $$
> *and*
> $$
> \widehat{\mathbf{Var}}=\frac{\xi^2\sigma^2}{N\_1}\bigg(d\mathrm{Tr}\mathbf{\Sigma}\_2[\mathbf{\Pi}\_2(-\lambda\_{\mathsf{t}})-\mathbf{\Pi}\_2(-\lambda\_{\mathsf{s}})]+e\mathrm{Tr}\mathbf{\Sigma}\_2\mathbf{\Pi}\_2'(-\lambda\_{\mathsf{t}})+f\mathrm{Tr}\mathbf{\Sigma}\_2\mathbf{\Pi}\_2'(-\lambda\_{\mathsf{s}})\bigg)\\
> +(1-\xi)^2\sigma^2\frac{1}{N\_2}\mathrm{Tr}\mathbf{\Sigma}\_2[\mathbf{\Pi}\_2(-\lambda\_{\mathsf{s}})-\lambda\_{\mathsf{s}}\mathbf{\Pi}\_2'(-\lambda\_{\mathsf{s}})],
> $$
> *where*
> $$
> a=\frac{2\xi \lambda\_1\lambda\_2}{\lambda\_{\mathsf{s}}-\lambda\_{\mathsf{t}} }+\frac{2\xi\lambda\_\mathsf{t}\lambda\_{\mathsf{s}}( \xi\lambda\_{\mathsf{\_{\mathsf{t}}}}-\lambda\_{\mathsf{s}} ) }{(\lambda\_{\mathsf{s}}-\lambda\_{\mathsf{t}})^2}-\frac{2\xi\lambda\_{\mathsf{s}}^2\lambda\_{\mathsf{t}}^2 }{(\lambda\_{\mathsf{s}}-\lambda\_{\mathsf{t}})^3}, \;b=\xi^2\lambda\_{\mathsf{t}}^2-\frac{2\xi^2\lambda\_{\mathsf{t}}^2\lambda\_{\mathsf{s}} }{\lambda\_{\mathsf{t}}-\lambda\_{\mathsf{s}}},
> $$
>
> $$
>  c=\lambda_{\mathsf{s}}^2-\frac{2\xi \lambda_{\mathsf{t}}\lambda_{\mathsf{s}}^2}{\lambda_{\mathsf{t}}-\lambda_{\mathbf{s}}},\; d=\frac{2\lambda_{\mathsf{s}}}{\lambda_{\mathsf{t}}-\lambda_{\mathsf{s}}}+\frac{\lambda_{\mathsf{s}}^2}{(\lambda_{\mathsf{t}}-\lambda_{\mathsf{s}})^2}+\frac{\lambda_{\mathsf{t}}\lambda_{\mathsf{s}}^2}{(\lambda_{\mathsf{s}}-\lambda_{\mathsf{t}})^3}+\frac{2\lambda_{\mathsf{s}}\lambda_{\mathsf{t}}}{(\lambda_{\mathsf{t}}-\lambda_{\mathsf{s}})^2},
> $$
> $$
> e=-\lambda_{\mathsf{t}}+ \frac{2\lambda_{\mathsf{s}}\lambda_{\mathsf{t}}}{(\lambda_{\mathsf{t}}-\lambda_{\mathsf{s}})^2}-\frac{\lambda_{\mathsf{t}}\lambda_{\mathsf{s}}^2}{(\lambda_{\mathsf{s}}-\lambda_{\mathsf{t}})^2},\quad f=\frac{\lambda_{\mathsf{s}}^2}{\lambda_{\mathsf{s}}-\lambda_{\mathsf{t}}}-\frac{\lambda_{\mathsf{t}}\lambda_{\mathsf{s}}^2}{(\lambda_{\mathsf{s}}-\lambda_{\mathsf{t}})^2}.
> $$
> *When $\lambda=\lambda_{\mathsf{s}}=\lambda_{\mathsf{t}}$,  $\widehat{\mathbf{Bias}}$ is given in (54) and*
> $$
> \widehat{\mathbf{Var}}=(1-\xi)^2\sigma^2\frac{1}{N\_2}\mathrm{Tr}\mathbf{\Sigma}\_2(\mathbf{\Pi}\_2+\lambda\mathbf{\Pi}\_2')+\frac{\xi^2\sigma^2}{N\_1}\mathrm{Tr}\mathbf{\Sigma}\_2[\mathbf{\Pi}\_2-3\lambda\mathbf{\Pi}\_2'+3\lambda^2\mathbf{\Pi}\_2^{(2)}-\lambda^3\mathbf{\Pi}\_2^{(3)}],
> $$
> *with* $\mathbf{\Pi}\_2^{(k)}=\frac{\mathrm{d}^k\mathbf{\Pi}\_2(z)}{\mathrm{d}z^k}\big|\_{z=-\lambda}.$
>
> ---

---

> ### Author Response · Authors · 2025-11-21
>
> + **Concern 2: Moment conditions**. In the original manuscript, we assumed that the entries of $\mathbf{Z}_i$ (where $\mathbf{X}_i=\mathbf{\Sigma}_i^{1/2}\mathbf{Z}_i$) have moments of all orders -- mainly for technical convenience and to simplify the exposition. However, as noted after Assumption 1, this requirement can be relaxed.  In particular, our analysis only necessitates the existence of moments of order
> $(8+c)$ for any positive constant $c>0$. We have replaced Lemma 5 in the original manuscript with Lemma 6 in the revised version to ensure applicability under the weaker moment condition.   To enhance clarity, we have now explicitly stated this relaxed moment condition following Assumption 1 in the revised manuscript (highlighted in red). Furthermore, at the end of Appendix B.1, we have added a new Remark detailing how the moment assumption can be relaxed. For your convenience, we  reproduce the remark below:
>
> ---
>
> ***Remark**
> To relax the moment assumption, we apply a standard truncation argument commonly used in random matrix theory (e.g., [1]).  This approach allows us to employ Lemma 4 under the weaker finite $(8+c)$-th moment condition, introducing only a negligible additional error term that depends on $M$ but does not affect the leading-order asymptotics of our results.
> Moreover, a careful examination of the proofs shows that the same moment condition is also sufficient to establish Lemma
> 7. Consequently, all of our theoretical conclusions remain valid under this relaxed assumption.*
>
> ---
> [1] Yang F et al. ``Precise high-dimensional asymptotics for quantifying heterogeneous transfers''. JMLR 2025.
>
>
> + **Concern 3: concentration assumptions.** We would like to clarify a possible misunderstanding. Our analysis does not assume any concentration properties as external conditions. Instead, the concentration inequalities used in the paper (Lemma 3 and Lemma 5 in the revised manuscript, which correspond to Lemma 2 and Lemma 4 in the original submission, respectively) are rigorously derived within our framework and can be proved directly.

---

### Official Review · Reviewer_aTNu · 2025-11-01

**Soundness:** 3
**Presentation:** 3
**Contribution:** 3
**Rating:** 6
**Confidence:** 4

**Summary:**

The paper analyzes cross-domain KD for linear regression with ridge (and ridgeless) estimators. A source-domain teacher (\beta_t) is trained on ((X_1,y_1)) and used to supervise a target-domain student on ((X_2,y_2)) via the imitation parameter (\xi) in the objective (L(\xi)=\xi \ell(y_2^t,y_2^s)+(1-\xi)\ell(y_2,y_2^s)). Closed-form resolvent-based expressions yield high-dimensional (HD) bias–variance formulas for the excess target risk (ER(\beta_s)) under covariate shift ((\Sigma_1\neq\Sigma_2)) and model shift ((\beta_1\neq \beta_2)). The paper also treats random-(\beta) and an under-parameterized ridgeless regime. Key theorems (1–3) give deterministic equivalents that expose dependence on (\Sigma_1,\Sigma_2,\beta_1,\beta_2), and on (\xi), and show conditions where the student beats the student-only baseline; (\xi) can even be negative (“anti-learning”).

**Strengths:**

1. In the under-parametrized, ridgeless setting you show the student is a convex combination of the two OLS estimators and give a closed-form (ER) with optimal (\xi\in(0,1)). This is intuitive and useful.
 **Allowing (\xi\in\mathbb{R}).** You justify that negative (\xi) can be optimal (isotropic corollary), and you prove existence of a strictly better (\xi) than the student-only baseline under natural conditions.

2. The paper formalizes (ER=\text{Bias}+\text{Var}) and then supplies deterministic equivalents for both parts under deterministic ((\beta_1,\beta_2)) and random (\beta) models (Theorems 1–2).

**Weaknesses:**

1. Because both squared losses share the same quadratic term in (\beta), the Hessian of (L(\xi)) is (N_2^{-1}X_2X_2^\top+\lambda_s I), **independent of (\xi)**; hence the problem is convex and well-posed for any real (\xi). It would help readers to add a short lemma right after (1) and your closed form for (\beta_s), making the “negative (\xi) is still safe” point explicit.

2. Expressions like Theorem 1’s Bias/Var have (o_{\text{a.s.}}(1)) terms while the main terms scale with traces ((\Theta(M)) in isotropic cases). Please specify whether your (o_{\text{a.s.}}(1)) is *absolute* or *per-dimension*, and consider normalizing (ER/M) in statements to make asymptotic orders unambiguous.

3.  Assumption 1(a) currently asks for all moments; you note it can be relaxed. Give a concrete bound ((4+\epsilon) or (8) moments) sufficient for Lemma 6 / local laws used later, so readers know what’s truly required.

4. Theorem 1 already shows dependence on (\Sigma_1,\Sigma_2) via (\Pi_1,\Pi_2,S_i(\cdot)). It would help to rewrite one or two key trace terms (e.g., ( \mathrm{Tr},[\Pi_1 \Pi_2 \Sigma_2])) in the eigen-bases of (\Sigma_1,\Sigma_2), or to use an overlap matrix to highlight principal-angle effects. You partly do this in App. B.6 (eq. (28)); elevating a compact “eigenvector-overlap” corollary to the main text would greatly aid intuition.

5. You note (and Proposition 1 leverages) that (ER(\beta_s)) is a convex quadratic in (\xi). Please collect coefficients (A,B,C) in closed form (from Theorems 1–2) and give a short table for common regimes (isotropic; shared (\beta); pure model shift). This immediately yields (\xi^\star!=!-B/2A) and clarifies when (\xi^\star<0) or (\xi^\star>1) without case-by-case reasoning.

6. Your analysis assumes/advocates Bayes-consistent teacher probabilities and studies how teacher quality controls the SGD variance term. Ye et al.[1] propose BCDE, a teacher-training objective based on conditional mutual information explicitly aimed at estimating the Bayes conditional distribution for KD. It’s a natural methodological precursor/neighbor to your “Bayesian teacher” prescription and directly relevant to your noise model and guidelines.

[1] Ye, Linfeng, et al. “Bayes Conditional Distribution Estimation for Knowledge Distillation Based on Conditional Mutual Information.” ICLR 2024 (Twelfth).

**Questions:**

see the weakness above

---

> ### Author Response · Authors · 2025-11-21
>
> We sincerely thank the reviewer for taking the time to provide  constructive feedback on our work. We greatly appreciate your recognition that our results on ridgeless regression are intuitive and useful. Below, we provide  point-by-point responses to your questions.
> >Because both squared losses share the same quadratic term in ($\beta$), the Hessian of ($L(\xi)$) is ($N_2^{-1}X_2X_2^\top+\lambda_s I$), independent of ($\xi$); hence the problem is convex and well-posed for any real ($\xi$). It would help readers to add a short lemma right after (1) and your closed form for ($\beta_s$), making the “negative ($\xi$) is still safe” point explicit.
>
> In response, we have added an informal lemma immediately after equation (1) to highlight the key insight, along with a clarifying sentence  following the closed-form expression of $\beta_s$ in equation (3).
>   Both additions are highlighted in red in the revised manuscript. For your convenience, we reproduce them below:
>
> The informal lemma:
> *Lemma 1. (informal)    Under mild conditions, the excess risk in linear regression with quadratic loss admits a unique minimizer $\xi^\*$, which can be negative.*
>
>
> The clarifying sentence:
> *From equation 3, the parameter $\xi$ is independent of $\mathbf{Q}_2$, making it possible to choose a negative $\xi$ that achieves better generalization performance.*
>
> >Expressions like Theorem 1’s Bias/Var have $(o_{\text{a.s.}}(1))$ terms while the main terms scale with traces (($\Theta(M)$) in isotropic cases). Please specify whether your ($o_{\text{a.s.}}(1)$) is absolute or per-dimension, and consider normalizing (ER/M) in statements to make asymptotic orders unambiguous.
>
> Thanks for this question.
> As stated in the Notation section (Section 1.2), we use the standard definition: $x_n=o_{a.s.}(a_n)$, if $x_n/a_n\to 0$ almost surely. In our results, the excess risk (**ER**), bias, and variance are all **scalar quantities**, so the $o_{a.s.}(1) $ terms are absolute. Moreover, all asymptotic expressions we provide for **ER, Bias, and Var** are of order $O(1)$ -- not $\Theta(M)$, as $M\to \infty.$ Therefore, normalization by $M$ (e.g., using **ER**/M) would not accurately represent the true scale of the risk.
>
> In response to your suggestion to reduce ambiguity, we have made this clarification explicit at the end of Section 2.2 in the revised manuscript (highlighted in red for ease of review). The added sentence reads as follows:
>
> *One may easily check that $\mathbf{ER}=O(1)$ almost surely.*
>
> > Assumption 1(a) currently asks for all moments; you note it can be relaxed. Give a concrete bound (($4+\epsilon$) or (8) moments) sufficient for Lemma 6 / local laws used later, so readers know what’s truly required.
>
> We appreciate the reviewer’s suggestion to clarify the moment conditions required for our analysis. In response, we have  explicitly stated, immediately after **Assumption 1** (highlighted in red for ease of review), that the existence of moments of order $(8+c)$ for any positive $c>0$ is sufficient to ensure the validity of our theoretical results. We have replaced Lemma 5 in the original manuscript with Lemma 6 in the revised version to ensure applicability under the weaker moment condition. In addition, we have added a remark at the end of
> Appendix B.1 to explain how the original moment assumption can be relaxed to this weaker condition. For your convenience, we reproduce the relevant revision below:
>
> **Remark.** *To relax the moment assumption, we apply a standard truncation argument commonly used in random matrix theory (e.g., [1]).  This approach allows us to employ Lemma 4 under the weaker finite $(8+c)$-th moment condition, introducing only a negligible additional error term that depends on $M$ but does not affect the leading-order asymptotics of our results.
> Moreover, a careful examination of the proofs shows that the same moment condition is also sufficient to establish Lemma
> 7. Consequently, all of our theoretical conclusions remain valid under this relaxed assumption.*
>
>
> >Theorem 1 already shows dependence on ($\Sigma_1,\Sigma_2$) via ($\Pi_1,\Pi_2,S_i(\cdot)$). It would help to rewrite one or two key trace terms (e.g., $( \mathrm{Tr},[\Pi_1 \Pi_2 \Sigma_2])$) in the eigen-bases of ($\Sigma_1,\Sigma_2$), or to use an overlap matrix to highlight principal-angle effects. You partly do this in App. B.6 (eq. (28)); elevating a compact “eigenvector-overlap” corollary to the main text would greatly aid intuition.
>
> We thank the reviewer for this helpful suggestion. In response, we have added an illustrative example immediately following **Theorem 1** in the revised manuscript (highlighted in red). The matrix $\mathbf{E}_5=\mathbf{\Sigma}_2\mathbf{\Pi}_2$ is introduced to simplify notation. For your convenience, we reproduce the newly added text below:

---

> ### Author Response · Authors · 2025-11-21
>
> *We provide an illustrative example here. Suppose that  $\mathbf{\Sigma}_i$ admits the spectral decomposition $\mathbf{\Sigma}\_i=\mathbf{U}\_i\mathbf{\Lambda}\_i\mathbf{U}\_i^{\mathsf{T}} $, for $i=1,2$. Consider the term $\boldsymbol{\beta}\_1^{\mathsf{T}}\mathbf{\Pi}\_1\mathbf{E}\_5\boldsymbol{\beta}\_2 $, which can be expressed as*
>
> $$
> \boldsymbol{\beta}\_1^{\mathsf{T}}(\lambda\_{\mathsf{t}}+\lambda\_{\mathsf{t}}m\_1\mathbf{\Sigma}\_1)^{-1}(\lambda\_{\mathsf{s}}+\lambda\_{\mathsf{s}}m\_2\mathbf{\Sigma}\_2 )^{-1}\mathbf{\Sigma}\_2\boldsymbol{\beta}\_2=(\lambda\_{\mathsf{s}}\lambda\_{\mathsf{t}})^{-1}\tilde{\boldsymbol{\beta}}\_1^{\mathsf{T}}(1+m\_1\mathbf{\Lambda}\_1)^{-1} \mathbf{U}\_1^{\mathsf{T}}\mathbf{U}\_2\tilde{\mathbf{\Lambda}}\_2\tilde{\boldsymbol{\beta}}\_2,\quad (1)
> $$
> *where $\tilde{\mathbf{\Lambda}}\_2$ is a diagonal matrix with entries $\tilde{\mathbf{\Lambda}}\_{2,jj}=\frac{\mathbf{\Lambda}\_{2,jj} }{1+m\_2\mathbf{\Lambda}\_{2,jj}}$. The vector $\tilde{\boldsymbol{\beta}}\_i=\mathbf{U}\_i\boldsymbol{\beta}\_i$ captures the alignment between $\boldsymbol{\beta}_i$ and the eigenvectors of $\mathbf{\Sigma}\_i$.
> The right-hand side of  (1)  explicitly reveals how the term
> $\boldsymbol{\beta}\_1^{\mathsf{T}}\mathbf{\Pi}\_1\mathbf{E}\_5\boldsymbol{\beta}\_2$  depends on $\tilde{\boldsymbol{\beta}}\_i$, the eigenvalues of $\mathbf{\Sigma}_1$ and $\mathbf{\Sigma}_2$, and the eigenvector overlap  $\mathbf{U}_1^{\mathsf{T}} \mathbf{U}_2$ between the two covariance matrices. In the special case where each $\boldsymbol{\beta}\_i$ is aligned with an eigenvector of $\mathbf{\Sigma}\_i$ -- for simplicity, suppose it corresponds to the first eigenvector -- the expression (1) further simplifies to*
>
> $$ \boldsymbol{\beta}\_1^{\mathsf{T}}\boldsymbol{\beta}\_2(\lambda\_{\mathsf{s}}\lambda\_{\mathsf{t}})^{-1}\frac{\mathbf{\Lambda}\_{2,11}}{(1+m\_1\mathbf{\Lambda}\_{1,11})(1+m_2\mathbf{\Lambda}\_{2,11})},
> $$
> *which depends on the eigenvalues of $\mathbf{\Sigma}_i$ and the  inner product $\boldsymbol{\beta}_1^{\mathsf{T}}\boldsymbol{\beta}_2$.*
>
> This dependence becomes even more explicit in the under-parameterized ridgeless regression regime, as demonstrated in Corollary 3 of the revised manuscript (formerly Theorem 3).
>
> >You note (and Proposition 1 leverages) that (ER($\beta_s$)) is a convex quadratic in ($\xi$). Please collect coefficients (A,B,C) in closed form (from Theorems 1–2) and give a short table for common regimes (isotropic; shared ($\beta$); pure model shift). This immediately yields ($\xi^\star!=!-B/2A$) and clarifies when ($\xi^\star<0$) or ($\xi^\star>1$) without case-by-case reasoning.
>
> In the revised manuscript, we provide closed-form expressions for the optimal value $ \xi^*
>  =-B/2A$ under several common settings. The derivations and the corresponding optimal  $\xi^*$ values
>   are presented in Appendix B.7. For your convenience, we reproduce the relevant text below:
>
> *Note that, up to asymptotically negligible terms, $\mathbf{ER}$ can be expressed as a quadratic function of $\xi:$ $\mathbf{ER}(\boldsymbol{\beta}_{\mathsf{s}},\xi)=A\xi^2+B\xi+C$.
> Below we provide closed-form expressions for the asymptotic optimal $\xi^\*=-\frac{B}{2A}$ under several common settings.*
>
> *(1) When $\boldsymbol{\gamma}=\boldsymbol{\beta}_1-\boldsymbol{\beta}_2,\frac{M}{N_1},\frac{M}{N_2}<1-\tau$*,
>
> $$
> \xi^\*=\bigg(\boldsymbol{\gamma}^{\mathsf{T}}\mathbf{\Sigma}_2\boldsymbol{\gamma}+\sigma^2\frac{M}{N_2-M}+\frac{\sigma^2}{N_1-M}\mathrm{Tr}\mathbf{\Sigma}_2\mathbf{\Sigma}_1^{-1} \bigg)^{-1} \frac{\sigma^2M}{N_2-M}\in (0,1) .
> $$
>
> *(2) When $\boldsymbol{\beta}=\boldsymbol{\beta}\_1=\boldsymbol{\beta}\_2$ is random, and* $\mathbf{\Sigma}_1=\mathbf{\Sigma}_2=\mathbf{I}_M,$
>
> $$
> \xi^*={ \frac{
>     \displaystyle \frac{M}{N_2}\left( \underline{m}_2 - \lambda_s\underline{m}_2' \right) - \frac{\tilde{\sigma}^2}{\sigma^2}\lambda_t\lambda_s\left( \underline{m}_1\underline{m}_2 - \lambda_s \underline{m}_2'\underline{m}_1 \right)
> }{A_1+A_2+A_3
> }},
> $$
> where
> $$
> A_1=\displaystyle \frac{\tilde{\sigma}^2}{\sigma^2}\left( \lambda_t^2 \underline{m}_1' - 2\lambda_t^2\lambda_s \underline{m}_2\underline{m}_1' + \lambda_t^2\lambda_s^2 \underline{m}_1'\underline{m}_2' \right),
> $$
> $$
> A_2 = \frac{M}{N_1}\left( \underline{m}_1 - 2\lambda_s\underline{m}_1\underline{m}_2 + \lambda_s^2\underline{m}_1\underline{m}_2' - \lambda_t\underline{m}_1' + 2\lambda_t\lambda_s\underline{m}_2\underline{m}_1' - \lambda_t\lambda_s^2\underline{m}_1'\underline{m}_2' \right),
> $$
> $$
>  A_3= \frac{M}{N_2}\left( \underline{m}_2 - \lambda_s\underline{m}_2' \right).
> $$

---

> ### Author Response · Authors · 2025-11-21
>
> *(3) When* $\boldsymbol{\beta}=\boldsymbol{\beta}\_1=\boldsymbol{\beta}\_2,\mathbf{\Sigma}\_2=\mathbf{I}\_M$,
>
> $$
> \xi^\*=\frac{\boldsymbol{\beta}^{\mathsf{T}}[\lambda\_{\mathsf{t}}\lambda\_{\mathsf{s}}^2\underline{m}\_2'\mathbf{\Pi}\_1-\lambda\_{\mathsf{t}}\lambda\_{\mathsf{s}}\underline{m}\_2\mathbf{\Pi}\_1]\boldsymbol{\beta}+\frac{\sigma^2M}{N\_2}(\underline{m}\_2'-\lambda\_{\mathsf{s}}\underline{m}\_2') }{\boldsymbol{\beta}^{\mathsf{T}}\lambda\_{\mathsf{t}}^2[1+\lambda\_{\mathsf{s}}^2\underline{m}\_2'-2\lambda\_{\mathsf{s}}\underline{m}\_2 ]\mathbf{\Pi}\_1'\boldsymbol{\beta}+\frac{\sigma^2(1-2\lambda\_{\mathsf{s}}\underline{m}\_2+\lambda\_{\mathsf{s}}^2\underline{m}\_2')}{N\_1}\operatorname{Tr}[\mathbf{\Pi}\_1-\lambda\_{\mathsf{t}}\mathbf{\Pi}\_1']+\frac{\sigma^2M}{N\_2}(\underline{m}\_2-\lambda\_{\mathsf{s}}\underline{m}\_2') }.
> $$
>
> >Your analysis assumes/advocates Bayes-consistent teacher probabilities and studies how teacher quality controls the SGD variance term. Ye et al.[1] propose BCDE, a teacher-training objective based on conditional mutual information explicitly aimed at estimating the Bayes conditional distribution for KD. It’s a natural methodological precursor/neighbor to your “Bayesian teacher” prescription and directly relevant to your noise model and guidelines.
>
> After reading the reference you recommended, we find that the MCMI method proposed by Ye et al. assists teachers in cross-domain knowledge distillation generate more accurate estimates of the Bayes conditional probability distribution (BCPD) by capturing richer contextual information. In contrast, our work shows that properly choosing $\xi$ -- even negatively -- can optimally leverage teacher model.  Therefore it is a relevant reference to our work, and we have added a citation to Ye et al. at the end of Page 2 in the revised manuscript. For ease of review, the newly added content (highlighted in red in the revised manuscript) is reproduced below:
>
> *[2] proposed the Maximum Conditional Mutual Information method, which enables the teacher model to capture more contextual information and generate more accurate estimates of the Bayes conditional probability distribution.*
>
> [1] Yang F et al. ``Precise high-dimensional
> asymptotics for quantifying heterogeneous transfers''. JMLR 2025.
>
> [2] Ye, Linfeng, et al. “Bayes Conditional Distribution Estimation for Knowledge Distillation Based on Conditional Mutual Information.” ICLR 2024 (Twelfth).

---

### Official Review · Reviewer_AThT · 2025-11-07

**Soundness:** 2
**Presentation:** 3
**Contribution:** 3
**Rating:** 4
**Confidence:** 2

**Summary:**

This paper studies cross-domain knowledge distillation with domain shift through a teacher-student framework. Authors leverage Random Matrix Theory and present a theoretical analysis in the context of linear regression, comprising a deterministic-parameter setting where the teacher and student parameters are non-random and a random-parameter setting where a shared parameter vector is drawn from prior distributions. This paper discovers that the knowledge distillation still works even under substantial domain discrepancies. Authors also observe a double-descent phenomenon in the knowledge distillation process.

**Strengths:**

- The overall theoretical analysis is clear and well-organized. Authors use tools from Random Matrix Theory and derive precise, high-dimensional asymptotic expressions for the excess risk. The analysis is mathematically sound.
- Interesting phenomenon that knowledge distillation is still possible when the source domain and target domain share substantial domain discrepancies. The Anti-Learning discovery is also insightful, which points out that the best imitation parameter $\xi$ is not limited to the [0, 1] range.

**Weaknesses:**

- All the derivations in this paper only work for linear regression. In the real world, complex, non-linear models like deep neural networks are much more popular. Therefore, it's unknown whether these insights remain applicable in practice and how much they can provide guidance for Knowledge Distillation.
- Lack of Quantification for the extent of the Substantial Shift. The paper claims efficacy even under substantial domain discrepancies, but the degree of "substantial" is not well quantified. The analysis shows that an optimal $\xi$ exists such that $\text{ER}(\beta_s) < \text{ER}_0$, but how does this performance gain degrade as a function of the domain shift? A potential weakness is that the paper does not introduce an explicit metric to quantify domain shift. The discrepancy between domains is only implicitly reflected through covariance geometry.
- The finding that $\xi < 0$ (anti-learning) can be optimal (Corollary 2) is intriguing but seems rather uncommon in practical scenarios. Does this imply that the teacher performs so poorly that the student benefits from learning the opposite of the guidance? A more in-depth discussion on the underlying reasons for this phenomenon and its practical implications would further strengthen the paper.

**Questions:**

Please see the weakness.

---

> ### Author Response · Authors · 2025-11-21
>
> We sincerely thank you for your thoughtful and constructive feedback. We greatly appreciate your recognition of this non-intuitive finding and its implications for KD under domain shift. We have addressed your concerns as follows:
>
> >Q1. All the derivations in this paper only work for linear regression. In the real world, complex, non-linear models like deep neural networks are much more popular. Therefore, it's unknown whether these insights remain applicable in practice and how much they can provide guidance for Knowledge Distillation.
>
> We thank the reviewer for this valuable suggestion.
> While our theoretical analysis is developed in the linear regression setting, we  anticipate that the key phenomena revealed by our theory may extend beyond the linear setting. To support this claim, we have added a new Section 5.1 describing the experimental setup for nonlinear models. The corresponding empirical results are presented in Figure 4 of Appendix C.1 in the revised manuscript. These simulations provide strong evidence that the mechanisms identified in the linear regime are robust and likely to generalize to more complex nonlinear settings. For your convenience, we summarize the experimental setup below; this same description appears in the newly added Section 5.
>
> ---
> *We assume the source data $\\{(\mathbf{x}_j^{(1)},{y}_j^{(1)} )\\}\_{j=1}^{N_1} $ are generated i.i.d. according to*
>
> \\[
> y_j^{(1)} =f(\mathbf{x}_j^{(1)} )+\varepsilon^{(1)}_j,\text{ for } 1\le j\le N_1.
> \\]
>
> *The target data $\{(\mathbf{x}_j^{(2)},y_j^{(2)})\} \_{j=1}^{N_2} $ are generated according to*
> \\[
> y_j^{(2)} =\tilde{f}(\mathbf{x}_j^{(2)} )+\varepsilon_j^{(2)},\text{ for } 1\le j\le N_2
> \\]
> *Suppose $\mathbf{x}_j^{(1)}\sim D_1,1\le j\le N_1$ and $\mathbf{x}_j^{(2)}\sim D_2,1\le j\le N_2. $ We refer to the case $D_1\neq D_2$ as a covariate shift, and the case where $f\neq \tilde{f}$ as a model shift.*
>
> *We consider learning the unknown function using a fully connected two-layer neural network with $n$ hidden neurons:
> $f_{\mathrm{NN}}(\mathbf{x})=\mathbf{a}^{\mathsf{T}}\sigma(\mathbf{Wx}),$ where $\mathbf{W}\in \mathbb{R}^{n\times M} $ is the  weight matrix, and $\sigma(\cdot)$ is an activation function applied entrywise.
> When the random weight matrix $\mathbf{W}$ is fixed and only the second-layer weight $\mathbf{a}$ is  optimized, the model reduces to a kernel regression model, where the kernel defined by $\mathbf{x}\to \sigma(\mathbf{Wx})$ is referred to as the conjugate kernel. The teacher model is given by*
>
> \\[
> f\_{\mathrm{NN}}^{\mathsf{t}}(\mathbf{x})=\mathbf{a}\_\mathsf{t}^{\mathsf{T}}\sigma(\widetilde{\mathbf{W}}\_1\mathbf{x}),
> \\]
> *with*
> $$
> \mathbf{a}\_{\mathsf{t}}=\arg\min\_{\mathbf{a}}\bigg\\{\frac{1}{N\_1}\\|\mathbf{y}\_1-\sigma(\mathbf{X}\_1^{\mathsf{T}}\widetilde{\mathbf{W}}\_1^{\mathsf{T}}) \mathbf{a}\\|^2+\lambda\_{\mathsf{t}} \\|\mathbf{a}\\|^2  \bigg\\}.
> $$
>
> *We use $f\_{\mathrm{NN}}^{\mathsf{t}}(\mathbf{x}\_j^{(2)})$ together with the covariates $\\{\mathbf{x}\_j^{(2)}\\}\_{j=1}^{N\_2}$ to generate predictions $\mathbf{y}\_2^{\mathsf{t}}$. Then the student model is finetuned on the target domain data and $\mathbf{y}\_2^{\mathsf{t}}$. The student model takes the form
> $f\_{\mathrm{NN}}^{\mathsf{s}}(\mathbf{x})=\mathbf{a}\_{\mathsf{s}}^{\mathsf{T}}\sigma(\mathbf{W}\_1\mathbf{x})$
> with*
> $$
> \mathbf{a}\_{\mathsf{s}}=\arg\min\_{\mathbf{a}}\xi\bigg(\frac{1}{N\_2} \\| \mathbf{y}\_2^{\mathsf{t}}-\sigma(\mathbf{X}\_2^{\mathsf{T}}\mathbf{W}\_1^{\mathsf{T}} )\mathbf{a} \\|^2 \bigg)+(1-\xi)\bigg( \frac{1}{N\_2} \\|\mathbf{y}_2-\sigma(\mathbf{X}\_2^{\mathsf{T}}\mathbf{W}\_1^{\mathsf{T}} )\mathbf{a} \\|^2\bigg)+\lambda\_{\mathsf{s}} \\|\mathbf{a} \\|^2.
> $$
> *We also examine a setting where the teacher model is a deeper neural network. Specifically, while keeping the student model fixed, we let the teacher be a 4-layer fully connected network:*
>
> $$
> f\_{\mathrm{NN}}^{\mathsf{t}}=\mathbf{a}\_{\mathsf{t}}^{\mathsf{T}}\sigma(\widetilde{\mathbf{W}}\_3\sigma(\widetilde{\mathbf{W}}\_2\sigma(\widetilde{\mathbf{W}}\_1\mathbf{x}))),
> $$
>
> *where*
>
> $$
> \mathbf{a}\_{\mathsf{t}}=\arg\min\_{\mathbf{a}}\frac{1}{N\_1}\\|\mathbf{y}_1-[\sigma(\widetilde{\mathbf{W}}\_3\sigma(\widetilde{\mathbf{W}}\_2\sigma(\widetilde{\mathbf{W}}\_1\mathbf{X}\_1))]^{\mathsf{T}}\mathbf{a}\\|^2+\lambda\_{\mathsf{t}}\\|\mathbf{a}\\|^2.
> $$
>
> *We set $D_1=\mathcal{N}(0,4\mathbf{I}_M)$ and $D_2=\mathcal{N}(0,\mathbf{I}_M)$. Let
> $$
> f(\mathbf{x})= (\boldsymbol{\beta}^{\mathsf{T}}\mathbf{x} )^2+1,\quad \tilde{f}(\mathbf{x})=(\boldsymbol{\beta}^{\mathsf{T}}\mathbf{x})^2.
> $$
> Because $D_1\neq D_2$ and $f\neq \tilde{f}$, both covariate shift and model shift are present in this setting. More details and the
> numerical results are provided in Appendix C.1.*
>
> ---

---

> ### Author Response · Authors · 2025-11-21
>
> Moreover, our analysis reveals a potentially counterintuitive phenomenon: the optimal imitation parameter $\xi$ may be negative. In contrast, standard KD practice typically restricts $\xi$ to the interval [0,1], implicitly assuming that the teacher's outputs are always beneficial. This discrepancy implies a **practical guideline: allowing $\xi<0$ may lead to better generalization,** especially under domain shift. To demonstrate this point, we conduct experiments on the Human Activity Recognition dataset (available at https://archive.ics.uci.edu/dataset/240/human+activity+recognition+using+smartphones).
> This dataset contains 561 dimensional sensor signals collected from 30 individuals performing various activities.  We are interested in a binary classification task, where the **Active** class includes walking, walking upstairs, walking downstairs, and standing, while the **Rest** class includes sitting and laying.
>
> We treat the data from Subject 25 (the largest individual dataset) as the target domain. The target dataset is divided into a training set (the last 70\%, containing 286 samples) and a testing set (the first 30\%, containing 123 samples). The remaining 29 subjects (a total of 9,890 samples) constitute the source domain.  We set $\lambda\_t=\lambda\_s=0.1$. Denote the training data by $( {X}\_2^{\mathsf{T}}, {y})$ and the test data by $(X\_{t,2}^{\mathsf{T}} ,y\_t )$, where $ {X}\_2=( {x}\_1,..., {x}\_{N\_2})$, $X\_{t,2}=(x\_{t,1},...,x_{t,n\_2})$ with $N\_2=286, n\_2=123$.  To obtain soft labels in [0,1], we apply the sigmoid function to teacher's predictions:
> $$ {y}\_2^{{t}}=\text{sigmoid}( {\beta}\_{t}^{\mathsf{T}}  {X}_{2} ),$$
> where the sigmoid function is applied entrywise.
> The student estimator $ {\beta}\_{\mathsf{s}}$ then takes the form:
> $$
>  {\beta}\_{s}=\frac{1}{N_2} {Q}\_2(\xi {X}\_2 {y}_2^{t}+(1-\xi) {X}\_2 {y} ).
> $$
> For each test sample, we apply the sigmoid function to the predicted output and assign label 1 if the value exceeds 0.5, and 0 otherwise.
>
> The results for different values of $\xi$ are summarized in Table 1 below, which indicate that **anti-learning against the teacher’s supervision $(\xi<0)$ may lead
> to better generalization ability**. Remarkably, the student achieves perfect classification accuracy when $\xi=-1$ or $\xi=-0.849$.  This finding provides empirical evidence supporting the necessity of optimizing $\xi$ over $\mathbb{R}$.
>
> Table 1: Test accuracy vs. imitation parameter $\xi$.
>
> | $\xi$     | Accuracy | $\xi$     | Accuracy |
> |-----------|----------|-----------|----------|
> | -1.000    | 1.0000   | 0.206     | 0.9918   |
> | -0.849    | 1.0000   | 0.357     | 0.9836   |
> | -0.698    | 0.9918   | 0.508     | 0.9754   |
> | -0.548    | 0.9918   | 0.658     | 0.9590   |
> | -0.397    | 0.9918   | 0.809     | 0.9016   |
> | -0.246    | 0.9918   | 0.960     | 0.7623   |
> | -0.095    | 0.9918   | 1.111     | 0.5492   |
> | 0.055     | 0.9918   | 1.261     | 0.5492   |
>
> >Q2. Lack of Quantification for the extent of the Substantial Shift. The paper claims efficacy even under substantial domain discrepancies, but the degree of "substantial" is not well quantified. The analysis shows that an optimal $\xi$
>  exists such that $ {ER}(\beta_s)< {ER}_0$, but how does this performance gain degrade as a function of the domain shift? A potential weakness is that the paper does not introduce an explicit metric to quantify domain shift. The discrepancy between domains is only implicitly reflected through covariance geometry.
>
> We appreciate this comment. Our responses are summarized as follows:
>
> + First,
> in certain special cases, it is indeed possible to identify explicit quantities that reflect the extent of domain shift and characterize the gain obtained from KD.
> For example,
> in the setting considered in Corollary 3 of the revised manuscript (corresponding to Theorem 3 in the original version),
> we can define such quantities
> $d-1=\frac{1}{M}{Tr} {\Sigma}\_2 {\Sigma}\_1^{-1}-1$ and
> $g=\gamma^{ {T}}\Sigma_2\gamma.$
> Up to an asymptotically negligible term ($o_{a.s.}(1)$), the excess risk takes the form
> $$
>  {ER}=g+(1-\xi)^2\sigma^2\frac{M}{N_2-M}+\xi^2\sigma^2\frac{M}{N_1-M}\underbrace{\frac{1}{M}    {Tr} {\Sigma}_2 {\Sigma}_1^{-1}}_d.          \quad (1)
> $$
> Denote $a=\frac{M}{N\_2-M}$, and $b=\frac{M}{N\_1-M}.$
> The optimal imitation parameter is then given by
> $
> \xi^\*=\frac{a\sigma^2}{g+\sigma^2(a+bd)},
> $
> and the corresponding gain from  KD is
> $$
>    U(g,d)\triangleq {ER}\_0-{ER}( {\beta\_s})|\_{\xi=\xi^\*}=\frac{\sigma^4a^2}{g+\sigma^2(a+bd)}>0.
> $$
> It is easy to check that $\partial\_g U<0$, and $\partial_d U<0$.  Equation (1) implies that larger values of $d$ or $g$ lead to smaller gains from KD. Intuitively, $(d-1)$ can be viewed as a measure of covariate shift, while $g$ quantifies  the extent of model shift.

---

> ### Author Response · Authors · 2025-11-21
>
> + Second, with respect to the reviewer's question -- ``how does this performance gain degrade as a function of the domain shift?'' -- the above equation (1) reveals that domain shift does not always degrade performance.  In certain regimes (e.g., when $d<1$), it can even be beneficial, provided that an appropriate $\xi$ is chosen. For instance, when $\Sigma_2=I$, the covariate shift is larger for $\Sigma_1=4I$ than for $\Sigma_1=2I$; nevertheless, the performance gain at the optimal $\xi$  can be greater in the former case.
>
> + Finally,  in the more general scenario, it is not possible  to define a universal  measure analogous to  $(d-1)$ such that ER can be expressed as an explicit  function of it. As our theoretical analysis shows,  the limiting ER depends on the joint geometry of $\Sigma_1,\Sigma_2,\beta_1$ and $\beta_2$ in a complex manner, involving interactions between the signal directions $ {\beta}_i,i=1,2,$ and the covariance structures $ {\Sigma}_j,j=1,2$. These quantities together fully characterize the distributions of the source and target domains under our modeling assumptions.
> We have added a discussion at the end of Section 3.1 (highlighted in red in the revised manuscript) to further illustrate this point. The matrix $\mathbf{E}_5=\mathbf{\Pi}_2\mathbf{\Sigma}_2$ is introduced to simplify notation. We reproduce it below for the reviewer's convenience:
>
> ---
> *We provide an illustrative example here. Suppose that  $\mathbf{\Sigma}_i$ admits the spectral decomposition $\mathbf{\Sigma}\_i=\mathbf{U}\_i\mathbf{\Lambda}\_i\mathbf{U}\_i^{\mathsf{T}} $, for $i=1,2$. Consider the term $\boldsymbol{\beta}\_1^{\mathsf{T}}\mathbf{\Pi}\_1\mathbf{E}\_5\boldsymbol{\beta}\_2 $, which can be expressed as*
>
> $$
> \boldsymbol{\beta}\_1^{\mathsf{T}}(\lambda\_{\mathsf{t}}+\lambda\_{\mathsf{t}}m\_1\mathbf{\Sigma}\_1)^{-1}(\lambda\_{\mathsf{s}}+\lambda\_{\mathsf{s}}m\_2\mathbf{\Sigma}\_2 )^{-1}\mathbf{\Sigma}\_2\boldsymbol{\beta}\_2=(\lambda\_{\mathsf{s}}\lambda\_{\mathsf{t}})^{-1}\tilde{\boldsymbol{\beta}}\_1^{\mathsf{T}}(1+m\_1\mathbf{\Lambda}\_1)^{-1} \mathbf{U}\_1^{\mathsf{T}}\mathbf{U}\_2\tilde{\mathbf{\Lambda}}\_2\tilde{\boldsymbol{\beta}}\_2,\quad (2)
> $$
> *where $\tilde{\mathbf{\Lambda}}\_2$ is a diagonal matrix with entries $\tilde{\mathbf{\Lambda}}\_{2,jj}=\frac{\mathbf{\Lambda}\_{2,jj} }{1+m\_2\mathbf{\Lambda}\_{2,jj}}$. The vector $\tilde{\boldsymbol{\beta}}\_i=\mathbf{U}\_i\boldsymbol{\beta}\_i$ captures the alignment between $\boldsymbol{\beta}_i$ and the eigenvectors of $\mathbf{\Sigma}\_i$.
> The right-hand side of  (2)  explicitly reveals how the term
> $\boldsymbol{\beta}\_1^{\mathsf{T}}\mathbf{\Pi}\_1\mathbf{E}\_5\boldsymbol{\beta}\_2$  depends on $\tilde{\boldsymbol{\beta}}\_i$, the eigenvalues of $\mathbf{\Sigma}_1$ and $\mathbf{\Sigma}_2$, and the eigenvector overlap  $\mathbf{U}_1^{\mathsf{T}} \mathbf{U}_2$ between the two covariance matrices. In the special case where each $\boldsymbol{\beta}\_i$ is aligned with an eigenvector of $\mathbf{\Sigma}\_i$ -- for simplicity, suppose it corresponds to the first eigenvector -- the expression (2) further simplifies to*
>
> $$ \boldsymbol{\beta}\_1^{\mathsf{T}}\boldsymbol{\beta}\_2(\lambda\_{\mathsf{s}}\lambda\_{\mathsf{t}})^{-1}\frac{\mathbf{\Lambda}\_{2,11}}{(1+m\_1\mathbf{\Lambda}\_{1,11})(1+m_2\mathbf{\Lambda}\_{2,11})},
> $$
> *which depends on the eigenvalues of $\mathbf{\Sigma}_i$ and the  inner product $\boldsymbol{\beta}_1^{\mathsf{T}}\boldsymbol{\beta}_2$.*
>
> ---
>
> + Furthermore, we would like to mention that even when the true parameters $\beta\_1,\beta\_2,\Sigma\_1$ and $\Sigma\_2$ are unknown, it is still possible to empirically estimate the gain from KD at the optimal imitation parameter $\xi^\*$, given by $ER_0-ER(\beta_s^{\xi^*})$. This allows one to quantitatively evaluate the performance gain from KD in practice. For each pair $(\lambda_t,\lambda_s)$, let $\beta_{s}^{\xi}$ denote the student model corresponding to imitation parameter $\xi$.
> The estimation procedure proceeds as follows:
>
>     **Step 1**. We select a small subset of the target domain of size $n_2$ as a validation set, where $n_2\to \infty$ and $n_2=o(N_2)$ and we denote the validation  set by $\\{x_i,y_i\\}\_{i=1}^{n\_2}. $
>
>     **Step 2**. Let the student's predictions on the validation set be $\hat{y}\_{\xi,i}=x\_i^{ {T}}  \beta\_{s}^{\xi}$. By the law of large number, the  excess risk can be estimated consistently by $\widehat{ER}(\beta\_s^\xi)=\frac{1}{n\_2}\sum\_{i=1}^{n\_2}|\hat{y}\_{\xi,i}-y\_i|^2.$ Therefore, $ER(\beta_s^{\xi^*})-ER_0$ can be estimated by $\min_\xi \widehat{ER}(\beta_s^\xi) $ consistently  by law of large number.

---

> ### Author Response · Authors · 2025-11-21
>
> >The finding that $\xi<0$
>  (anti-learning) can be optimal (Corollary 2) is intriguing but seems rather uncommon in practical scenarios. Does this imply that the teacher performs so poorly that the student benefits from learning the opposite of the guidance? A more in-depth discussion on the underlying reasons for this phenomenon and its practical implications would further strengthen the paper.
>
> We thank the reviewer for recognizing the intriguing nature of our finding that $\xi<0$ can be optimal. We agree with your insightful comment that such a negative $\xi$ may result from the teacher’s poor performance. Our newly added real-data analysis in the response to your Question 1 further supports this phenomenon.
>
> Moreover, a negative $\xi$ may appear even when the teacher's performance is not ``poor''. To illustrate this, we refer to Corollary 2, which establishes the following sufficient condition for obtaining a negative $\xi$:
>
> $$
> \lambda_{ {s}}\lambda_{ {t}} \underline{m}_1   \mathrm{SNR}-\frac{M}{N_2}>0.
> $$
>
>  Recall that $\tilde{\sigma}^2=\\|\beta\\|^2+o_{a.s.}(1)$, and $\sigma^2$ denotes the noise variance. Under the settings of Corollary 2, where $\Sigma_1=\Sigma_2=I_M$ and $\beta=\beta_1=\beta_2\sim N(0,I_M/M)$, we can express SNR=$
> \tilde{\sigma}^2/\sigma^2$.  When this ratio is sufficiently large, a negative $\xi$ arises naturally, even without any degradation in teacher quality. The underlying reason is that, in this setting, the target-domain data already provide a highly informative signal and can yield a reliable estimator on their own. Incorporating the teacher's  information in such a regime may inadvertently introduce additional bias or variance through its predictions.

---

### Official Review · Reviewer_VYeN · 2025-11-10

**Soundness:** 3
**Presentation:** 3
**Contribution:** 3
**Rating:** 6
**Confidence:** 2

**Summary:**

This paper presents a theoretical framework for \textbf{cross-domain knowledge distillation (KD)} under both \emph{covariate} and \emph{model shifts} in a teacher--student ridge regression setting.
Using tools from \textbf{random matrix theory}, it derives high-dimensional asymptotic expressions for the student's excess risk via bias--variance decomposition.
The analysis shows that even under substantial domain shift, there exists an optimal imitation parameter $\xi$ such that the student model outperforms the student-only baseline, and the generalization risk exhibits a clear \textbf{double descent} behavior.

**Strengths:**

The paper provides a rigorous theoretical analysis of cross-domain knowledge distillation (KD) using random matrix theory, deriving precise high-dimensional asymptotic characterizations that extend previous student-only or fixed-ξ formulations. It analytically demonstrates that an appropriately chosen imitation parameter ξenables the student model to outperform the baseline even under significant domain shifts. Moreover, simulation results closely align with the theoretical predictions, validating the framework and revealing a clear double-descent behavior in the student’s excess risk.

**Weaknesses:**

1. The theoretical analysis is limited to linear ridge regression.
2. Relies on bounded spectral norms, independence between domains, and high-moment conditions, potentially unrealistic for real-world KD settings.
3. The ridgeless regression analysis only covers the under-parameterized case (M<N_1,N_2).
4. The dependence of the optimal imitation parameter ξ and the interaction between λ_tand λ_s lacks intuitive or empirical guidance.
5. Experiments are entirely synthetic, with no demonstrations on real KD applications (e.g., vision or language models).

Minors:
1. Theorem 1 and related derivations are notation-heavy; adding a concise notation summary table would improve readability.

**Questions:**

N/A

---

> ### Author Response · Authors · 2025-11-21
>
> We are grateful to the your thorough and constructive evaluation of our work. We especially appreciate your acknowledgment that our results extend previous theoretical work on KD.
> Below we address the main concerns point-by-point.
>
> >The theoretical analysis is limited to linear ridge regression.
>
> Thanks for this comment. In the revised manuscript, we have made two major generalizations:
>
> + First, we have extended our original analysis of ridge regression ($\lambda>0$) and ridgeless regression ($\lambda=0$) in the under-parameterized setting ($M<N_1, N_2$) to a  comprehensive theoretical treatment of ridgeless regression without the under-parameterization restriction. As a result, our revised theory now covers both ridge and ridgeless linear regression across general dimensions and sample sizes. All newly added material, including the extended theoretical analysis in Section 3.3 and the corresponding proofs in Appendix B.5, is highlighted in the revised manuscript for ease of reference. For your convenience, we summarize the key additions below.
>
> ---
> *In this section, we  consider the
>  minimum $\ell_2$ norm least squares (ridgeless) regression estimator. Specifically, the teacher model is defined by*
> $$
> \boldsymbol{\beta}\_{\mathsf{t}}=(\mathbf{X}\_1\mathbf{X}\_1^{\mathsf{T}} )^+\mathbf{X}\_1\mathbf{y}\_1,
> $$
> *where $(\mathbf{X}\_1\mathbf{X}\_1^{\mathsf{T}})^+$  denotes the Moore-Penrose inverse of $\mathbf{X}\_1\mathbf{X}\_1^{\mathsf{T}}. $
> Similarly, the ridgeless estimator of $\boldsymbol{\beta}\_{\mathsf{s}}$ takes the form*
> $$
> \boldsymbol{\beta}\_{\mathsf{s}}(\mathbf{X}\_2\mathbf{X}\_2^{\mathsf{T}} )^+[\xi \mathbf{X}\_2\mathbf{X}\_2^{\mathsf{T}}\boldsymbol{\beta}\_\mathsf{t}+(1-\xi)\mathbf{X}\_2\mathbf{y}\_2].
> $$
> **Theorem 3 in the revised paper**.
> *(1)  Suppose $\boldsymbol{\beta}\_1,\boldsymbol{\beta}\_2$ are deterministic, and Assumptions 1-2 hold. We further assume*
> $$
> \left|\frac{M}{N\_i}-1\right|\ge \tau,\quad \tau\le \sigma\_{\min}(\mathbf{\Sigma}\_i)\le \cdots\le \sigma\_{\max}(\mathbf{\Sigma}\_i)\le \tau^{-1}, \text{ for }i=1,2.
> $$
> *Define $f(\lambda)=\widehat{\mathbf{Bias}}$ and $g(\lambda)=\widehat
> {\mathbf{Var}}$, with $\lambda=\lambda_{\mathsf{s}}=\lambda_{\mathsf{t}}$,  where the expressions for $\widehat{\mathbf{Bias}}$ and $\widehat{\mathbf{Var}}$  are provided in (7) and (8), respectively.
> We have*
> $$
> \mathbf{Bias}=f(0^+)+o_{a.s.}(1),\quad \mathbf{Var}=g(0^+)+o_{a.s.}(1). \quad (1)
> $$
> *(2) Suppose $\boldsymbol{\beta}=\boldsymbol{\beta}_1=\boldsymbol{\beta}_2$ are random and Assumptions 1-3 hold.
> Then,
> the estimated expressions in (1) still hold with $f(\lambda)$ replaced by the $\widehat{\mathbf{Bias}}$ defined in Theorem 2.*
>
> ---
>
> + Second, we extend our investigation to nonlinear models. We anticipate that the key phenomena revealed by our theory may persist beyond the linear regime. To support this claim, we have added a new Section 5.1 describing the experimental setup for nonlinear models. The corresponding empirical results are presented in Figure 4 of Appendix C.1 in the revised manuscript. These simulations provide strong evidence that the mechanisms identified in the linear regime are robust and likely to generalize to more complex nonlinear settings. For your convenience, we summarize the experimental setup below; this same description appears in the newly added Section 5.
>
> ---
> *We assume the source data $(\mathbf{x}_j^{(1)},{y}_j^{(1)} )\_{j=1}^{N_1} $ are generated i.i.d. according to*
>
> \\[
> y_j^{(1)} =f(\mathbf{x}_j^{(1)} )+\varepsilon^{(1)}_j,\text{ for } 1\le j\le N_1.
> \\]
>
> *The target data $\{(\mathbf{x}_j^{(2)},y_j^{(2)})\} \_{j=1}^{N_2} $ are generated according to*
> \\[
> y_j^{(2)} =\tilde{f}(\mathbf{x}_j^{(2)} )+\varepsilon_j^{(2)},\text{ for } 1\le j\le N_2
> \\]
> *Suppose $\mathbf{x}_j^{(1)}\sim D_1,1\le j\le N_1$ and $\mathbf{x}_j^{(2)}\sim D_2,1\le j\le N_2. $ We refer to the case $D_1\neq D_2$ as a covariate shift, and the case where $f\neq \tilde{f}$ as a model shift.*
>
> *We consider learning the unknown function using a fully connected two-layer neural network with $n$ hidden neurons:
> $f_{\mathrm{NN}}(\mathbf{x})=\mathbf{a}^{\mathsf{T}}\sigma(\mathbf{Wx}),$ where $\mathbf{W}\in \mathbb{R}^{n\times M} $ is the  weight matrix, and $\sigma(\cdot)$ is an activation function applied entrywise.
> When the random weight matrix $\mathbf{W}$ is fixed and only the second-layer weight $\mathbf{a}$ is  optimized, the model reduces to a kernel regression model, where the kernel defined by $\mathbf{x}\to \sigma(\mathbf{Wx})$ is referred to as the conjugate kernel. The teacher model is given by*
> \\[
> f\_{\mathrm{NN}}^{\mathsf{t}}(\mathbf{x})=\mathbf{a}\_\mathsf{t}^{\mathsf{T}}\sigma(\widetilde{\mathbf{W}}\_1\mathbf{x}),
> \\]
> *with*
> $$
> \mathbf{a}\_{\mathsf{t}}=\arg\min\_{\mathbf{a}}\bigg\\{\frac{1}{N\_1}\\|\mathbf{y}\_1-\sigma(\mathbf{X}\_1^{\mathsf{T}}\widetilde{\mathbf{W}}\_1^{\mathsf{T}}) \mathbf{a}\\|^2+\lambda\_{\mathsf{t}} \\|\mathbf{a}\\|^2  \bigg\\}.
> $$
>
>
>
> ---

---

> ### Author Response · Authors · 2025-11-21
>
> ---
> *We use $f\_{\mathrm{NN}}^{\mathsf{t}}(\mathbf{x}\_j^{(2)})$ together with the covariates $\\{\mathbf{x}\_j^{(2)}\\}\_{j=1}^{N\_2}$ to generate predictions $\mathbf{y}\_2^{\mathsf{t}}$. Then the student model is finetuned on the target domain data and $\mathbf{y}\_2^{\mathsf{t}}$. The student model takes the form
> $f\_{\mathrm{NN}}^{\mathsf{s}}(\mathbf{x})=\mathbf{a}\_{\mathsf{s}}^{\mathsf{T}}\sigma(\mathbf{W}\_1\mathbf{x})$
> with*
> $$
> \mathbf{a}\_{\mathsf{s}}=\arg\min\_{\mathbf{a}}\xi\bigg(\frac{1}{N\_2} \\| \mathbf{y}\_2^{\mathsf{t}}-\sigma(\mathbf{X}\_2^{\mathsf{T}}\mathbf{W}\_1^{\mathsf{T}} )\mathbf{a} \\|^2 \bigg)+(1-\xi)\bigg( \frac{1}{N\_2} \\|\mathbf{y}_2-\sigma(\mathbf{X}\_2^{\mathsf{T}}\mathbf{W}\_1^{\mathsf{T}} )\mathbf{a} \\|^2\bigg)+\lambda\_{\mathsf{s}} \\|\mathbf{a} \\|^2.
> $$
> *We also examine a setting where the teacher model is a deeper neural network. Specifically, while keeping the student model fixed, we let the teacher be a 4-layer fully connected network:*
>
> $$
> f\_{\mathrm{NN}}^{\mathsf{t}}=\mathbf{a}\_{\mathsf{t}}^{\mathsf{T}}\sigma(\widetilde{\mathbf{W}}\_3\sigma(\widetilde{\mathbf{W}}\_2\sigma(\widetilde{\mathbf{W}}\_1\mathbf{x}))),
> $$
>
> *where*
>
> $$
> \mathbf{a}\_{\mathsf{t}}=\arg\min\_{\mathbf{a}}\frac{1}{N\_1}\\|\mathbf{y}_1-[\sigma(\widetilde{\mathbf{W}}\_3\sigma(\widetilde{\mathbf{W}}\_2\sigma(\widetilde{\mathbf{W}}\_1\mathbf{X}\_1))]^{\mathsf{T}}\mathbf{a}\\|^2+\lambda\_{\mathsf{t}}\\|\mathbf{a}\\|^2.
> $$
>
> *We set $D_1=\mathcal{N}(0,4\mathbf{I}_M)$ and $D_2=\mathcal{N}(0,\mathbf{I}_M)$. Let*
>
> $$
> f(\mathbf{x})= (\boldsymbol{\beta}^{\mathsf{T}}\mathbf{x} )^2+1,\quad \tilde{f}(\mathbf{x})=(\boldsymbol{\beta}^{\mathsf{T}}\mathbf{x})^2.
> $$
>
> *Because $D\_1\neq D\_2$ and $f\neq \tilde{f}$, both covariate shift and model shift are present in this setting. More details and the
> numerical results are provided in Appendix C.1.*
>
> ---
>
> >Relies on bounded spectral norms, independence between domains, and high-moment conditions, potentially unrealistic for real-world KD settings.
>
> To address the reviewer's concerns regarding the assumptions on bounded spectral norms, domain independence, and high-moment conditions, we provide detailed responses to each of these three points below.
>
> + **Concern 1: Bounded spectral norms.**  In cases where the population covariance matrix exhibits diverging eigenvalues, our framework for ridge regression can still accommodate such behavior under certain assumptions on the divergence, through appropriate data preprocessing.  For instance, suppose $\mathbf{\Sigma}\_i$ has $n(M)$ diverging eigenvalues that are of the same order as $\sigma\_1=\\|\mathbf{\Sigma}\_i\\|$, while the remaining eigenvalues stay bounded, and assume that $n(M)/M\sim 1$. In this setting, scaling the covariates by $\hat{\sigma}\_1$ (with $\hat{\sigma}\_1\sim \sigma\_1$, which can be estimated from the data) yields a normalized covariance matrix. It can be verified that the resulting $\mathbf{\Sigma}\_i$
>   continues to satisfy Assumption 2, and therefore our theoretical results (Theorems 1 and 2) apply directly to the scaled model.
>
> + **Concern 2: Independence between domains.**  Following your suggestion, we have extended our theoretical analysis in the revised manuscript to accommodate dependent source and target domains, which more accurately reflect practical KD scenarios. Specifically, we have introduced two forms of domain dependence: Case 1, corresponding to weak dependence, and Case 2, corresponding to strong dependence. For your convenience, we summarize the corresponding setups and key findings below:
>
> ---
> ***Case 1 (Weak dependence)**: Assume that $\mathbf{X}_1$ exhibits weak dependence on  $\mathbf{X}_2$ in the following sense: $\mathbf{X}_1=\alpha\mathbf{X}_2+\widetilde{\mathbf{X}}_1 $, where $\alpha\to 0$ as $M\to \infty$, and $\widetilde{\mathbf{X}}_1$ is independent of $\mathbf{X}_2$ with the form $\widetilde{\mathbf{X}}_1=\mathbf{\Sigma}_1^{1/2}\mathbf{Z}_1$. It follows that $\mathrm{Cov}(\mathbf{x}_j^{(1)},\mathbf{x}_j^{(2)} )=\mathrm{Cov}(\mathbf{x}_j^{(2)},\alpha\mathbf{x}_j^{(2)} )=\alpha\mathbf{\Sigma}_2$. We refer to this regime as weak dependence between domain covariates.*
>
> ***Proposition 1** Suppose that $\widetilde{\mathbf{X}}_{1},\mathbf{X}_2,\boldsymbol{\varepsilon}_1$ and $\boldsymbol{\varepsilon}_2$ satisfy Assumptions 1-2. Then, Theorem 1 continues to hold. Moreover, if we additionally impose Assumption 3, then Theorem 2  remains valid.*
>
> ***Case 2 (Strong dependence):** Suppose the matrix $\mathbf{X}_1$ follows a signal-plus-noise structure: $\mathbf{X}_1=\mathbf{X}_2+\mathbf{A}$, where $\mathbf{A}$ is a deterministic signal matrix satisfying $\\|\mathbf{A}\\|=o(\sqrt{M}).$ This model captures realistic domain adaptation scenarios in which the source and target domains share a common underlying data matrix but differ by a small deterministic shift -- such as a weak shared signal across features in source domain.*
>
> ---

---

> ### Author Response · Authors · 2025-11-21
>
> ---
> ***Proposition 3**  Suppose $\mathbf{X}_1=\mathbf{X}_2+\mathbf{A}$ is a signal-plus-noise data matrix, with $\\|\mathbf{A}\\|=o(\sqrt{M})$. The regression parameter vector $\boldsymbol{\beta}=\boldsymbol{\beta}\_1=\boldsymbol{\beta}\_2$  satisfies Assumption 3.
> When $\lambda\_{\mathsf{s}}\neq \lambda\_{\mathsf{t}}$, we have*
> $$
> \widehat{\mathbf{Bias}}=\frac{a}{M}\mathrm{Tr}\mathbf{\Sigma}\_2[\mathbf{\Pi}\_2(-\lambda\_{\mathsf{t}} )-\mathbf{\Pi}\_2(-\lambda\_{\mathsf{s}})]+\frac{b}{M}\mathrm{Tr}\mathbf{\Sigma}\_2\mathbf{\Pi}\_2'(-\lambda\_{\mathsf{t}})+\frac{c}{M}\mathrm{Tr}\mathbf{\Sigma}\_2\mathbf{\Pi}\_2'(-\lambda\_{\mathsf{s}}),
> $$
> *and*
> $$
> \widehat{\mathbf{Var}}=\frac{\xi^2\sigma^2}{N\_1}\bigg(d\mathrm{Tr}\mathbf{\Sigma}\_2[\mathbf{\Pi}\_2(-\lambda\_{\mathsf{t}})-\mathbf{\Pi}\_2(-\lambda\_{\mathsf{s}})]+e\mathrm{Tr}\mathbf{\Sigma}\_2\mathbf{\Pi}\_2'(-\lambda\_{\mathsf{t}})+f\mathrm{Tr}\mathbf{\Sigma}\_2\mathbf{\Pi}\_2'(-\lambda\_{\mathsf{s}})\bigg)\\
> +(1-\xi)^2\sigma^2\frac{1}{N\_2}\mathrm{Tr}\mathbf{\Sigma}\_2[\mathbf{\Pi}\_2(-\lambda\_{\mathsf{s}})-\lambda\_{\mathsf{s}}\mathbf{\Pi}\_2'(-\lambda\_{\mathsf{s}})],
> $$
> *where*
> $$
> a=\frac{2\xi \lambda\_1\lambda\_2}{\lambda\_{\mathsf{s}}-\lambda\_{\mathsf{t}} }+\frac{2\xi\lambda\_\mathsf{t}\lambda\_{\mathsf{s}}( \xi\lambda\_{\mathsf{\_{\mathsf{t}}}}-\lambda\_{\mathsf{s}} ) }{(\lambda\_{\mathsf{s}}-\lambda\_{\mathsf{t}})^2}-\frac{2\xi\lambda\_{\mathsf{s}}^2\lambda\_{\mathsf{t}}^2 }{(\lambda\_{\mathsf{s}}-\lambda\_{\mathsf{t}})^3}, \;b=\xi^2\lambda\_{\mathsf{t}}^2-\frac{2\xi^2\lambda\_{\mathsf{t}}^2\lambda\_{\mathsf{s}} }{\lambda\_{\mathsf{t}}-\lambda\_{\mathsf{s}}},
> $$
>
> $$
>  c=\lambda_{\mathsf{s}}^2-\frac{2\xi \lambda_{\mathsf{t}}\lambda_{\mathsf{s}}^2}{\lambda_{\mathsf{t}}-\lambda_{\mathbf{s}}},\; d=\frac{2\lambda_{\mathsf{s}}}{\lambda_{\mathsf{t}}-\lambda_{\mathsf{s}}}+\frac{\lambda_{\mathsf{s}}^2}{(\lambda_{\mathsf{t}}-\lambda_{\mathsf{s}})^2}+\frac{\lambda_{\mathsf{t}}\lambda_{\mathsf{s}}^2}{(\lambda_{\mathsf{s}}-\lambda_{\mathsf{t}})^3}+\frac{2\lambda_{\mathsf{s}}\lambda_{\mathsf{t}}}{(\lambda_{\mathsf{t}}-\lambda_{\mathsf{s}})^2},
> $$
> $$
> e=-\lambda_{\mathsf{t}}+ \frac{2\lambda_{\mathsf{s}}\lambda_{\mathsf{t}}}{(\lambda_{\mathsf{t}}-\lambda_{\mathsf{s}})^2}-\frac{\lambda_{\mathsf{t}}\lambda_{\mathsf{s}}^2}{(\lambda_{\mathsf{s}}-\lambda_{\mathsf{t}})^2},\quad f=\frac{\lambda_{\mathsf{s}}^2}{\lambda_{\mathsf{s}}-\lambda_{\mathsf{t}}}-\frac{\lambda_{\mathsf{t}}\lambda_{\mathsf{s}}^2}{(\lambda_{\mathsf{s}}-\lambda_{\mathsf{t}})^2}.
> $$
> *When $\lambda=\lambda_{\mathsf{s}}=\lambda_{\mathsf{t}}$,  $\widehat{\mathbf{Bias}}$ is given in (54) and*
> $$
> \widehat{\mathbf{Var}}=(1-\xi)^2\sigma^2\frac{1}{N\_2}\mathrm{Tr}\mathbf{\Sigma}\_2(\mathbf{\Pi}\_2+\lambda\mathbf{\Pi}\_2')+\frac{\xi^2\sigma^2}{N\_1}\mathrm{Tr}\mathbf{\Sigma}\_2[\mathbf{\Pi}\_2-3\lambda\mathbf{\Pi}\_2'+3\lambda^2\mathbf{\Pi}\_2^{(2)}-\lambda^3\mathbf{\Pi}\_2^{(3)}],
> $$
> *with* $\mathbf{\Pi}\_2^{(k)}=\frac{\mathrm{d}^k\mathbf{\Pi}\_2(z)}{\mathrm{d}z^k}\big|\_{z=-\lambda}.$
>
> ---
> + **Concern 3: Moment conditions**. In the original manuscript, we assumed that the entries of $\mathbf{Z}_i$ (where $\mathbf{X}_i=\mathbf{\Sigma}_i^{1/2}\mathbf{Z}_i$) have moments of all orders -- mainly for technical convenience and to simplify the exposition. However, as noted after Assumption 1, this requirement can be relaxed.  In particular, our analysis only necessitates the existence of moments of order
> $(8+c)$ for any positive constant $c>0$. We have replaced Lemma 5 in the original manuscript with Lemma 6 in the revised version to ensure applicability under the weaker moment condition.   To enhance clarity, we have now explicitly stated this relaxed moment condition following Assumption 1 in the revised manuscript (highlighted in red). Furthermore, at the end of Appendix B.1, we have added a new Remark detailing how the moment assumption can be relaxed. For your convenience, we  reproduce the remark below:
>
> ---
>
> ***Remark**
> To relax the moment assumption, we apply a standard truncation argument commonly used in random matrix theory (e.g., [1]).  This approach allows us to employ Lemma 4 under the weaker finite $(8+c)$-th moment condition, introducing only a negligible additional error term that depends on $M$ but does not affect the leading-order asymptotics of our results.
> Moreover, a careful examination of the proofs shows that the same moment condition is also sufficient to establish Lemma
> 7. Consequently, all of our theoretical conclusions remain valid under this relaxed assumption.*
>
> ---
> [1] Yang F et al. ``Precise high-dimensional asymptotics for quantifying heterogeneous transfers''. JMLR 2025.
>
> > The ridgeless regression analysis only covers the under-parameterized case ($M<N_1,N_2$).
>
> We have removed this restriction in the revised manuscript. A detailed explanation can be found in our response to your Question 1.

---

> ### Author Response · Authors · 2025-11-21
>
> > The dependence of the optimal imitation parameter $\xi$ and the interaction between $\lambda_t$ and  $\lambda_s$ lacks intuitive or empirical guidance.
>
> We appreciate this comment. In the original manuscript -- specifically, at the end of the Appendix (now Section C.3 in the revised version) -- we  included visualizations of the empirical excess risk of the student model as a function of $\lambda_t$
>   and $ \lambda_s$ for different values of $\xi$. From these visualizations, several empirical patterns emerge. In particular, when $\xi>1$, the influence of $\lambda_{\mathsf{t}}$ becomes more pronounced, whereas for  $\xi=-0.5$, $\lambda_{\mathsf{s}} $ almost dominates the variation of the excess risk, indicating a diminished effect of the teacher's guidance (anti-learning against the teacher's supervision).
>
> Following your suggestion, we now provide empirical guidance for selecting the optimal triple $(\xi,\lambda_t,\lambda_s)$.  Given $(\lambda_t,\lambda_s)$, let $\xi^*(\lambda_t,\lambda_s)$ denote the corresponding optimal imitation parameter. We assume that $(\lambda_t,\lambda_s)$ lies in a compact subset  $\mathbf{\Lambda}\subset \mathbb{R}^{+}\times \mathbb{R}^{+}$ (e.g., $[C^{-1},C]\times [C^{-1},C] $
>   for some constant $C$), which is standard in hyperparameter tuning. Our procedure proceeds in the following three steps:
>
> + **Step 1.** We randomly select a small subset of the target domain of size $n_2$ as a validation set, where $n_2\to \infty$ and $n_2=o(N_2)$.  We denote this validation  set by $\\{x_i,y_i\\}_{i=1}^{n_2}. $
>
> + **Step 2.** Let $\beta_{s}^{\xi}$ be the student model corresponding to imitation parameter $\xi$, and denote its predictions on the validation set by $\hat{y}\_{\xi,i}=x\_i^{\mathsf{T}}  \beta\_{s}^{\xi}$.
>
>     Let $A$ be a fine grid covering the compact hyperparameter space $\mathbf{\Lambda}$. For each pair $(\lambda\_t,\lambda\_s )\in A$, we estimate $\xi^\*(\lambda\_t,\lambda\_s)$ by solving $\hat{\xi}^{\*}(\lambda\_t,\lambda\_s)=\arg\min\_{\xi}\widehat{ER}(\beta\_s^{\xi})=\arg\min\_{\xi} \frac{1}{n\_2}\sum\_{i=1}^{n\_2}|\hat{y}\_{\xi,i}-y\_i|^2. $
>     By the law of large number, $\hat{\xi}^{\*}(\lambda\_t,\lambda\_s) $ is a consistent estimation of $\xi^{\*}(\lambda\_t,\lambda\_s).$
>
> + **Step 3.**  Finally, we we identify the optimal hyperparameter triple by minimizing the empirical excess risk over all candidate pairs:
> $$
> (\hat{\xi}^{\*}(\lambda\_t^{\*},\lambda\_s^{\*}),{\lambda}\_t^{\*},{\lambda}\_s^{\*})=\arg\min\_{({\lambda}\_t,{\lambda}\_s)\in A}\widehat{ER}(\beta\_s^{\hat{\xi}^{\*}(\lambda\_t,\lambda\_s)}).
> $$
>
> > Experiments are entirely synthetic, with no demonstrations on real KD applications (e.g., vision or language models).
>
> Following the reviewer's suggestion, we now apply our approach to the Human Activity Recognition Using Smartphones dataset from the commonly used UCI repository (available at https://archive.ics.uci.edu/dataset/240/human+activity+recognition+using+smartphones). This dataset contains 561 dimensional sensor signals collected from 30 individuals performing various activities.  We are interested in a binary classification task, where the **Active** class includes walking, walking upstairs, walking downstairs, and standing, while the **Rest** class includes sitting and laying.
>
> We treat the data from Subject 25 (the largest individual dataset) as the target domain. The target dataset is divided into a training set (the last 70\%, containing 286 samples) and a testing set (the first 30\%, containing 123 samples). The remaining 29 subjects (a total of 9,890 samples) constitute the source domain.  We set $\lambda\_t=\lambda\_s=0.1$. Denote the training data by $( {X}\_2^{\mathsf{T}}, {y})$ and the test data by $(X\_{t,2}^{\mathsf{T}} ,y\_t )$, where $ {X}\_2=( {x}\_1,..., {x}\_{N\_2})$, $X\_{t,2}=(x\_{t,1},...,x_{t,n\_2})$ with $N\_2=286, n\_2=123$.  To obtain soft labels in [0,1], we apply the sigmoid function to teacher's predictions:
> $$ {y}\_2^{{t}}=\text{sigmoid}( {\beta}\_{t}^{\mathsf{T}}  {X}_{2} ),$$
> where the sigmoid function is applied entrywise.
> The student estimator $ {\beta}\_{\mathsf{s}}$ then takes the form:
> $$
>  {\beta}\_{s}=\frac{1}{N_2} {Q}\_2(\xi {X}\_2 {y}_2^{t}+(1-\xi) {X}\_2 {y} ).
> $$
>
> For each test sample, we apply the sigmoid function to the predicted output and assign label 1 if the value exceeds 0.5, and 0 otherwise. The results for different values of $\xi$ are summarized in Table 1 below, which indicate that **anti-learning against the teacher’s supervision $(\xi<0)$ may lead
> to better generalization ability**. Remarkably, the student achieves perfect classification accuracy when $\xi=-1$ or $\xi=-0.849$.  This finding provides empirical evidence supporting the necessity of optimizing $\xi$ over $\mathbb{R}$.

---

> ### Author Response · Authors · 2025-11-21
>
> Table 1: Test accuracy vs. imitation parameter $\xi$.
> | $\xi$     | Accuracy | $\xi$     | Accuracy |
> |-----------|----------|-----------|----------|
> | -1.000    | 1.0000   | 0.206     | 0.9918   |
> | -0.849    | 1.0000   | 0.357     | 0.9836   |
> | -0.698    | 0.9918   | 0.508     | 0.9754   |
> | -0.548    | 0.9918   | 0.658     | 0.9590   |
> | -0.397    | 0.9918   | 0.809     | 0.9016   |
> | -0.246    | 0.9918   | 0.960     | 0.7623   |
> | -0.095    | 0.9918   | 1.111     | 0.5492   |
> | 0.055     | 0.9918   | 1.261     | 0.5492   |
>
> > Minors:
> Theorem 1 and related derivations are notation-heavy; adding a concise notation summary table would improve readability.
>
> We have added a summary of the key quantities used in Theorem 1 immediately before its statement in the revised manuscript.  The updated presentation now reads as follows:
>
> ---
> *The other related quantities are summarized in Table  2 (Table 1 in the revised manuscript)*.
> | Table 2: Some notations used in the theoretical results |
> |:--------:|
> | $\mathbf{E}_1 = \mathbf{\Pi}_1\mathcal{S}_1(\mathbf{\Sigma}_2)\mathbf{\Pi}_1, \quad \mathbf{E}_2 = \mathbf{\Pi}_1\mathcal{S}_1(\mathbf{\Pi}_2\mathcal{S}_2(\mathbf{\Sigma}_2)\mathbf{\Pi}_2)\mathbf{\Pi}_1$ |
> | $\mathbf{E}_3 = \mathbf{\Pi}_1\mathcal{S}_1(\mathbf{\Sigma}_2\mathbf{\Pi}_2)\mathbf{\Pi}_1, \quad \mathbf{E}_4 = \mathbf{\Pi}_2\mathcal{S}_2(\mathbf{\Sigma}_2)\mathbf{\Pi}_2, \quad \mathbf{E}_5 = \mathbf{\Sigma}_2\mathbf{\Pi}_2$ |
>
> *We now state our first main result.*
>
> **Theorem 1** *Let $\boldsymbol{\gamma}=\boldsymbol{\beta}\_1-\boldsymbol{\beta}\_2.$ For the deterministic vectors $\\|\boldsymbol{\beta}\_1\\|$ and $\\|\boldsymbol{\beta}\_2\\|$,
> assume that $\\|\boldsymbol{\beta}\_1\\|,\\|\boldsymbol{\beta}\_2\\|\le c$ for some constant $c.$ Under Assumptions 1-2, the following results hold:*
>
> $$
> \mathbf{Bias}=\widehat{\mathbf{Bias}}+o\_{a.s.}(1),\quad \mathbf{Var}=\widehat{\mathbf{Var}}+o\_{a.s.}(1),
> $$
>
> *where*
> $$
> \begin{aligned}
> \widehat{\mathbf{Bias}}&=\xi^2\boldsymbol{\beta}\_1^\mathsf{T}\big[\lambda\_{\mathsf{t}}^2\mathbf{E}\_1+\lambda\_{\mathsf{s}}^2\lambda\_{\mathsf{t}}^2\mathbf{E}\_2-2\lambda\_{\mathsf{t}}^2\lambda\_{\mathsf{s}}\mathbf{E}\_3\big]\boldsymbol{\beta}\_1+2\xi\boldsymbol{\beta}\_2^\mathsf{T}\big[\lambda\_{\mathsf{s}}^2 \mathbf{E}\_4 -\lambda\_{\mathsf{s}}\mathbf{E}\_5\big]\boldsymbol{\gamma}\\\\
> &+\lambda\_{\mathsf{s}}^2\boldsymbol{\beta}\_2^\mathsf{T}\mathbf{E}\_4\boldsymbol{\beta}\_2+2\xi\boldsymbol{\beta}\_1^\mathsf{T}\big[ \lambda\_{\mathsf{t}}\lambda\_{\mathsf{s}}\mathbf{\Pi}\_1\mathbf{E}\_5-\lambda\_{\mathsf{t}}\lambda\_{\mathsf{s}}^2\mathbf{\Pi}\_1\mathbf{E}\_4\big]\boldsymbol{\beta}\_2+\xi^2\boldsymbol{\gamma}^{\mathsf{T}}\big[-2\lambda\_{\mathsf{s}}\mathbf{E}\_5+\lambda\_{\mathsf{s}}^2\mathbf{E}\_4 \big]\boldsymbol{\gamma}\\\\
> &+2\xi^2\boldsymbol{\gamma}^\mathsf{T}\big[\lambda\_{\mathsf{s}}\lambda\_{\mathsf{t}} \mathbf{E}\_5\mathbf{\Pi}\_1 -\lambda\_{\mathsf{t}}\lambda\_{\mathsf{s}}^2 \mathbf{E}\_4\mathbf{\Pi}\_1 -\lambda\_{\mathsf{t}}\mathbf{\Sigma}\_2\mathbf{\Pi}\_1+\lambda\_{\mathsf{t}}\lambda\_{\mathsf{s}}\mathbf{E}\_5\mathbf{\Pi}\_1
> \big]\boldsymbol{\beta}\_1,
> \end{aligned}
> $$
>
> *and*
> $$
> \widehat{\mathbf{Var}}=\frac{\xi^2\sigma^2}{N\_1}\mathrm{Tr}\big[\big(\mathbf{\Sigma}\_2-2\lambda\_{\mathsf{s}}\mathbf{E}\_5+\lambda\_{\mathsf{s}}^2\mathbf{E}\_4\big)\big(\mathbf{\Pi}\_1-\lambda\_{\mathsf{t}}\mathbf{\Pi}\_1' \big)\big]+\frac{(1-\xi)^2\sigma^2}{N\_2}\mathrm{Tr}[\mathbf{E}\_5-\lambda\_{\mathsf{s}}\mathbf{\Sigma}\_2\mathbf{\Pi}\_2'].
> $$
>
> ---

---

### Meta-Review · Area_Chair_RtKz · 2026-01-08

**Summary:**

The paper provides a random-matrix-theoretic analysis of cross-domain knowledge distillation under covariate/model shift in (ridge/ridgeless) linear regression. Reviewers find the theory solid and the “anti-learning” phenomenon (negative imitation weight) interesting, but the discussion converges on limited practical relevance: the setting is narrow, and the empirical evidence (mostly synthetic, with only small add-ons in rebuttal) is not strong enough for ICLR. Overall, the work is technically good but still below the acceptance bar.

**Reviewer Concerns:**

The added “real-data” study is still a small-scale classification setup that does not resemble common KD applications (vision/language, large teachers, logits/temperature, etc.). The overall evaluation is still dominated by synthetic validation of asymptotic formulas, which leaves the practical message unclear.

The core results are derived for linear regression with specific high-dimensional asymptotics. The nonlinear add-on is suggestive but not a substitute for either (i) theory in a broader model class, or (ii) convincing empirical demonstrations in realistic KD settings.

Even with some relaxations, the framework still relies on modeling choices that may not match real KD under domain shift (feature distributions, teacher noise model, independence structure). Reviewers’ concerns here are only partially alleviated.

The negative imitation parameter is interesting, but it remains unclear how often it should occur in practice, how robust it is across tasks, and how it should be used as actionable guidance beyond the proposed validation procedure. In other words, the phenomenon is explained in the model, but still not convincingly connected to mainstream KD practice.

**Reviewer Scores:**

Reviewer VYeN (6: marginally above): The rebuttal addresses several concrete theoretical gaps (ridgeless regime, assumption relaxations, tuning guidance), but the reviewer’s original hesitation about practical validation is only partially resolved.

Reviewer aTNu (6: marginally above): Many requested presentation/technical clarifications were addressed (e.g., lemma on well-posedness for negative imitation weight, clearer asymptotic statements, eigen-overlap intuition moved up, closed forms for optimal parameters in special cases).

Reviewer AThT (4: marginally below): the broader “how useful is this for real KD?” concern remains.

Reviewer rJMR (4: marginally below): The reviewer’s core point about limited empirical validation and strong modeling assumptions is not fully removed.

---

### Decision · Program_Chairs · 2026-01-26

Reject